# LION: Latent Point Diffusion Models for 3D Shape Generation

**Xiaohui Zeng**[1,2,3,*]     **Arash Vahdat**[1]     **Francis Williams**[1]

**Zan Gojcic**[1]     **Or Litany**[1]     **Sanja Fidler**[1,2,3]     **Karsten Kreis**[1]

[1]NVIDIA     [2]University of Toronto     [3]Vector Institute

{xzeng,avahdat,fwilliams,zgojcic,olitany,sfidler,kkreis}@nvidia.com

## Abstract

Denoising diffusion models (DDMs) have shown promising results in 3D point cloud synthesis. To advance 3D DDMs and make them useful for digital artists, we require *(i)* high generation quality, *(ii)* flexibility for manipulation and applications such as conditional synthesis and shape interpolation, and *(iii)* the ability to output smooth surfaces or meshes. To this end, we introduce the hierarchical *Latent Point Diffusion Model (LION)* for 3D shape generation. LION is set up as a variational autoencoder (VAE) with a hierarchical latent space that combines a global shape latent representation with a point-structured latent space. For generation, we train two hierarchical DDMs in these latent spaces. The hierarchical VAE approach boosts performance compared to DDMs that operate on point clouds directly, while the point-structured latents are still ideally suited for DDM-based modeling. Experimentally, LION achieves state-of-the-art generation performance on multiple ShapeNet benchmarks. Furthermore, our VAE framework allows us to easily use LION for different relevant tasks: LION excels at multimodal shape denoising and voxel-conditioned synthesis, and it can be adapted for text- and image-driven 3D generation. We also demonstrate shape autoencoding and latent shape interpolation, and we augment LION with modern surface reconstruction techniques to generate smooth 3D meshes. We hope that LION provides a powerful tool for artists working with 3D shapes due to its high-quality generation, flexibility, and surface reconstruction. Project page and code: https://nv-tlabs.github.io/LION.

## 1 Introduction

Generative modeling of 3D shapes has extensive applications in 3D content creation and has become an active area of research [1–52]. However, to be useful as a tool for digital artists, generative models of 3D shapes have to fulfill several criteria: **(i)** Generated shapes need to be realistic and of high-quality without artifacts. **(ii)** The model should enable flexible and interactive use and refinement: For example, a user may want to refine a generated shape and synthesize versions with varying details. Or an artist may provide a coarse or noisy input shape, thereby guiding the model to produce multiple realistic high-quality outputs. Similarly, a user may want to interpolate different shapes. **(iii)** The model should output smooth meshes, which are the standard representation in most graphics software.

Existing 3D generative models build on various frameworks, including generative adversarial networks (GANs) [1–23], variational autoencoders (VAEs) [24–30], normalizing flows [31–34], autoregressive models [35–38], and more [39–44]. Most recently, denoising diffusion models (DDMs)

---

*Work done during internship at NVIDIA.

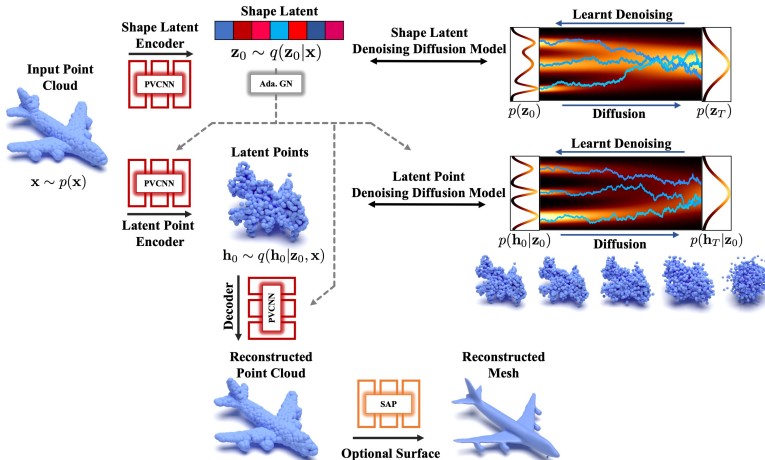

Figure 1: *LION* is set up as a hierarchical point cloud VAE with denoising diffusion models over the shape latent and latent point distributions. Point-Voxel CNNs (PVCNN) with adaptive Group Normalization (Ada. GN) are used as neural networks. The latent points can be interpreted as a smoothed version of the input point cloud. *Shape As Points* (SAP) is optionally used for mesh reconstruction.

have emerged as powerful generative models, achieving outstanding results not only on image synthesis [53–64] but also for point cloud-based 3D shape generation [45–47]. In DDMs, the data is gradually perturbed by a diffusion process, while a deep neural network is trained to denoise. This network can then be used to synthesize novel data in an iterative fashion when initialized from random noise [53, 65–67]. However, existing DDMs for 3D shape synthesis struggle with simultaneously satisfying all criteria discussed above for practically useful 3D generative models.

Here, we aim to develop a DDM-based generative model of 3D shapes overcoming these limitations. We introduce the *Latent Point Diffusion Model (LION)* for 3D shape generation (see Fig. 1). Similar to previous 3D DDMs, LION operates on point clouds, but it is constructed as a VAE with DDMs in latent space. LION comprises a hierarchical latent space with a vector-valued global shape latent and another point-structured latent space. The latent representations are predicted with point cloud processing encoders, and two latent DDMs are trained in these latent spaces. Synthesis in LION proceeds by drawing novel latent samples from the hierarchical latent DDMs and decoding back to the original point cloud space. Importantly, we also demonstrate how to augment LION with modern surface reconstruction methods [68] to synthesize smooth shapes as desired by artists. LION has multiple advantages:

**Expressivity:** By mapping point clouds into regularized latent spaces, the DDMs in latent space are effectively tasked with learning a smoothed distribution. This is easier than training on potentially complex point clouds directly [58], thereby improving expressivity. However, point clouds are, in principle, an ideal representation for DDMs. Because of that, we use *latent points*, this is, we keep a point cloud structure for our main latent representation. Augmenting the model with an additional global shape latent variable in a hierarchical manner further boosts expressivity. We validate LION on several popular ShapeNet benchmarks and achieve state-of-the-art synthesis performance.

**Varying Output Types:** Extending LION with *Shape As Points* (SAP) [68] geometry reconstruction allows us to also output smooth meshes. Fine-tuning SAP on data generated by LION's autoencoder reduces synthesis noise and enables us to generate high-quality geometry. LION combines (latent) point cloud-based modeling, ideal for DDMs, with surface reconstruction, desired by artists.

**Flexibility:** Since LION is set up as a VAE, it can be easily adapted for different tasks without re-training the latent DDMs: We can efficiently fine-tune LION's encoders on voxelized or noisy inputs, which a user can provide for guidance. This enables multimodal voxel-guided synthesis and shape de-noising. We also leverage LION's latent spaces for shape interpolation and autoencoding. Optionally training the DDMs conditioned on CLIP embeddings enables image- and text-driven 3D generation.

In summary, we make the following contributions: **(i)** We introduce LION, a novel generative model for 3D shape synthesis, which operates on point clouds and is built on a hierarchical VAE framework with two latent DDMs. **(ii)** We validate LION's high synthesis quality by reaching state-of-the-art performance on widely used ShapeNet benchmarks. **(iii)** We achieve high-quality and diverse 3D shape synthesis with LION even when trained jointly over many classes without conditioning. **(iv)** We propose to combine LION with SAP-based surface reconstruction. **(v)** We demonstrate the flexibility of our framework by adapting it to relevant tasks such as multimodal voxel-guided synthesis.

## 2 Background

Traditionally, DDMs were introduced in a discrete-step fashion: Given samples $\mathbf{x}_0 \sim q(\mathbf{x}_0)$ from a data distribution, DDMs use a Markovian fixed forward diffusion process defined as [65, 53]

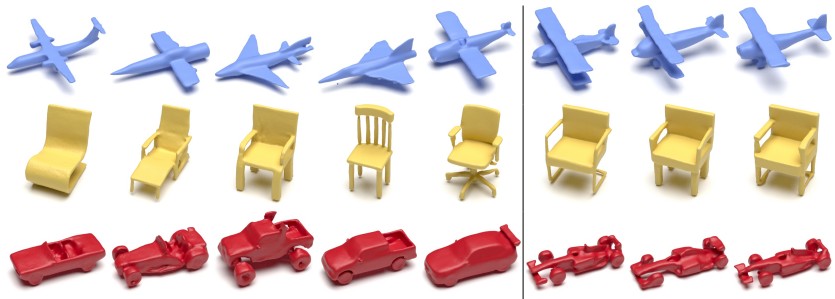

Figure 2: Generated meshes with LION. *Right*: Synthesizing different details by *diffuse-denoise* (see Sec. 3.1) in latent space, while preserving overall shapes.

$$q(\mathbf{x}_{1:T}|\mathbf{x}_0) := \prod_{t=1}^{T} q(\mathbf{x}_t|\mathbf{x}_{t-1}), \qquad q(\mathbf{x}_t|\mathbf{x}_{t-1}) := \mathcal{N}(\mathbf{x}_t; \sqrt{1-\beta_t}\mathbf{x}_{t-1}, \beta_t \mathbf{I}), \tag{1}$$

where $T$ denotes the number of steps and $q(\mathbf{x}_t|\mathbf{x}_{t-1})$ is a Gaussian transition kernel, which gradually adds noise to the input with a variance schedule $\beta_1, ..., \beta_T$. The $\beta_t$ are chosen such that the chain approximately converges to a standard Gaussian distribution after $T$ steps, $q(\mathbf{x}_T) \approx \mathcal{N}(\mathbf{x}_T; \mathbf{0}, \mathbf{I})$. DDMs learn a parametrized reverse process (model parameters $\boldsymbol{\theta}$) that inverts the forward diffusion:

$$p_{\boldsymbol{\theta}}(\mathbf{x}_{0:T}) := p(\mathbf{x}_T) \prod_{t=1}^{T} p_{\boldsymbol{\theta}}(\mathbf{x}_{t-1}|\mathbf{x}_t), \qquad p_{\boldsymbol{\theta}}(\mathbf{x}_{t-1}|\mathbf{x}_t) := \mathcal{N}(\mathbf{x}_{t-1}; \mu_{\boldsymbol{\theta}}(\mathbf{x}_t, t), \rho_t^2 \mathbf{I}). \tag{2}$$

This generative reverse process is also Markovian with Gaussian transition kernels, which use fixed variances $\rho_t^2$. DDMs can be interpreted as latent variable models, where $\mathbf{x}_1, ..., \mathbf{x}_T$ are latents, and the forward process $q(\mathbf{x}_{1:T}|\mathbf{x}_0)$ acts as a fixed approximate posterior, to which the generative $p_{\boldsymbol{\theta}}(\mathbf{x}_{0:T})$ is fit. DDMs are trained by minimizing the variational upper bound on the negative log-likelihood of the data $\mathbf{x}_0$ under $p_{\boldsymbol{\theta}}(\mathbf{x}_{0:T})$. Up to irrelevant constant terms, this objective can be expressed as [53]

$$\min_{\boldsymbol{\theta}} \mathbb{E}_{t \sim U\{1,T\}, \mathbf{x}_0 \sim p(\mathbf{x}_0), \boldsymbol{\epsilon} \sim \mathcal{N}(\mathbf{0}, \mathbf{I})} \left[ w(t) ||\boldsymbol{\epsilon} - \boldsymbol{\epsilon}_{\boldsymbol{\theta}}(\alpha_t \mathbf{x}_0 + \sigma_t \boldsymbol{\epsilon}, t)||_2^2 \right], \quad w(t) = \frac{\beta_t^2}{2\rho_t^2(1-\beta_t)(1-\alpha_t^2)}, \tag{3}$$

where $\alpha_t = \sqrt{\prod_{s=1}^{t}(1-\beta_s)}$ and $\sigma_t = \sqrt{1-\alpha_t^2}$ are the parameters of the tractable diffused distribution after $t$ steps $q(\mathbf{x}_t|\mathbf{x}_0) = \mathcal{N}(\mathbf{x}_t; \alpha_t \mathbf{x}_0, \sigma_t^2 \mathbf{I})$. Furthermore, Eq. (3) employs the widely used parametrization $\mu_{\boldsymbol{\theta}}(\mathbf{x}_t, t) := \frac{1}{\sqrt{1-\beta_t}} \left( \mathbf{x}_t - \frac{\beta_t}{\sqrt{1-\alpha_t^2}} \boldsymbol{\epsilon}_{\boldsymbol{\theta}}(\mathbf{x}_t, t) \right)$. It is common practice to set $w(t) = 1$, instead of the one in Eq. (3), which often promotes perceptual quality of the generated output. In the objective of Eq. (3), the model $\boldsymbol{\epsilon}_{\boldsymbol{\theta}}$ is, for all possible steps $t$ along the diffusion process, effectively trained to predict the noise vector $\boldsymbol{\epsilon}$ that is necessary to denoise an observed diffused sample $\mathbf{x}_t$. After training, the DDM can be sampled with ancestral sampling in an iterative fashion:

$$\mathbf{x}_{t-1} = \frac{1}{\sqrt{1-\beta_t}}(\mathbf{x}_t - \frac{\beta_t}{\sqrt{1-\alpha_t^2}}\boldsymbol{\epsilon}_{\boldsymbol{\theta}}(\mathbf{x}_t, t)) + \rho_t \boldsymbol{\eta}, \tag{4}$$

where $\boldsymbol{\eta} \sim \mathcal{N}(\boldsymbol{\eta}; \mathbf{0}, \mathbf{I})$. This sampling chain is initialized from a random sample $\mathbf{x}_T \sim \mathcal{N}(\mathbf{x}_T; \mathbf{0}, \mathbf{I})$. Furthermore, the noise injection in Eq. 4 is usually omitted in the last sampling step.

DDMs can also be expressed with a continuous-time framework [67, 69]. In this formulation, the diffusion and reverse generative processes are described by differential equations. This approach allows for deterministic sampling and encoding schemes based on ordinary differential equations (ODEs). We make use of this framework in Sec. 3.1 and we review this approach in more detail in App. B.

## 3 Hierarchical Latent Point Diffusion Models

We first formally introduce LION, then discuss various applications and extensions in Sec. 3.1, and finally recapitulate its unique advantages in Sec. 3.2. See Fig. 1 for a visualization of LION.

We are modeling point clouds $\mathbf{x} \in \mathbb{R}^{3 \times N}$, consisting of $N$ points with $xyz$-coordinates in $\mathbb{R}^3$. LION is set up as a hierarchical VAE with DDMs in latent space. It uses a vector-valued global shape latent $\mathbf{z}_0 \in \mathbb{R}^{D_\mathbf{z}}$ and a point cloud-structured latent $\mathbf{h}_0 \in \mathbb{R}^{(3+D_\mathbf{h}) \times N}$. Specifically, $\mathbf{h}_0$ is a *latent point cloud* consisting of $N$ points with $xyz$-coordinates in $\mathbb{R}^3$. In addition, each latent point can carry additional $D_\mathbf{h}$ latent features. Training of LION is then performed in two stages—first, we train it as a regular VAE with standard Gaussian priors; then, we train the latent DDMs on the latent encodings.

**First Stage Training.** Initially, LION is trained by maximizing a modified variational lower bound on the data log-likelihood (ELBO) with respect to the encoder and decoder parameters $\phi$ and $\boldsymbol{\xi}$ [70, 71]:

$$\begin{aligned} \mathcal{L}_{\text{ELBO}}(\phi, \boldsymbol{\xi}) = \mathbb{E}_{p(\mathbf{x}), q_\phi(\mathbf{z}_0|\mathbf{x}), q_\phi(\mathbf{h}_0|\mathbf{x}, \mathbf{z}_0)} \big[ &\log p_{\boldsymbol{\xi}}(\mathbf{x}|\mathbf{h}_0, \mathbf{z}_0) \\ &- \lambda_\mathbf{z} D_{\text{KL}}\left(q_\phi(\mathbf{z}_0|\mathbf{x})|p(\mathbf{z}_0)\right) - \lambda_\mathbf{h} D_{\text{KL}}\left(q_\phi(\mathbf{h}_0|\mathbf{x}, \mathbf{z}_0)|p(\mathbf{h}_0)\right) \big]. \end{aligned} \tag{5}$$

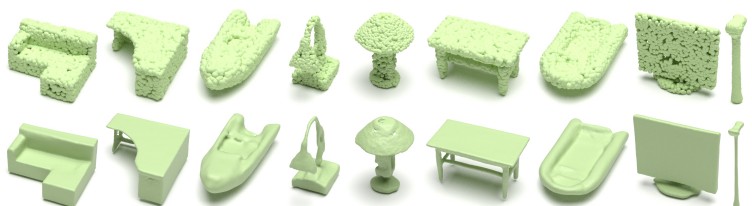

Figure 3: Generated shapes (*top*: point clouds, *bottom*: corresponding meshes) from LION trained jointly over 13 classes of ShapeNet-vol without conditioning (Sec. 5.2).

Here, the global shape latent $\mathbf{z}_0$ is sampled from the posterior distribution $q_\phi(\mathbf{z}_0|\mathbf{x})$, which is parametrized by factorial Gaussians, whose means and variances are predicted via an encoder network. The point cloud latent $\mathbf{h}_0$ is sampled from a similarly parametrized posterior $q_\phi(\mathbf{h}_0|\mathbf{x}, \mathbf{z}_0)$, while also conditioning on $\mathbf{z}_0$ ($\phi$ denotes the parameters of both encoders). Furthermore, $p_\xi(\mathbf{x}|\mathbf{h}_0, \mathbf{z}_0)$ denotes the decoder, parametrized as a factorial Laplace distribution with predicted means and fixed unit scale parameter (corresponding to an $L_1$ reconstruction loss). $\lambda_\mathbf{z}$ and $\lambda_\mathbf{h}$ are hyperparameters balancing reconstruction accuracy and Kullback-Leibler regularization (note that only for $\lambda_\mathbf{z} = \lambda_\mathbf{h} = 1$ we are optimizing a rigorous ELBO). The priors $p(\mathbf{z}_0)$ and $p(\mathbf{h}_0)$ are $\mathcal{N}(\mathbf{0}, \mathbf{I})$. Also see Fig. 1 again.

**Second Stage Training.** In principle, we could use the VAE's priors to sample encodings and generate new shapes. However, the simple Gaussian priors will not accurately match the encoding distribution from the training data and therefore produce poor samples (*prior hole problem* [58, 72–79]). This motivates training highly expressive latent DDMs. In particular, in the second stage we freeze the VAE's encoder and decoder networks and train two latent DDMs on the encodings $\mathbf{z}_0$ and $\mathbf{h}_0$ sampled from $q_\phi(\mathbf{z}_0|\mathbf{x})$ and $q_\phi(\mathbf{h}_0|\mathbf{x}, \mathbf{z}_0)$, minimizing score matching (SM) objectives similar to Eq. (2):

$$\mathcal{L}_{\text{SM}^\mathbf{z}}(\boldsymbol{\theta}) = \mathbb{E}_{t\sim U\{1,T\}, p(\mathbf{x}), q_\phi(\mathbf{z}_0|\mathbf{x}), \boldsymbol{\epsilon}\sim\mathcal{N}(\mathbf{0},\mathbf{I})}||\boldsymbol{\epsilon} - \boldsymbol{\epsilon}_{\boldsymbol{\theta}}(\mathbf{z}_t, t)||_2^2, \tag{6}$$

$$\mathcal{L}_{\text{SM}^\mathbf{h}}(\boldsymbol{\psi}) = \mathbb{E}_{t\sim U\{1,T\}, p(\mathbf{x}), q_\phi(\mathbf{z}_0|\mathbf{x}), q_\phi(\mathbf{h}_0|\mathbf{x},\mathbf{z}_0), \boldsymbol{\epsilon}\sim\mathcal{N}(\mathbf{0},\mathbf{I})}||\boldsymbol{\epsilon} - \boldsymbol{\epsilon}_{\boldsymbol{\psi}}(\mathbf{h}_t, \mathbf{z}_0, t)||_2^2, \tag{7}$$

where $\mathbf{z}_t = \alpha_t \mathbf{z}_0 + \sigma_t \boldsymbol{\epsilon}$ and $\mathbf{h}_t = \alpha_t \mathbf{h}_0 + \sigma_t \boldsymbol{\epsilon}$ are the diffused latent encodings. Furthermore, $\boldsymbol{\theta}$ denotes the parameters of the global shape latent DDM $\boldsymbol{\epsilon}_{\boldsymbol{\theta}}(\mathbf{z}_t, t)$, and $\boldsymbol{\psi}$ refers to the parameters of the conditional DDM $\boldsymbol{\epsilon}_{\boldsymbol{\psi}}(\mathbf{h}_t, \mathbf{z}_0, t)$ trained over the latent point cloud (note the conditioning on $\mathbf{z}_0$).

**Generation.** With the latent DDMs, we can formally define a hierarchical generative model $p_{\xi,\psi,\theta}(\mathbf{x}, \mathbf{h}_0, \mathbf{z}_0) = p_\xi(\mathbf{x}|\mathbf{h}_0, \mathbf{z}_0)p_\psi(\mathbf{h}_0|\mathbf{z}_0)p_\theta(\mathbf{z}_0)$, where $p_\theta(\mathbf{z}_0)$ denotes the distribution of the global shape latent DDM, $p_\psi(\mathbf{h}_0|\mathbf{z}_0)$ refers to the DDM modeling the point cloud-structured latents, and $p_\xi(\mathbf{x}|\mathbf{h}_0, \mathbf{z}_0)$ is LION's decoder. We can hierarchically sample the latent DDMs following Eq. (4) and then translate the latent points back to the original point cloud space with the decoder.

**Network Architectures and DDM Parametrization.** Let us briefly summarize key implementation choices. The encoder networks, as well as the decoder and the latent point DDM, operating on point clouds $\mathbf{x}$, are all implemented based on Point-Voxel CNNs (PVCNNs) [80], following Zhou et al. [46]. PVCNNs efficiently combine the point-based processing of PointNets [81, 82] with the strong spatial inductive bias of convolutions. The DDM modeling the global shape latent uses a ResNet [83] structure with fully-connected layers (implemented as $1\times1$-convolutions). All conditionings on the global shape latent are implemented via adaptive Group Normalization [84] in the PVCNN layers. Furthermore, following Vahdat et al. [58] we use a *mixed score parametrization* in both latent DDMs. This means that the score models are parametrized to predict a residual correction to an analytic standard Gaussian score. This is beneficial since the latent encodings are regularized towards a standard Gaussian distribution during the first training stage (see App. D for all details).

### 3.1 Applications and Extensions

Here, we discuss how LION can be used and extended for different relevant applications.

**Multimodal Generation.** We can synthesize different variations of a given shape, enabling multi-modal generation in a controlled manner: Given a shape, i.e., its point cloud $\mathbf{x}$, we encode it into latent space. Then, we diffuse its encodings $\mathbf{z}_0$ and $\mathbf{h}_0$ for a small number of steps $\tau < T$ towards intermediate $\mathbf{z}_\tau$ and $\mathbf{h}_\tau$ along the diffusion process such that only local details are destroyed. Running the reverse generation process from this intermediate $\tau$, starting at $\mathbf{z}_\tau$ and $\mathbf{h}_\tau$, leads to variations of the original shape with different details (see, for instance, Fig. 2). We refer to this procedure as *diffuse-denoise* (details in App. C.1). Similar techniques have been used for image editing [85].

**Encoder Fine-tuning for Voxel-Conditioned Synthesis and Denoising.** In practice, an artist using a 3D generative model may have a rough idea of the desired shape. For instance, they may be able to quickly construct a coarse voxelized shape, to which the generative model then adds realistic details.

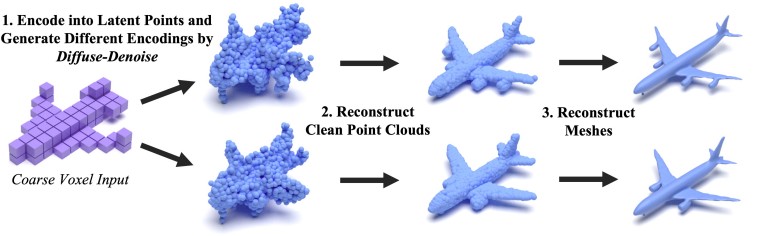

**1. Encode into Latent Points and Generate Different Encodings by** *Diffuse-Denoise*

*Coarse Voxel Input*

**2. Reconstruct Clean Point Clouds**

**3. Reconstruct Meshes**

*Latent Points*  *Generated Point Clouds*  *Generated Meshes*

Figure 4: Voxel-guided synthesis with *LION*. We run *diffuse-denoise* in latent space (see Sec. 3.1) to generate diverse plausible clean shapes.

In LION, we can support such applications: using a similar ELBO as in Eq. (5), but with a frozen decoder, we can fine-tune LION's encoder networks to take voxelized shapes as input (we simply place points at the voxelized shape's surface) and map them to the corresponding latent encodings $\mathbf{z}_0$ and $\mathbf{h}_0$ that reconstruct the original non-voxelized point cloud. Now, a user can utilize the fine-tuned encoders to encode voxelized shapes and generate plausible detailed shapes. Importantly, this can be naturally combined with the *diffuse-denoise* procedure to clean up imperfect encodings and to generate different possible detailed shapes (see Fig. 4).

Furthermore, this approach is general. Instead of voxel-conditioned synthesis, we can also fine-tune the encoder networks on noisy shapes to perform multimodal shape denoising, also potentially combined with *diffuse-denoise*. LION supports these applications easily without re-training the latent DDMs due to its VAE framework with additional encoders and decoders, in contrast to previous works that train DDMs on point clouds directly [46, 47]. See App. C.2 for technical details.

**Shape Interpolation.** LION also enables shape interpolation: We can encode different point clouds into LION's hierarchical latent space and use the *probability flow ODE* (see App. B) to further encode into the latent DDMs' Gaussian priors, where we can safely perform spherical interpolation and expect valid shapes along the interpolation path. We can use the intermediate encodings to generate the interpolated shapes (see Fig. 7; details in App. C.3).

**Surface Reconstruction.** While point clouds are an ideal 3D representation for DDMs, artists may prefer meshed outputs. Hence, we propose to combine LION with modern geometry reconstruction methods (see Figs. 2, 4 and 5). We use *Shape As Points* (SAP) [68], which is based on differentiable Poisson surface reconstruction and can be trained to extract smooth meshes from noisy point clouds. Moreover, we fine-tune SAP on training data generated by LION's autoencoder to better adjust SAP to the noise distribution in point clouds generated by LION. Specifically, we take clean shapes, encode them into latent space, run a few steps of *diffuse-denoise* that only slightly modify some details, and decode back. The *diffuse-denoise* in latent space results in noise in the generated point clouds similar to what is observed during unconditional synthesis (details in App. C.4).

Figure 5: Reconstructing a mesh from LION's generated points.

## 3.2 LION's Advantages

We now recapitulate LION's unique advantages. LION's structure as a hierarchical VAE with latent DDMs is inspired by latent DDMs on images [57, 58, 77]. This framework has key benefits:

**(i) Expressivity:** First training a VAE that regularizes the latent encodings to approximately fall under standard Gaussian distributions, which are also the DDMs' equilibrium distributions towards which the diffusion processes converge, results in an easier modeling task for the DDMs: They have to model only the remaining mismatch between the actual encoding distributions and their own Gaussian priors [58]. This translates into improved expressivity, which is further enhanced by the additional decoder network. However, point clouds are, in principle, an ideal representation for the DDM framework, because they can be diffused and denoised easily and powerful point cloud processing architectures exist. Therefore, LION uses *point cloud latents* that combine the advantages of both latent DDMs and 3D point clouds. Our point cloud latents can be interpreted as smoothed versions of the original point clouds that are easier to model (see Fig. 1). Moreover, the hierarchical VAE setup with an additional global shape latent increases LION's expressivity even further and results in natural disentanglement between overall shape and local details captured by the shape latents and latent points (Sec. 5.2).

**(ii) Flexibility:** Another advantage of LION's VAE framework is that its encoders can be fine-tuned for various relevant tasks, as discussed previously, and it also enables easy shape interpolation. Other 3D point cloud DDMs operating on point clouds directly [47, 46] do not offer simultaneously as much flexibility and expressivity out-of-the-box (see quantitative comparisons in Secs. 5.1 and 5.4).

**(iii) Mesh Reconstruction:** As discussed, while point clouds are ideal for DDMs, artists likely prefer meshed outputs. As explained above, we propose to use LION together with modern surface recon-

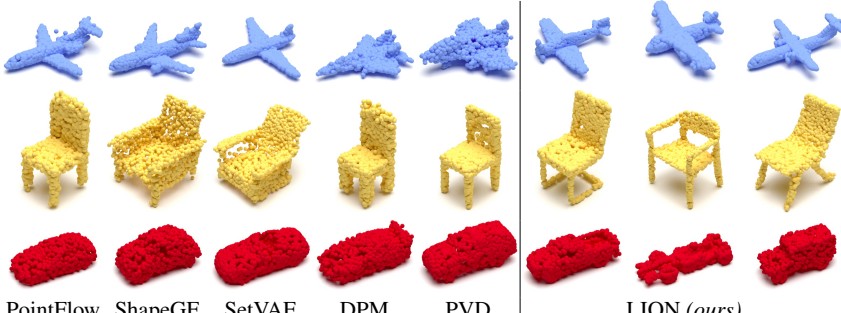

Figure 6: Unconditional shape generation with 2,048 points for *airplane*, *car* and *chair* classes (class-specific models trained on PointFlow's ShapeNet data with global normalization).

PointFlow  ShapeGF  SetVAE  DPM  PVD  |  LION *(ours)*

struction techniques [68], again combining the best of both worlds—a point cloud-based VAE backbone ideal for DDMs, and smooth geometry reconstruction methods operating on the synthesized point clouds to generate practically useful smooth surfaces, which can be easily transformed into meshes.

## 4   Related Work

We are building on DDMs [53, 65–67], which have been used most prominently for image [53–63] and speech synthesis [86–91]. We train DDMs in latent space, an idea that has been explored for image [57, 58, 77] and music [92] generation, too. However, these works did not train separate conditional DDMs. Hierarchical DDM training has been used for generative image upsampling [54], text-to-image generation [63, 64], and semantic image modeling [60]. Most relevant among these works is Preechakul et al. [60], which extracts a high-level semantic representation of an image with an auxiliary encoder and then trains a DDM that adds details directly in image space. We are the first to explore related concepts for 3D shape synthesis and we also train both DDMs in latent space. Furthermore, DDMs and VAEs have also been combined in such a way that the DDM improves the output of the VAE [93].

Most related to LION are "Point-Voxel Diffusion" (PVD) [46] and "Diffusion Probabilistic Models for 3D Point Cloud Generation" (DPM) [47]. PVD trains a DDM directly on point clouds, and our decision to use PVCNNs is inspired by this work. DPM, like LION, uses a shape latent variable, but models its distribution with Normalizing Flows [94, 95], and then trains a weaker point-wise conditional DDM directly on the point cloud data (this allows DPM to learn useful representations in its latent variable, but sacrifices generation quality). As we show below, neither PVD nor DPM easily enables applications such as multimodal voxel-conditioned synthesis and denoising. Furthermore, LION achieves significantly stronger generation performance. Finally, neither PVD nor DPM reconstructs meshes from the generated point clouds. Point cloud and 3D shape generation have also been explored with other generative models: PointFlow [31], DPF-Net [33] and SoftFlow [32] rely on Normalizing Flows [94–97]. SetVAE [29] treats point cloud synthesis as set generation and uses VAEs. ShapeGF [45] learns distributions over gradient fields that model shape surfaces. Both IM-GAN [7], which models shapes as neural fields, and l-GAN [2] train GANs over latent variables that encode the shapes, similar to other works [3], while r-GAN [2] generates point clouds directly. PDGN [52] proposes progressive deconvolutional networks within a point cloud GAN. SP-GAN [19] uses a spherical point cloud prior. Other progressive [22, 37] and graph-based architectures [4, 6] have been used, too. Also generative cellular automata (GCAs) can be employed for voxel-based 3D shape generation [43]. In orthogonal work, point cloud DDMs have been used for generative shape completion [46, 98].

Recently, image-driven [8–16, 44] training of 3D generative models as well as text-driven 3D generation [34, 49–51] have received much attention. These are complementary directions to ours; in fact, augmenting LION with additional image-based training or including text-guidance are promising future directions. Finally, we are relying on SAP [68] for mesh generation. Strong alternative approaches for reconstructing smooth surfaces from point clouds exist [99–103].

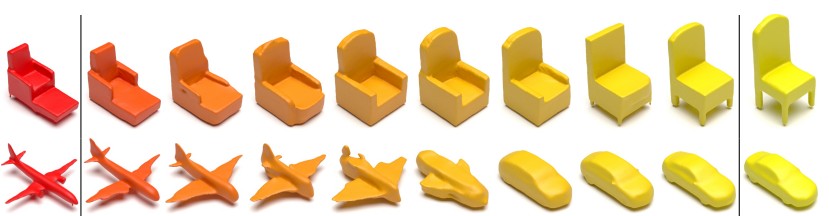

Figure 7: Interpolating different shapes by interpolating their encodings in the standard Gaussian priors of LION's latent DDMs (details in App. C.3).

Table 1: Generation metrics (1-NNA↓) on *airplane*, *chair*, *car* categories from ShapeNet dataset from PointFlow [31]. Training and test data normalized globally into [-1, 1].

|  | Airplane | | Chair | | Car | |
|---|---|---|---|---|---|---|
|  | CD | EMD | CD | EMD | CD | EMD |
| r-GAN [2] | 98.40 | 96.79 | 83.69 | 99.70 | 94.46 | 99.01 |
| l-GAN (CD) [2] | 87.30 | 93.95 | 68.58 | 83.84 | 66.49 | 88.78 |
| l-GAN (EMD) [2] | 89.49 | 76.91 | 71.90 | 64.65 | 71.16 | 66.19 |
| PointFlow [31] | 75.68 | 70.74 | 62.84 | 60.57 | 58.10 | 56.25 |
| SoftFlow [32] | 76.05 | 65.80 | 59.21 | 60.05 | 64.77 | 60.09 |
| SetVAE [29] | 76.54 | 67.65 | 58.84 | 60.57 | 59.94 | 59.94 |
| DPF-Net [33] | 75.18 | 65.55 | 62.00 | 58.53 | 62.35 | 54.48 |
| DPM [47] | 76.42 | 86.91 | 60.05 | 74.77 | 68.89 | 79.97 |
| PVD [46] | 73.82 | 64.81 | 56.26 | 53.32 | 54.55 | 53.83 |
| LION *(ours)* | **67.41** | **61.23** | **53.70** | **52.34** | **53.41** | **51.14** |

Table 2: Generation results (1-NNA↓) on ShapeNet dataset from PointFlow [31]. All data normalized individually into [-1, 1].

|  | Airplane | | Chair | | Car | |
|---|---|---|---|---|---|---|
|  | CD | EMD | CD | EMD | CD | EMD |
| TreeGAN [6] | 97.53 | 99.88 | 88.37 | 96.37 | 89.77 | 94.89 |
| ShapeGF [45] | 81.23 | 80.86 | 58.01 | 61.25 | 61.79 | 57.24 |
| SP-GAN [19] | 94.69 | 93.95 | 72.58 | 83.69 | 87.36 | 85.94 |
| PDGN [52] | 94.94 | 91.73 | 71.83 | 79.00 | 89.35 | 87.22 |
| GCA [43] | 88.15 | 85.93 | 64.27 | 64.50 | 70.45 | 64.20 |
| LION *(ours)* | **76.30** | **67.04** | **56.50** | **53.85** | **59.52** | **49.29** |

Table 3: Results (1-NNA↓) on ShapeNet-vol.

|  | Airplane | | Chair | | Car | |
|---|---|---|---|---|---|---|
|  | CD | EMD | CD | EMD | CD | EMD |
| IM-GAN [7] | 79.70 | 77.85 | 57.09 | 58.20 | 88.92 | 84.58 |
| DPM [47] | 83.04 | 96.04 | 61.96 | 74.96 | 77.30 | 87.12 |
| PVD [46] | 66.46 | 56.06 | 61.89 | 57.90 | 64.49 | 55.74 |
| LION *(ours)* | **53.47** | **53.84** | **52.07** | **48.67** | **54.81** | **50.53** |

# 5 Experiments

We provide an overview of our most interesting experimental results in the main paper. All experiment details and extensive additional experiments can be found in App. E and App. F, respectively.

## 5.1 Single-Class 3D Shape Generation

**Datasets.** To compare LION against existing methods, we use ShapeNet [104], the most widely used dataset to benchmark 3D shape generative models. Following previous works [31, 46, 47], we train on three categories: *airplane*, *chair*, *car*. Also like previous methods, we primarily rely on PointFlow's [31] dataset splits and preprocssing. It normalizes the data globally across the whole dataset. However, some baselines require per-shape normalization [19, 43, 45, 52]; hence, we also train on such data. Furthermore, training SAP requires signed distance fields (SDFs) for volumetric supervision, which the PointFlow data does not offer. Hence, for simplicity we follow Peng et al. [68, 101] and also use their data splits and preprocessing, which includes SDFs.We train LION, DPM, PVD, and IM-GAN (which synthesizes shapes as SDFs) also on this dataset version (denoted as *ShapeNet-vol* here). This data is also per-shape normalized. Dataset details in App. E.1.

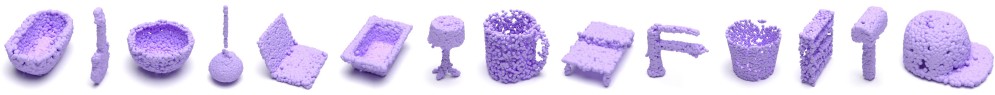

Figure 8: Samples from our unconditional 13-class model: In each column, we use the same global shape latent $z_0$.

**Evaluation.** Model evaluation follows previous works [31, 46]. Various metrics to evaluate point cloud generative models exist, with different advantages and disadvantages, discussed in detail by Yang et al. [31]. Following recent works [31, 46], we use *1-NNA* (with both Chamfer distance (CD) and earth mover distance (EMD)) as our main metric. It quantifies the distributional similarity between generated shapes and validation set and measures both quality and diversity [31]. For fair comparisons, all metrics are computed on point clouds, not meshed outputs (App. E.2 discusses different metrics; further results on coverage (COV) and minimum matching distance (MMD) in App. F.2).

Table 4: Generation results (1-NNA↓) of LION trained jointly on 13 classes of ShapeNet-vol.

| Model | CD | EMD |
|---|---|---|
| TreeGAN [6] | 96.80 | 96.60 |
| PointFlow [31] | 63.25 | 66.05 |
| ShapeGF [45] | 55.65 | 59.00 |
| SetVAE [29] | 79.25 | 95.25 |
| PDGN [52] | 71.05 | 86.00 |
| DPF-Net [33] | 67.10 | 64.75 |
| DPM [47] | 62.30 | 86.50 |
| PVD [46] | 58.65 | 57.85 |
| LION *(ours)* | **51.85** | **48.95** |

**Results.** Samples from LION are shown in Fig. 6 and quantitative results in Tabs. 1-3 (see Sec. 4 for details about baselines—to reduce the number of baselines to train, we are focusing on the most recent and competitive ones). LION outperforms all baselines and achieves state-of-the-art performance on all classes and dataset versions. Importantly, we outperform both PVD and DPM, which also leverage DDMs, by large margins. Our samples are diverse and appear visually pleasing.

Figure 9: Generated point clouds from LION trained jointly over 55 classes of ShapeNet-vol (no conditioning).

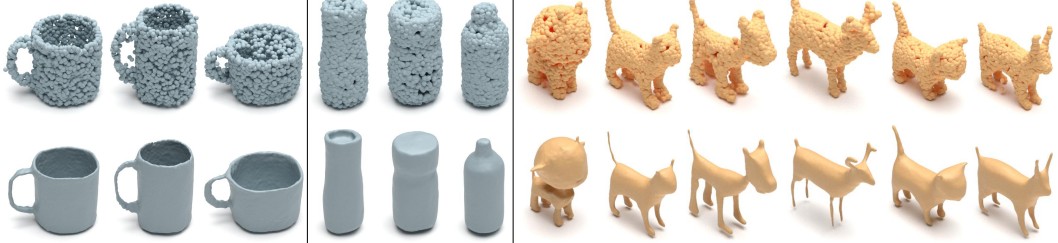

Figure 10: Samples from LION trained on ShapeNet's Mug and Bottle classes, and on Turbosquid animals.

**Mesh Reconstruction.** As explained in Sec. 3.1, we combine LION with mesh reconstruction, to directly synthesize practically useful meshes. We show generated meshes in Fig. 2, which look smooth and of high quality. In Fig. 2, we also visually demonstrate how we can vary the local details of synthesized shapes while preserving the overall shape with our *diffuse-denoise* technique (Sec. 3.1). Details about the number of diffusion steps for all *diffuse-denoise* experiments are in App. E.

**Shape Interpolation.** As discussed in Sec. 3.1, LION also enables shape interpolation, potentially useful for shape editing applications. We show this in Fig. 7, combined with mesh reconstruction. The generated shapes are clean and semantically plausible along the entire interpolation path. In App. F.12.1, we also show interpolations from PVD [46] and DPM [47] for comparison.

## 5.2 Many-class Unconditional 3D Shape Generation

**13-Class LION Model.** We train a LION model *jointly without any class conditioning* on 13 different categories (*airplane, chair, car, lamp, table, sofa, cabinet, bench, telephone, loudspeaker, display, watercraft, rifle*) from ShapeNet (ShapeNet-vol version). Training a single model without conditioning over such diverse shapes is challenging, as the data distribution is highly complex and multimodal. We show LION's generated samples in Fig. 3, including meshes: LION synthesizes high-quality and diverse plausible shapes even when trained on such complex data. We report the model's quantitative generation performance in Tab. 4, and we also trained various strong baseline methods under the same setting for comparison. We find that LION significantly outperforms all baselines by a large margin. We further observe that the hierarchical VAE architecture of LION becomes crucial: The shape latent variable $z_0$ captures global shape, while the latent points $h_0$ model details. This can be seen in Fig. 8: we show samples when fixing the global shape latent $z_0$ and only sample $h_0$ (details in App. F.3).

**55-Class LION Model.** Encouraged by these results, we also trained a LION model again *jointly without any class conditioning* on all 55 different categories from ShapeNet. Note that we did on purpose not use class-conditioning in these experiments to create a difficult 3D generation task and thereby explore LION's scalability to highly complex and multimodal datasets. We show generated point cloud samples in Fig. 9 (we did not train an SAP model on the 55 classes data): LION synthesizes high-quality and diverse shapes. It can even generate samples from the *cap* class, which contributes with only 39 training data samples, indicating that LION has an excellent mode coverage that even includes the very rare classes. To the best of our knowledge no previous 3D shape generative models have demonstrated satisfactory generation performance for such diverse and multimodal 3D data without relying on conditioning information (details in App. F.4). In conclusion, we observe that LION out-of-the-box easily scales to highly complex multi-category shape generation.

## 5.3 Training LION on Small Datasets

Next, we explore whether LION can also be trained successfully on very small datasets. To this end, we train models on the Mug and Bottle ShapeNet classes. The number of training samples

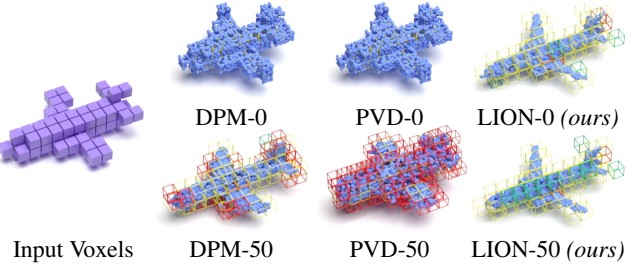

Input Voxels    DPM-50    PVD-50    LION-50 *(ours)*

Figure 11: Voxel-guided synthesis. We show different methods with 0 and 50 steps of *diffuse-denoise*. Voxelizations of generated points are also shown: Yellow boxes indicate generated points correctly fill input voxels, green boxes indicate voxels should be filled but are left empty, red boxes indicate extra voxels.

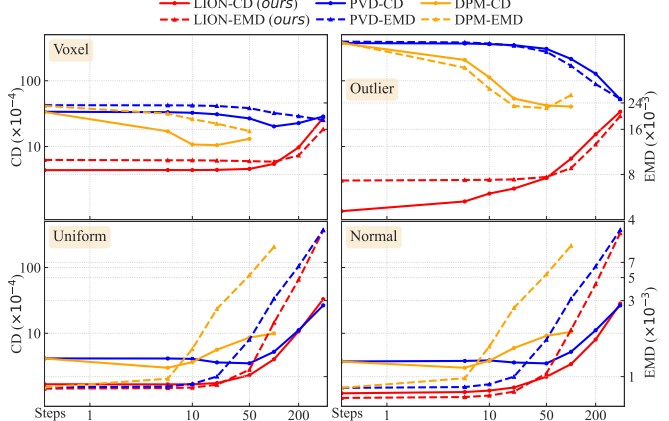

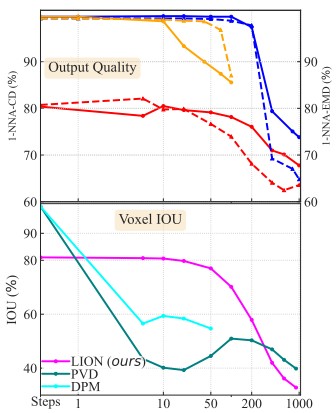

Figure 12: Reconstruction metrics with respect to clean inputs for *airplane* category (lower is better) when guiding synthesis with voxelized or noisy inputs (using uniform, outlier, and normal noise, see App. F.7). *x*-axes denote number of *diffuse-denoise* steps.

Figure 13: Voxel-guided generation. Quality metrics for output points (lower is better) and voxel IOU with respect to input (higher is better). *x*-axes denote *diffuse-denoise* steps.

is 149 and 340, respectively, which is much smaller than the common classes like chair, car and airplane. Furthermore, we also train LION on 553 animal assets from the TurboSquid data repository. Generated shapes from the three models are shown in Fig. 10. LION is able to generate correct mugs and bottles as well as diverse and high-quality animal shapes. We conclude that LION also performs well even when training in the challenging low-data setting (details in Apps. F.5 and F.6).

## 5.4 Voxel-guided Shape Synthesis and Denoising with Fine-tuned Encoders

Next, we test our strategy for multimodal voxel-guided shape synthesis (see Sec. 3.1) using the *airplane* class LION model (experiment details in App. E, more experiments in App. F.7). We first voxelize our training set and fine-tune our encoder networks to produce the correct encodings to decode back the original shapes. When processing voxelized shapes with our point-cloud networks, we sample points on the surface of the voxels. As discussed, we can use different numbers of *diffuse-denoise* steps in latent space to generate various plausible shapes and correct for poor encodings. Instead of voxelizations, we can also consider different noisy inputs (we use *normal*, *uniform*, and *outlier* noise, see App. F.7) and achieve multimodal denoising with the same approach. The same tasks can be attempted with the important DDM-based baselines PVD and DPM, by directly—not in a latent space—diffusing and denoising voxelized (converted to point clouds) or noisy point clouds.

Fig. 12 shows the reconstruction performance of LION, DPM and PVD for different numbers of *diffuse-denoise* steps (we voxelized or noised the validation set to measure this). We see that for almost all inputs—voxelized or different noises—LION performs best. PVD and DPM perform acceptably for normal and uniform noise, which is similar to the noise injected during training of their DDMs, but perform very poorly for outlier noise or voxel inputs, which is the most relevant case to us, because voxels can be easily placed by users. It is LION's unique framework with additional fine-tuned encoders in its VAE and only latent DDMs that makes this possible. Performing more *diffuse-denoise* steps means that more independent, novel shapes are generated. These will be cleaner and of higher quality, but also correspond less to the noisy or voxel inputs used for guidance. In Fig. 13, we show this trade-off for the voxel-guidance experiment (other experiments in App. F.7), where *(top)* we measured the outputs' synthesis quality by calculating 1-NNA with respect to the validation set, and *(bottom)* the average intersection over union (IOU) between the input voxels and the voxelized outputs. We gener-

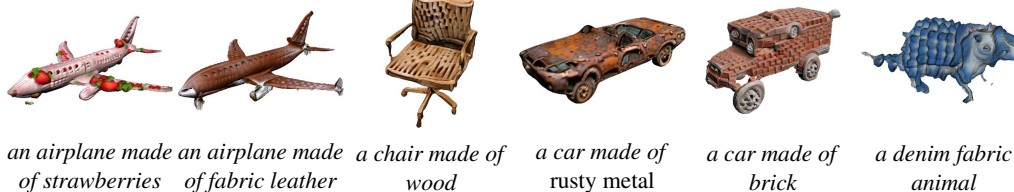

*an airplane made of strawberries*    *an airplane made of fabric leather*    *a chair made of wood*    *a car made of rusty metal*    *a car made of brick*    *a denim fabric animal*

Figure 14: We apply Text2Mesh [49] on meshes generated by LION. In Text2Mesh, textures are generated and meshes refined such that rendered images of the 3D objects are aligned with user-provided text prompts [105].

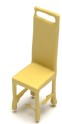 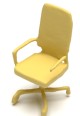 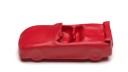 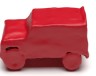

Figure 16: Text-driven shape generation of chairs and cars with LION. Bottom row is the text prompt used as input.

*narrow chair*   *office chair*   *bmw convertible*   *jeep*

ally see a trade-off: More diffuse-denoise steps result in lower 1-NNA (better quality), but also lower IOU. LION strikes the best balance by a large gap: Its additional encoder network directly generates plausible latent encodings from the perturbed inputs that are both high quality and also correspond well to the input. This trade-off is visualized in Fig. 11 for LION, DPM, and PVD, where we show generated point clouds and voxelizations (note that performing no diffuse-denoise at all for PVD and DPM corresponds to simply keeping the input, as these models' DDMs operate directly on point clouds). We see that running 50 *diffuse-denoise* steps to generate diverse outputs for DPM and especially PVD results in a significant violation of the input voxelization. In contrast, LION generates realistic outputs that also obey the driving voxels. Overall, LION wins out both in this task and also in unconditional generation with large gaps over these previous DDM-based point cloud generative models. We conclude that LION does not only offer state-of-the-art 3D shape generation quality, but is also very versatile. Note that guided synthesis can also be combined with mesh reconstruction, as shown in Fig. 4.

## 5.5  Sampling Time

While our main experiments use 1,000-step DDPM-based synthesis, which takes $\approx 27.12$ seconds, we can significantly accelerate generation without significant loss in quality. Using DDIM-based sampling [106], we can generate high quality shapes in under one second (Fig. 15), which would enable real-time interactive applications. More analyses in App. F.9.

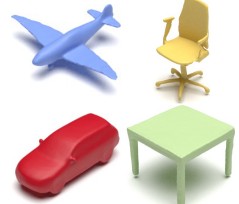

## 5.6  Overview of Additional Experiments in Appendix

**(i)** In App. F.1, we perform various ablation studies. The experiments quantitatively validate LION's architecture choices and the advantage of our

Figure 15: 25-step DDIM [106] samples (0.89 seconds per shape).

hierarchical VAE setup with conditional latent DDMs. **(ii)** In App. F.8, we measure LION's autoencoding performance. **(iii)** To demonstrate the value of directly outputting meshes, in App. F.10 we use Text2Mesh [49] to generate textures based on text prompts for synthesized LION samples (Fig. 14). This would not be possible, if we only generated point clouds. **(iv)** To qualitatively show that LION can be adapted easily to other relevant tasks, in App. F.11 we condition LION on CLIP embeddings of the shapes' rendered images, following CLIP-Forge [34] (Fig. 16). This enables text-driven 3D shape generation and single view 3D reconstruction (Fig. 17). **(v)** We also show many more samples (Apps. F.2-F.6) and shape interpolations (App. F.12) from our models, more examples of voxel-guided and noise-guided synthesis (App. F.7), and we further analyze our 13-class LION model (App. F.3.2).

## 6  Conclusions

We introduced LION, a novel generative model of 3D shapes. LION uses a VAE framework with hierarchical DDMs in latent space and can be combined with SAP for mesh generation. LION achieves state-of-the-art shape generation performance and enables applications such as voxel-conditioned synthesis, multimodal shape denoising, and shape interpolation. LION is currently trained on 3D point clouds only and can not directly generate textured shapes. A promising extension would be to include image-based training by incorporating neural or differentiable rendering [17, 107–111] and to also synthesize textures [16, 112–114]. Furthermore, LION currently focuses on single object generation only. It would be interesting to extend it to full 3D scene synthesis. Moreover, synthesis could be further accelerated by building on works on accelerated sampling from DDMs [61, 62, 67, 106, 115–121].

**Broader Impact.** We believe that LION can potentially improve 3D content creation and assist the workflow of digital artists. We designed LION with such applications in mind and hope that it can grow into a practical tool enhancing artists' creativity. Although we do not see any immediate negative use-cases for LION, it is important that practitioners apply an abundance of caution to mitigate impacts given generative modeling more generally can also be used for malicious purposes, discussed for instance in Vaccari and Chadwick [122], Nguyen et al. [123], Mirsky and Lee [124].

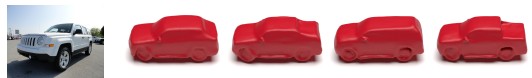

Figure 17: Single view 3D reconstructions of a car from an RGB image. LION can generate multiple plausible outputs using our *diffuse-denoise* technique.

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
