# Contents

## A  Funding Disclosure

This work was fully funded by NVIDIA.

## B  Continuous-Time Diffusion Models and Probability Flow ODE Sampling

Here, we are providing additional background on denoising diffusion models (DDMs). In Sec. 2, we have introduced DDMs in the "discrete-time" setting, where we have a fixed number $T$ of diffusion and denoising steps [53, 65]. However, DDMs can also be expressed in a continuous-time framework, in which the fixed forward diffusion and the generative denoising process in a continuous manner gradually perturb and denoise, respectively [67]. In this formulation, these processes can be described by stochastic differential equations (SDEs). In particular, the fixed forward diffusion process is given by (for the "variance-preserving" SDE [67], which we use. Other diffusion processes are possible [67, 58, 61]):

$$d\mathbf{x}_t = -\frac{1}{2}\beta_t \mathbf{x}_t \, dt + \sqrt{\beta_t} \, d\mathbf{w}_t, \tag{8}$$

where time $t \in [0, 1]$ and $\mathbf{w}_t$ is a standard Wiener process. In the continuous-time formulation, we usually consider times $t \in [0, 1]$, while in the discrete-time setting it is common to consider discrete time values $t \in \{0, ..., T\}$ (with $t = 0$ corresponding to no diffusion at all). This is just a convention and we can easily translate between them as $t_{\text{cont.}} = \frac{t_{\text{disc.}}}{T}$. We always take care of these conversions here when appropriate without explicitly noting this to keep the notation concise. The function $\beta_t$ in Eq. (8) above is a continuous-time generalization of the set of $\beta_t$'s used in the discrete formulation (denoted as variance schedule in Sec. 2). Usually, the $\beta_t$'s in the discrete-time setting are generated by discretizing an underlying continuous function $\beta_t$—in our case $\beta_t$ is simply a linear function of $t$—, which is now used in Eq. (8) above directly.

It can be shown that a corresponding reverse diffusion process exists that effectively inverts the forward diffusion from Eq. (8) [67, 129, 130]:

$$d\mathbf{x}_t = -\frac{1}{2}\beta_t \left[\mathbf{x}_t + 2\nabla_{\mathbf{x}_t} \log q_t(\mathbf{x}_t)\right] dt + \sqrt{\beta_t} \, d\mathbf{w}_t. \tag{9}$$

Here, $q_t(\mathbf{x}_t)$ is the marginal diffused data distribution after time $t$, and $\nabla_{\mathbf{x}_t} \log q_t(\mathbf{x}_t)$ is the *score function*. Hence, if we had access to this score function, we could simulate this reverse SDE in reverse time direction, starting from random noise $\mathbf{x}_1 \sim \mathcal{N}(\mathbf{x}_1; \mathbf{0}, \boldsymbol{I})$, and thereby invert the forward diffusion process and generate novel data. Consequently, the problem reduces to learning a model for the usually intractable score function. This is where the discrete-time and continuous-time frameworks connect: Indeed, the objective in Eq. (3) for training the denoising model also corresponds to *denoising score matching* [131, 66, 53], i.e., it represents an objective to learn a model for the score function. We have

$$\nabla_{\mathbf{x}_t} \log q_t(\mathbf{x}_t) \approx -\frac{\boldsymbol{\epsilon}_{\boldsymbol{\theta}}(\mathbf{x}_t, t)}{\sigma_t}. \tag{10}$$

However, we trained $\boldsymbol{\epsilon}_{\boldsymbol{\theta}}(\mathbf{x}_t, t)$ for $T$ discrete steps only, rather than for continuous times $t$. In principle, the objective in Eq. (3) can be easily adapted to the continuous-time setting by simply sampling continuous time values rather than discrete ones. In practice, $T = 1000$ steps, as used in our models, represents a fine discretization of the full integration interval and the model generalizes well when queried at continuous $t$ "between" steps, due to the smooth cosine-based time step embeddings.

A unique advantage of the continuous-time framework based on differential equations is that it allows us to construct an ordinary differential equation (ODE), which, when simulated with samples from the same random noise distribution $\mathbf{x}_1 \sim \mathcal{N}(\mathbf{x}_1; \mathbf{0}, \boldsymbol{I})$ as inputs (where $t = 1$, with $\mathbf{x}_{t=1}$, denotes the end of the diffusion for continuous $t \in [0, 1]$), leads to the same marginal distributions along the reverse diffusion process and can therefore also be used for synthesis [67]:

$$d\mathbf{x}_t = -\frac{1}{2}\beta_t \left[\mathbf{x}_t + \nabla_{\mathbf{x}_t} \log p_t(\mathbf{x}_t)\right] dt. \tag{11}$$

This is an instance of continuous Normalizing flows [96, 97] and often called *probability flow ODE*. Plugging in our score function estimate, we have

$$d\mathbf{x}_t = -\frac{1}{2}\beta_t \left[\mathbf{x}_t - \frac{\boldsymbol{\epsilon}_{\boldsymbol{\theta}}(\mathbf{x}_t, t)}{\sigma_t}\right] dt, \tag{12}$$

which we refer to as the *generative ODE*. Given a sample from $\mathbf{x}_1 \sim \mathcal{N}(\mathbf{x}_1; \mathbf{0}, \boldsymbol{I})$, the generative process of this generative ODE is fully deterministic. Similarly, we can also use this ODE to encode given data into the DDM's own prior distribution $\mathbf{x}_1 \sim \mathcal{N}(\mathbf{x}_1; \mathbf{0}, \boldsymbol{I})$ by simulating the ODE in the other direction.

These properties allow us to perform interpolation: Due to the deterministic generation process with the generative ODE, smoothly changing an encoding $\mathbf{x}_1$ will result in a similarly smoothly changing generated output $\mathbf{x}_0$. We are using this for our interpolation experiments (see Sec. 3.1 and App. C.3)

## C    Technical Details on LION's Applications and Extensions

In this section, we provide additional methodological details on the different applications and extensions of LION that we discussed in Sec. 3.1 and demonstrated in our experiments.

### C.1    Diffuse-Denoise

Our *diffuse-denoise* technique is essentially a tool to inject diversity into the generation process in a controlled manner and to "clean up" imperfect encodings when working with encoders operating on noisy or voxelized data (see Sec. 3.1 and App. C.2). It is related to similar methods that have been used for image editing [85].

Specifically, assume we are given an input shape $\mathbf{x}$ in the form of a point cloud. We can now use LION's encoder networks to encode it into the latent spaces of LION's autoencoder and obtain the shape latent encoding $\mathbf{z}_0$ and the latent points $\mathbf{h}_0$. Now, we can diffuse those encodings for $\tau < T$ steps (using the Gaussian transition kernel defined in Eq. (1)) to obtain intermediate $\mathbf{z}_\tau$ and $\mathbf{h}_\tau$ along the diffusion process. Next, we can denoise them back to new $\bar{\mathbf{z}}_0$ and $\bar{\mathbf{h}}_0$ using the generative stochastic sampling defined in Eq. (4), starting from the intermediate $\mathbf{z}_\tau$ and $\mathbf{h}_\tau$. Note that we first need to generate the new $\bar{\mathbf{z}}_0$, since denoising $\mathbf{h}_\tau$ is conditioned on $\bar{\mathbf{z}}_0$ according to LION's hierarchical latent DDM setup.

The forward diffusion of DDMs progressively destroys more and more details of the input data. Hence, diffusing LION's latent encodings only for small $\tau$, and then denoising again, results in new $\bar{\mathbf{z}}_0$ and $\bar{\mathbf{h}}_0$ that have only changed slightly compared to the original $\mathbf{z}_0$ and $\mathbf{h}_0$. In other words for small $\tau$, the diffuse-denoised $\bar{\mathbf{z}}_0$ and $\bar{\mathbf{h}}_0$ will be close to the original $\mathbf{z}_0$ and $\mathbf{h}_0$. This observation was also made by Meng et al. [85]. Similarly, we find that when $\bar{\mathbf{z}}_0$ and $\bar{\mathbf{h}}_0$ are sent through LION's decoder network the corresponding point cloud $\bar{\mathbf{x}}$ resembles the input point cloud $\mathbf{x}$ in overall shape well, and only has different details. Diffusing for more steps, i.e., larger $\tau$, corresponds to resampling the shape also more globally (with $\tau = T$ meaning that an entirely new shape is generated), while using smaller $\tau$ implies that the original shape is preserved more faithfully (with $\tau = 0$ meaning that the original shape is preserved entirely). Hence, we can use this technique to inject diversity into any given shape and resample different details in a controlled manner (as shown, for instance, in Fig. 2).

We can use this *diffuse-denoise* approach not only for resampling different details from clean shapes, but also to "clean up" poor encodings. For instance, when LION's encoders operate on very noisy or coarsely voxelized input point clouds (see Sec. 3.1 and App. C.2), the predicted shape encodings may be poor. The encoder networks may roughly recognize the overall shape but not capture any details due to the noise or voxelizations. Hence, we can perform some *diffuse-denoise* to essentially partially discard the poor encodings and regenerate them from the DDMs, which have learnt a model of clean detailed shapes, while preserving the overall shape. This allows us to perform multimodal generation when using voxelized or noisy input point clouds as guidance, because we can sample various different plausible versions using *diffuse-denoise*, while always approximately preserving the overall input shape (see examples in Figs. 4, 29, 30, and 31).

### C.2    Encoder Fine-Tuning for Voxel-Conditioned Synthesis and Denoising

A crucial advantage of LION's underlying VAE framework with latent DDMs is that we can adapt the encoder neural networks for different relevant tasks, as discussed in Sec. 3.1 and demonstrated in our experiments. For instance, a digital artist may have a rough idea about the shape they desire to synthesize and they may be able to quickly put together a coarse voxelization according to whatever

they imagine. Or similarly, a noisy version of a shape may be available and the user may want to guide LION's synthesis accordingly.

To this end, we propose to fine-tune LION's encoder neural networks for such tasks: In particular, we take clean shapes $\mathbf{x}$ from the training data and voxelize them or add noise to them. Specifically, we test three different noise types as well as voxelization (see Figs. 31 and 30): we either perturb the point cloud with uniform noise, Gaussian noise, outlier noise, or we voxelize it. We denote the resulting coarse or noisy shapes as $\tilde{\mathbf{x}}$ here. Given a fully trained LION model, we can fine-tune its encoder networks to ingest the perturbed $\tilde{\mathbf{x}}$, instead of the clean $\mathbf{x}$. For that, we are using the following ELBO-like (maximization) objective:

$$\mathcal{L}_{\text{finetune}}(\boldsymbol{\phi}) = \mathcal{L}_{\text{reconst}}(\boldsymbol{\phi}) - \mathbb{E}_{p(\tilde{\mathbf{x}}), q_\phi(\mathbf{z}_0|\tilde{\mathbf{x}})} \left[ \lambda_{\mathbf{z}} D_{\text{KL}} \left( q_\phi(\mathbf{z}_0|\tilde{\mathbf{x}}) | p(\mathbf{z}_0) \right) + \lambda_{\mathbf{h}} D_{\text{KL}} \left( q_\phi(\mathbf{h}_0|\tilde{\mathbf{x}}, \mathbf{z}_0) | p(\mathbf{h}_0) \right) \right]. \tag{13}$$

When training the encoder to denoise uniform or Gaussian noise added to the point cloud, we use the same reconstruction objective as during original LION training, i.e.,

$$\mathcal{L}_{\text{reconst}}^{L_1}(\boldsymbol{\phi}) = \mathbb{E}_{p(\tilde{\mathbf{x}}), q_\phi(\mathbf{z}_0|\tilde{\mathbf{x}}), q_\phi(\mathbf{h}_0|\tilde{\mathbf{x}}, \mathbf{z}_0)} \log p_\xi(\mathbf{x}|\mathbf{h}_0, \mathbf{z}_0). \tag{14}$$

However, when training with voxelized inputs or outlier noise, there is no good corresponce to define the point-wise reconstruction loss with the Laplace distribution (corresponding to an $L_1$ loss). Therefore, in these cases we instead rely on Chamfer Distance (CD) and Earth Mover Distance (EMD) for the reconstruction term:

$$\mathcal{L}_{\text{reconst}}^{\text{CD/EMD}}(\boldsymbol{\phi}) = \mathbb{E}_{p(\tilde{\mathbf{x}}), q_\phi(\mathbf{z}_0|\tilde{\mathbf{x}}), q_\phi(\mathbf{h}_0|\tilde{\mathbf{x}}, \mathbf{z}_0)} \left[ \mathcal{L}^{\text{CD}}\left(\boldsymbol{\mu}_\xi(\mathbf{h}_0, \mathbf{z}_0), \mathbf{x}\right) + \mathcal{L}^{\text{EMD}}\left(\boldsymbol{\mu}_\xi(\mathbf{h}_0, \mathbf{z}_0), \mathbf{x}\right) \right] \tag{15}$$

Here, $\mathcal{L}^{\text{CD}}$ and $\mathcal{L}^{\text{EMD}}$ denote CD and EMD losses:

$$\mathcal{L}^{\text{CD}}(\mathbf{x}, \mathbf{y}) = \sum_{x \in \mathbf{x}} \min_{y \in \mathbf{y}} ||x - y||_1 + \sum_{y \in \mathbf{y}} \min_{x \in \mathbf{x}} ||x - y||_1, \tag{16}$$

$$\mathcal{L}^{\text{EMD}}(\mathbf{x}, \mathbf{y}) = \min_{\gamma: \mathbf{x} \to \mathbf{y}} \sum_{x \in \mathbf{x}} ||x - \gamma(x)||_2, \tag{17}$$

where $\gamma$ denotes a bijection between the point clouds $\mathbf{x}$ and $\mathbf{y}$ (with the same number of points). Note that we are using an $L_1$ loss for the distance calculation in the CD, which we found to work well and corresponds to the $L_1$ loss we are relying on during original LION training.

Furthermore, $\boldsymbol{\mu}_\xi(\mathbf{h}_0, \mathbf{z}_0)$ in Eq. (15) formally denotes the deterministic decoder output given the latent encodings $\mathbf{z}_0$ and $\mathbf{h}_0$, this is, the mean of the Laplace distribution $p_\xi(\mathbf{x}|\mathbf{h}_0, \mathbf{z}_0)$. The weights for the Kullback-Leibler (KL) terms in Eq. (13), $\lambda_{\mathbf{z}}$ and $\lambda_{\mathbf{h}}$, are generally kept the same as during original LION training. Training with the above objectives ensures that the encoder maps perturbed inputs $\tilde{\mathbf{x}}$ to latent encodings that will decode to the original clean shapes $\mathbf{x}$. This is because maximizing the ELBO (or an adaptation like above) with respect to the encoders $q_\phi(\mathbf{z}_0|\tilde{\mathbf{x}})$ and $q_\phi(\mathbf{h}_0|\tilde{\mathbf{x}}, \mathbf{z}_0)$ while keeping the

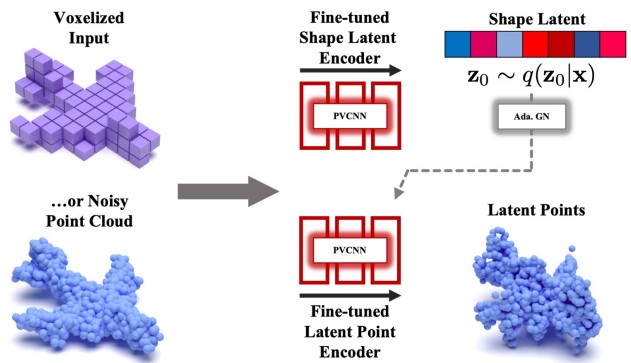

Figure 18: LION's encoder networks can be fine-tuned to process voxel or noisy point cloud inputs, which can provide guidance to the generative model.

decoder fixed is equivalent to minimizing the KL divergences $D_{\text{KL}}\left(q_\phi(\mathbf{z}_0|\tilde{\mathbf{x}})|p_\xi(\mathbf{z}_0|\mathbf{x})\right)$ and $\mathbb{E}_{q_\phi(\mathbf{z}_0|\tilde{\mathbf{x}})}[D_{\text{KL}}\left(q_\phi(\mathbf{h}_0|\tilde{\mathbf{x}}, \mathbf{z}_0)|p_\xi(\mathbf{h}_0|\mathbf{x}, \mathbf{z}_0)\right)]$ with respect to the encoders, where $p_\xi(\mathbf{z}_0|\mathbf{x})$ and $p_\xi(\mathbf{h}_0|\mathbf{x}, \mathbf{z}_0)$ are the true posterior distributions given *clean* shapes $\mathbf{x}$ [70, 132, 133]. Consequently, the fine-tuned encoders are trained to use the noisy or voxel inputs $\tilde{\mathbf{x}}$ to predict the true posterior distributions over the latent variables $\mathbf{z}_0$ and $\mathbf{h}_0$ given the clean shapes $\mathbf{x}$. A user can therefore utilize these fine-tuned encoders to reconstruct clean shapes from noisy or voxelized inputs. Importantly, once the fine-tuned encoder predicts an encoding we can further refine it, clean up imperfect encodings, and sample different shape variations by a few steps of *diffuse-denoise* in latent space (see previous App. C.1). This allows for multimodal denoising and voxel-driven synthesis (also see Fig. 4).

One question that naturally arises is regarding the processing of the noisy or voxelized input shapes. Our PVCNN-based encoder networks can easily process noisy point clouds, but not voxels. Therefore, given a voxelized shape, we uniformly distribute points over the voxelized shape's surface, such that it can be consumed by LION's point cloud processing networks (see details in App. E.4).

We would like to emphasize that LION supports these applications easily without re-training the latent DDMs due to its VAE framework with additional encoders and decoders, in contrast to previous works that train DDMs on point clouds directly [46, 47]. For instance, PVD [46] operates directly on the voxelized or perturbed point clouds with its DDM. Because of that PVD needs to perform many steps of *diffuse-denoise* to remove all the noise from the input—there is no encoder that can help with that. However, this has the drawback that this induces significant shape variations that do not well correspond to the original noisy or voxelized inputs (see experiments and discussion in Sec. 5.4).

### C.3 Shape Interpolation

Here, we explain in detail how exactly we perform shape interpolation. It may be instructive to take a step back first and motivate our approach. Of course, we cannot simply linearly interpolate two point clouds, this is, the points' $xyz$-coordinates, directly. This would result in unrealistic outputs along the interpolation path. Rather, we should perform interpolation in a space where semantically similar point clouds are mapped near each other. One option that comes to mind is to use the latent space, this is, both the shape latent space and the latent points, of LION's point cloud VAE. We could interpolate two point clouds' encodings, and then decode back to point cloud space. However, we also do not have any guarantees in this situation, either, due to the VAE's prior hole problem [58, 72–79], this is, the problem that the distribution of all encodings of the training data won't perfectly form a Gaussian, which it was regularized towards during VAE training (see Eq. (5)). Hence, when simply interpolating directly in the VAE's latent space, we would pass regions in latent space for which the decoder does not produce a realistic sample. This would result in poor outputs.

Therefore, we rather interpolate in the prior spaces of our latent DDMs themselves, this is, the spaces that emerge at the end of the forward diffusion processes. Since the diffusion process of DDMs by construction perturbs all data points into almost perfectly Gaussian $\mathbf{x}_1 \sim \mathcal{N}(\mathbf{x}_1; \mathbf{0}, \mathbf{I})$ (where $t = 1$ denotes the end of the diffusion for continuous $t \in [0, 1]$), DDMs do not suffer from any prior hole challenges—the denoising model is essentially well trained for all possible $\mathbf{x}_1 \sim \mathcal{N}(\mathbf{x}_1; \mathbf{0}, \mathbf{I})$. Hence, given two $\mathbf{x}_1^A$ and $\mathbf{x}_1^B$, in DDMs we can safely interpolate them according to

$$\mathbf{x}_1^s = \sqrt{s}\,\mathbf{x}_1^A + \sqrt{1-s}\,\mathbf{x}_1^B \qquad (18)$$

for $s \in [0, 1]$ and expect meaningful outputs when generating the corresponding denoised samples.

But why do we choose the square root-based interpolation? Since we are working in a very high-dimensional space, we know that according to the Gaussian annulus theorem both $\mathbf{x}_1^A$ and $\mathbf{x}_1^B$ are almost certainly lying on a thin (high-dimensional) spherical shell that supports almost all probability mass of $p_1(\mathbf{x}_1) \approx \mathcal{N}(\mathbf{x}_1; \mathbf{0}, \mathbf{I})$. Furthermore, since $\mathbf{x}_1^A$ and $\mathbf{x}_1^B$ are almost certainly orthogonal to each other, again due to the high dimensionality, our above interpolation in Eq. (18) between $\mathbf{x}_1^A$ and $\mathbf{x}_1^B$ corresponds to performing *spherical interpolation* along the spherical shell where almost all probability mass concentrates. In contrast, linear interpolation would leave this shell, which resulted in poorer results, because the model wasn't well trained for denoising samples outside the typical set. Note that we found spherical interpolation to be crucial (in DDMs of images, linear interpolation tends to still work decently; for our latent point DDM, however, linear interpolation performed very poorly).

In LION, we have two DDMs operating on the shape latent variables $\mathbf{z}_0$ and the latent points $\mathbf{h}_0$. Concretely, for interpolating two shapes $\mathbf{x}^A$ and $\mathbf{x}^B$ in LION, we first encode them into $\mathbf{z}_0^A$ and $\mathbf{h}_0^A$, as well as $\mathbf{z}_0^B$ and $\mathbf{h}_0^B$. Now, using the generative ODE (see App. B) we further encode these latents into the DDMs' prior distributions, resulting in encodings $\mathbf{z}_1^A$ and $\mathbf{h}_1^A$, as well as $\mathbf{z}_1^B$ and $\mathbf{h}_1^B$ (note that we need to correctly capture the conditioning when using $\boldsymbol{\epsilon}_\psi(\mathbf{h}_t, \mathbf{z}_0, t)$ in the generative ODE for $\mathbf{h}_t$). Next, we first interpolate the shape latent DDM encodings $\mathbf{z}_1^s = \sqrt{s}\mathbf{z}_1^A + \sqrt{1-s}\mathbf{z}_1^B$ and use the generative ODE to deterministically generate all $\mathbf{z}_0^s$ along the interpolation path. Then, we also interpolate the latent point DDM encodings $\mathbf{h}_1^s = \sqrt{s}\mathbf{h}_1^A + \sqrt{1-s}\mathbf{h}_1^B$ and, conditioned on the corresponding $\mathbf{z}_0^s$ along the interpolation path, also generate deterministically all $\mathbf{h}_0^s$ along the interpolation path using the generative ODE. Finally, we can decode all $\mathbf{z}_0^s$ and $\mathbf{h}_0^s$ along the

interpolation $s \in [0, 1]$ back to point cloud space and obtain the interpolated point clouds $\mathbf{x}^s$, which we can optionally convert into meshes with SAP.

Note that instead of using given shapes and encoding them into the VAE's latent space and further into the DDMs' prior, we can also directly sample novel encodings in the DDM priors and interpolate those.

In practice, to solve the generative ODE both for encoding and generation, we are using an adaptive step size Runge-Kutta4(5) [67, 134] solver with error tolerances $10^{-5}$. Furthermore, we don't actually solve the ODE all the way to exactly 0, but only up to a small time $10^{-5}$ for numerical reasons (hence, the actual integration interval for the ODE solver is $[10^{-5}, 1]$). We are generally relying on our LION models whose latent DDMs were trained with 1000 discrete time steps (see objectives Eqs. (6) and (7)) and found them to generalize well to the continuous-time setting where the model is also queried for intermediate times $t$ (see discussion in App. B).

## C.4 Mesh Reconstruction with Shape As Points

Before explaining in App. C.4.2 how we incorporate Shape As Points [68] into LION to reconstruct smooth surfaces, we first provide background on Shape As Points in App. C.4.1.

### C.4.1 Background on Shape As Points

Shape As Points (SAP) [68] reconstructs 3D surfaces from points by finding an indicator function $\chi : \mathbb{R}^3 \to \mathbb{R}$ whose zero level set corresponds to the reconstructed surface. To recover $\chi$, SAP first densifies the input point cloud $X = \{x_i \in \mathbb{R}^3\}_{i=1}^N$ by predicting $k$ offsetted points and normals for each input point—such that in total we have additional points $X' = \{x'_i\}_{i=1}^{kN}$ and normals $N' = \{n'_i\}_{i=1}^{kN}$—using a neural network $f_\theta(X)$ with parameters $\theta$ conditioned on the input point cloud $X$.

After upsampling the point cloud and predicting normals, SAP solves a Poisson partial differential equation (PDE) to recover the function $\chi$ from the densified point cloud. Casting surface reconstruction as a Poisson problem is a widely used approach first introduced by Kazhdan et al. [135]. Unlike Kazhdan et al. [135], which encodes $\chi$ as a linear combination of sparse basis functions and solves the PDE using a finite element solver on an octree, SAP represents $\chi$ in a discrete Fourier basis on a dense grid and solves the problem using a spectral solver. This spectral approach has the benefits of being fast and differentiable, at the expense of cubic (with respect to the grid size) memory consumption.

To train the upsampling network $f$, SAP minimizes the $L_2$ distance between the predicted indicator function $\chi$ (sampled on a dense, regular grid) and a pseudo-ground-truth indicator function $\chi_{\text{gt}}$ recovered by solving the same Poisson PDE on a dense set of points and normals. Denoting the differentiable Poisson solve as $\chi = \text{Poisson}(X', N')$, we can write the loss minimized by SAP as

$$\mathcal{L}(\theta) = \mathbb{E}_{\{X_i, \chi_i \sim \mathcal{D}\}} \|\text{Poisson}(f_\theta(X_i)) - \chi_i\|_2^2 \qquad (19)$$

where $\mathcal{D}$ is the training data distribution of indicator functions $\chi_i$ for shapes and point samples on the surface of those shapes $X_i$.

Since the ideas from Poisson Surface Reconstruction (PSR) [135] lie at the core of Shape As Points, we give a brief overview of the Poisson formulation for surface reconstruction: Given input points $X = \{x_i \in \mathbb{R}^3\}_{i=1}^N$ and normals $N = \{n_i \in \mathbb{R}^3\}_{i=1}^N$, we aim to recover an indicator function $\chi : \mathbb{R}^3 \to \mathbb{R}$ such that the reconstructed surface $S$ is the zero level set of $\chi$, i.e., $S = \{x : \chi(x) = 0\}$.

Intuitively, we would like the recovered $\chi$ to change sharply between a positive value and a negative value at the surface boundary along the direction orthogonal to the surface. Thus, PSR treats the surface normals $N$ as noisy samples of the gradient of $\chi$. In practice, PSR first constructs a smoothed vector field $\vec{V}$ from $N$ by convolving these with a filter (e.g. a Gaussian), and recovers $\chi$ by minimizing

$$\min_\chi \|\nabla\chi - \vec{V}\|_2^2 \qquad (20)$$

over the input domain. Observe that applying the (linear) divergence operator to the problem in Eq. (20) does not change the solution. Thus, we can apply the divergence operator to Eq. (20) to

transform it into a Poisson problem

$$\Delta\chi = \nabla \cdot \vec{V}, \tag{21}$$

which can be solved using standard numerical methods for solving Elliptic PDEs. Since PSR is effectively integrating $\vec{V}$ to recover $\chi$, the solution is ambiguous up to an additive constant. To remedy this, PSR subtracts the mean value of $\chi$ at the input points, i.e., $\frac{1}{N}\sum_{i=1}^{N}\chi(x_i)$, yielding a unique solution.

### C.4.2   Incorporating Shape As Points in LION

LION is primarily set up as a point cloud generative model. However, an artist may prefer a mesh as output of the model, because meshes are still the most commonly used shape representation in graphics software. Therefore, we are augmenting LION with mesh reconstruction, leveraging SAP. In particular, given a generated point cloud from LION, we use SAP to predict an indicator function $\chi$ defining a smooth shape surface in $\mathbb{R}^3$ as its zero level set, as explained in detail in the previous section. Then, we extract polygonal meshes from $\chi$ via marching cubes [136].

SAP is commonly trained using slightly noisy perturbed point clouds as input to its neural network $f_\theta$ [68]. This results in robustness and generalization to noisy shapes during inference. Also, the point clouds generated by LION are not perfectly clean and smooth but subject to some noise. In principle, to make our SAP model ideally suited for reconstructing surfaces from LION's generated point clouds, it would be best to train SAP using inputs that are subject to the same noise as generated by LION. Although we do not know the exact form of LION's noise, we propose to nevertheless specialize the SAP model for LION: Specifically, we take SAP's clean training data (i.e. densely sampled point clouds from which accurate pseudo-ground-truth indicator functions can be calculated via PSR; see previous App. C.4.1) and encode it into LION's latent spaces $\mathbf{z}_0$ and $\mathbf{h}_0$. Then, we perform a few *diffuse-denoise* steps in latent space (see App. C.1) that create small shape variations of the input shapes when decoded back to point clouds. However, when doing these *diffuse-denoise* steps, we are exactly using LION's generation mechanism, i.e., the stochastic sampling in Eq. (4), to generate the slightly perturbed encodings. Hence, we are injecting the same noise that is also seen in generation. Therefore, the correspondingly generated point clouds can serve as slightly noisy versions of the original clean point clouds before encoding, diffuse-denoise, and decoding, and we can use this data to train SAP. We found experimentally that this LION-specific training of SAP can indeed improve SAP's performance when reconstructing meshes from LION's generated point clouds. We investigate this experimentally in App. F.1.4.

Note that in principle an even tighter integration of SAP with LION would be possible. In future versions of LION, it would be interesting to study joint end-to-end LION and SAP training, where LION's decoder directly predicts a dense set of points with normals that is then matched to a pseudo-ground-truth indicator function using differentiable PSR. However, we are leaving this to future research. To the best of our knowledge, LION is the first point cloud generative model that directly incorporates modern surface and mesh reconstruction at all. In conclusion, using SAP we can convert LION into a mesh generation model, while under the hood still leveraging point clouds, which are ideal for DDM-based modeling.

## D   Implementation

In Fig. 19, we plot the building blocks used in LION:

- Multilayer perceptron (MLP), point-voxel convolution (PVC), set abstraction (SA), and feature propagation (FP) represent the building modules for our PVCNNs. The Grouper block (in SA) consists of the sampling layer and grouping layer introduced by PointNet++ [82].

- PVCNN visualizes a typical network used in LION. Both the latent points encoder, decoder and the latent point prior share this high-level architecture design, which is modified from the base network of PVD[2] [46]. It consists of some set abstraction levels and feature propagation levels. The details of these levels can be found in PointNet++ [82].

- ResSE denotes a ResNet block with squeeze-and-excitation (SE) [137] layers.

---

[2] https://github.com/alexzhou907/PVD

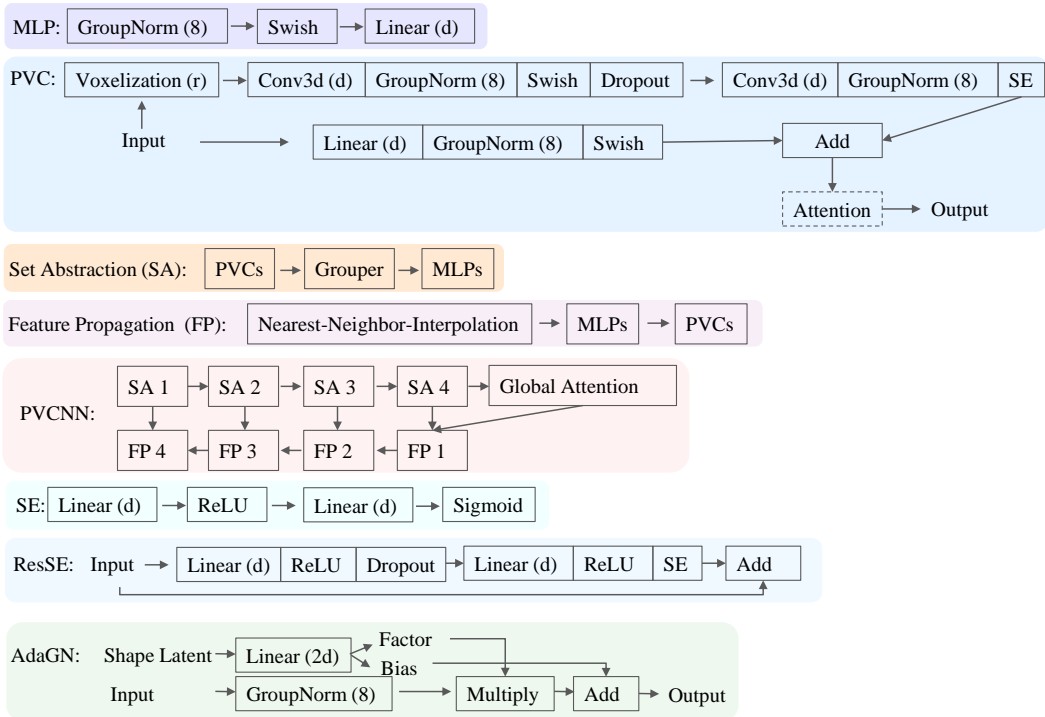

Figure 19: Building blocks of LION. We denote the voxel grid size as $r$, and the hidden dimension as $d$. They are hyperparameters of our model. All GroupNorm layers use 8 groups.

- AdaGN is the adaptive group normalization (GN) layer that is used for conditioning on the shape latent.

## D.1 VAE Backbone

Our VAE backbone consists of two encoder networks, and a decoder network. The PVCNNs we used are based on PointNet++ [82] with point-voxel convolutions [80].

We show the details of the shape latent encoder in Tab. 5, the latent points encoder in Tab. 6, and the details of the decoder in Tab. 7.

We use a dropout probability of 0.1 for all dropout layers in the VAE. All group normalization layers in the latent points encoder as well as in the decoder are replaced by adaptive group normalization (AdaGN) layers to condition on the shape latent. For the AdaGN layers, we initialized the weight of the linear layer with scale at 0.1. The bias for the output factor is set as 1.0 and the bias for the output bias is set as 0.0. The AdaGN is also plot in Fig. 19.

**Model Initialization.** We initialize our VAE model such that it acts as an identity mapping between the input, the latent space, and reconstructed points at the beginning of training. We achieve this by scaling down the variances of encoders and by weighting the skip connections accordingly.

**Weighted Skip Connection.** We add skip connections in different places to improve information propagation. In the latent points encoder, the clean point cloud coordinates (in 3 channels) are added to the mean of the predicted latent point coordinates (in 3 channels), which is multiplied by 0.01 before the addition. In the decoder, the sampled latent points coordinates are added to the output point coordinates (in 3 channels). The predicted output point coordinates are multiplied by 0.01 before the addition.

**Variance Scaling.** We subtract the log of the standard deviation of the posterior Normal distribution with a constant value. The constant value helps pushing the variance of the posterior towards zero when the LION model is initialized. In our experiments, we set this offset value as 6.

With the above techniques, at the beginning of training the input point cloud is effectively copied into the latent point cloud and then directly decoded back to point cloud space, and the shape latent variables are not active. This prevents diverging reconstruction losses at the beginning of training.

| Input: point clouds ($2048 \times 3$) | | |
|---|---|---|
| Output: shape latent ($128 \times 1$) | | |
| | SA 1 | SA 2 |
| # PVC layers | 2 | 1 |
| # PVC hidden dimension | 32 | 32 |
| # PVC voxel grid size | 32 | 16 |
| # Grouper center | 1024 | 256 |
| # Grouper radius | 0.1 | 0.2 |
| # Grouper neighbors | 32 | 32 |
| # MLP layers | 2 | 2 |
| # MLP output dimension | 32,32 | 32,64 |
| Use attention | False | True |
| # Attention dimension | - | 128 |
| Linear: (64, 128) | | |

Table 5: Shape Latent Encoder Architecture Hyperparameters.

### D.2 Shape Latent DDM Prior

We show the details of the shape latent DDM prior in Tab. 8. We use a dropout probability of 0.3, 0.3, and 0.4 for the airplane, car, and chair category, respectively. The time embeddings are added to the features of each ResSE layer.

### D.3 Latent Points DDM Prior

We show the details of the latent points DDM prior in Tab. 9. We use a dropout probability of 0.1 for all dropout layers in this DDM prior. All group normalization layers are replaced by adaptive group normalization layers to condition on the shape latent variable. The time embeddings are concatenated with the point features for the inputs of each SA and FP layer.

Note that both latent DDMs use a *mixed denoising score network* parametrization, directly following Vahdat et al. [58]. In short, the DDM's denoising model is parametrized as the analytically ideal denoising network assuming a normal data distribution plus a neural network-based correction. This can be advantageous, if the distribution that is modeled by the DDM is close to normal. This is indeed the case in our situation, because during the first training stage all latent encodings were regularized to fall under a standard normal distribution due to the VAE objective's Kullback-Leibler regularization. Our implementation of the mixed denoising score network technique directly follows Vahdat et al. [58] and we refer the reader there for further details.

### D.4 Two-stage Training

The training of LION consists of two stages:

**First Stage Training.** LION optimizes the modified ELBO of Eq. (5) with respect to the two encoders and the decoder as shown in the main paper. We use the same value for $\lambda_{\mathbf{z}}$ and $\lambda_{\mathbf{h}}$. These KL weights, starting at $10^{-7}$, are annealed linearly for the first 50% of the maximum number of epochs. Their final value is set to 0.5 at the end of the annealing process.

**Second Stage Training.** In this stage, the encoders and the decoder are frozen, and only the two DDM prior networks are trained using the objectives in Eqs. (6) and (7). During training, we first

| Input: point clouds ($2048 \times 3$), shape latent ($1 \times 128$) | | | | |
| :---: | :---: | :---: | :---: | :---: |
| Output: latent points ($2048 \times 2 \times (3 + D_h)$) | | | | |
| | SA 1 | SA 2 | SA 3 | SA 4 |
| # PVC layers | 2 | 1 | 1 | - |
| # PVC hidden dimension | 32 | 64 | 128 | - |
| # PVC voxel grid size | 32 | 16 | 8 | - |
| # Grouper center | 1024 | 256 | 64 | 16 |
| # Grouper radius | 0.1 | 0.2 | 0.4 | 0.8 |
| # Grouper neighbors | 32 | 32 | 32 | 32 |
| # MLP layers | 2 | 2 | 2 | 3 |
| # MLP output dimension | 32,32 | 64,128 | 128,256 | 128,128,128 |
| Use attention | False | True | False | False |
| # Attention dimension | - | 128 | - | - |
| Global attention layer, hidden dimension: 256 | | | | |
| | FP 1 | FP 2 | FP 3 | FP 4 |
| # MLP layers | 2 | 2 | 2 | 3 |
| # MLP output dimension | 128,128 | 128,128 | 128,128 | 128,128,64 |
| # PVC layers | 3 | 3 | 2 | 2 |
| # PVC hidden dimension | 128 | 128 | 128 | 64 |
| # PVC voxel grid size | 8 | 8 | 16 | 32 |
| Use attention | False | True | False | False |
| # Attention dimension | - | 128 | - | - |
| MLP: (64, 128) | | | | |
| Dropout | | | | |
| Linear: ($128, 2 \times (3 + D_h)$) | | | | |

Table 6: Latent Point Encoder Architecture Hyperparameters.

encode the clean point clouds $\mathbf{x}$ with the encoders and sample $\mathbf{z}_0 \sim q_\phi(\mathbf{z}_0|\mathbf{x})$, $\mathbf{h}_0 \sim q_\phi(\mathbf{h}_0|\mathbf{x}, \mathbf{z}_0)$. We then draw the time steps $t$ uniformly from $U\{1, ..., T\}$, then sample the diffused shape latent $\mathbf{z}_t$ and latent points $\mathbf{h}_t$. Our shape latent DDM prior takes $\mathbf{z}_t$ with $t$ as input, and the latent points DDM prior takes $(\mathbf{z}_0, t, \mathbf{h}_t)$ as input. We use the un-weighted training objective (i.e., $w(t) = 1$).

During second stage training, we regularize the prior DDM neural networks by adding spectral normaliation (SN) [138] and a group normalization (GN) loss similar to Vahdat et al. [58]. Furthermore, we record the exponential moving average (EMA) of the latent DDMs' weight parameters, and use the parameter EMAs during inference when calling the DDM priors.

# E   Experiment Details

## E.1   Different Datasets

For the unconditional 3D point cloud generation task, we follow previous works and use the ShapeNet [104] dataset, as pre-processed and released by PointFlow [31]. Also following previous works [31, 46, 47] and to be able to compare with many different baseline methods, we train on three categories: *airplane*, *chair* and *car*. The ShapeNet dataset released by PointFlow consists of 15k points for each shape. During training, 2,048 points are randomly sampled from the 15k points at each iteration. The training set consists of 2,832, 4,612, and 2,458 shapes for airplane, chair and car, respectively. The sample quality metrics are reported with respect to the standard reference set, which consists of 405, 662, and 352 shapes for airplane, chair and car, respectively. During training, we use the same normalization as in PointFlow [31] and PVD [46], where the data is normalized globally across all shapes. We compute the means for each axis across the whole training set, and one standard deviation across all axes and the whole training set. *Note that there is a typo in the caption*

| Input feature size: latent points ($2048 \times (3 + D_h)$), shape latent ($1 \times 128$) | | | | |
|---|---|---|---|---|
| Output feature size: point clouds ($2048 \times 3$) | | | | |
| | SA 1 | SA 2 | SA 3 | SA 4 |
| # PVC layers | 2 | 1 | 1 | - |
| # PVC hidden dimension | 32 | 64 | 128 | - |
| # PVC voxel grid size | 32 | 16 | 8 | - |
| # Grouper center | 1024 | 256 | 64 | 16 |
| # Grouper radius | 0.1 | 0.2 | 0.4 | 0.8 |
| # Grouper neighbors | 32 | 32 | 32 | 32 |
| # MLP layers | 2 | 2 | 2 | 3 |
| # MLP output dimension | 32,64 | 64,128 | 128,256 | 128,128,128 |
| Use attention | False | True | False | False |
| # Attention dimension | - | 128 | - | - |
| Global attention layer, hidden dimension: 256 | | | | |
| | FP 1 | FP 2 | FP 3 | FP 4 |
| # MLP layers | 2 | 2 | 2 | 3 |
| # MLP output dimension | 128,128 | 128,128 | 128,128 | 128,128,64 |
| # PVC layers | 3 | 3 | 2 | 2 |
| # PVC hidden dimension | 128 | 128 | 128 | 64 |
| # PVC voxel grid size | 8 | 8 | 16 | 32 |
| Use attention | False | False | False | False |
| MLP: (64, 128) | | | | |
| Dropout | | | | |
| Linear: (128, 3) | | | | |

Table 7: Decoder Architecture Hyperparameters.

| Input: latent points ($2048 \times (3 + D_h)$) at $t$ |
|---|
| shape latent ($1 \times 128$) |
| Output: $2048 \times (3 + D_h)$ |
| Time embedding layer: |
| Sinusoidal embedding dimension = 128 |
| Linear (128, 512) |
| LeakyReLU (0.1) |
| Linear (2048) |
| Linear (128, 2048) |
| Addition (linear output, time embedding) |
| ResSE (2048, 2048) x 8 |
| Linear (2048, 128) |

Table 8: Shape Latent DDM Architecture Hyperparameters.

*of Tab. 1 in the main text: In fact, this kind of global normalization using standard deviation does not result in $[-1, 1]$ point coordinate bounds, but the coordinate values usually extend beyond that.*

When reproducing the baselines on the ShapeNet dataset released by PointFlow [31], we found that some methods [19, 43, 45, 52] require per-shape normalization, where the mean is computed for each axis for each shape, and the scale is computed as the maximum length across all axes for each shape. As a result, the $xyz$-values of the point coordinates will be bounded within $[-1, 1]$. We train and evaluate LION following this convention [45] when comparing it to these methods. Note that these different normalizations imply different generative modeling problems. Therefore, it is important to carefully distinguish these different setups for fair comparisons.

| Input: latent points ($2048 \times (3 + D_h)$) at $t$, shape latent ($1 \times 128$) |
|:---:|
| Output: $2048 \times (3 + D_h)$ |
| Time embedding Layer: |
| Sinusoidal embedding dimension = 64
Linear (64, 64)
LeakyReLU(0.1)
Linear (64, 64) |

| | SA 1 | SA 2 | SA 3 | SA 4 |
|---|:---:|:---:|:---:|:---:|
| # PVC layers | 2 | 1 | 1 | - |
| # PVC hidden dimension | 32 | 64 | 128 | - |
| # PVC voxel grid size | 32 | 16 | 8 | - |
| # Grouper center | 1024 | 256 | 64 | 16 |
| # Grouper radius | 0.1 | 0.2 | 0.4 | 0.8 |
| # Grouper neighbors | 32 | 32 | 32 | 32 |
| # MLP layers | 2 | 2 | 2 | 3 |
| # MLP output dimension | 32,64 | 64,128 | 128,128 | 128,128,128 |
| Use attention | False | True | False | False |
| # Attention dimension | - | 128 | - | - |

| Global Attention Layer, hidden dimension: 256 |
|:---:|

| | FP 1 | FP 2 | FP 3 | FP 4 |
|---|:---:|:---:|:---:|:---:|
| # MLP layers | 2 | 2 | 2 | 3 |
| # MLP output dimension | 128,128 | 128,128 | 128,128 | 128,128,64 |
| # PVC layers | 3 | 3 | 2 | 2 |
| # PVC hidden dimension | 128 | 128 | 128 | 64 |
| # PVC voxel grid size | 8 | 8 | 16 | 32 |
| Use attention | False | False | False | False |

| MLP: (64, 128)
Dropout
Linear: (128, 3) |
|:---:|

Table 9: Latent Point DDM Architecture Hyperparameters.

When training the SAP model, we follow Peng et al. [68], Mescheder et al. [100] and also use their data splits and data pre-processing to get watertight meshes. Watertight meshes are required to properly determine whether points are in the interior of the meshes or not, and to define signed distance fields (SDFs) for volumetric supervision, which the PointFlow data does not offer. More details of the data processing can be found in Mescheder et al. [100] (Sec. 1.2 in the Supplementary Material). This dataset variant is denoted as *ShapeNet-vol*. This data is per-shape normalized, i.e., the points' coordinates are bounded by $[-1, 1]$. To combine LION and SAP, we also train LION on the same data used by the SAP model. Therefore, we report sample quality of LION as well as the most relevant baselines DPM, PVD, and also IM-GAN (which synthesizes shapes as SDFs) also on this dataset variant. The number of training shapes is 2,832, 1,272, 1,101, 5,248, 4,746, 767, 1,624, 1,134, 1,661, 2,222, 5,958, 737, and 1,359 for airplane, bench, cabinet, car, chair, display, lamp, loudspeaker, rifle, sofa, table, telephone, and watercraft, respectively. The number of shapes in the reference set is 404, 181, 157, 749, 677, 109, 231, 161, 237, 317, 850, 105, and 193 for airplane, bench, cabinet, car, chair, display, lamp, loudspeaker, rifle, sofa, table, telephone, and watercraft, respectively.

## E.2 Evaluation Metrics

Different metrics to quantitatively evaluate the generation performance of point cloud generative models have been proposed, and some of them suffer from certain drawbacks. Given a generated set of point clouds $S_g$ and a reference set $S_r$, the most popular metrics are (we are following Yang et al. [31]):

- **Coverage (COV)**:

$$\text{COV}(S_g, S_r) = \frac{|\{\arg\min_{Y \in S_r} D(X, Y) | X \in S_g\}|}{|S_r|}, \tag{22}$$

where $D(\cdot, \cdot)$ is either the Chamfer distance (CD) or earth mover distance (EMD). COV measures the number of reference point clouds that are matched to at least one generated shape. COV can quantify diversity and is sensitive to mode dropping, but it does not quantify the quality of the generated point clouds. Also low quality but diverse generated point clouds can achieve high coverage scores.

- **Minimum matching distance (MMD)**:

$$\text{MMD}(S_g, S_r) = \frac{1}{|S_r|} \sum_{Y \in S_r} \min_{X \in S_g} D(X, Y), \tag{23}$$

where again $D(\cdot, \cdot)$ is again either CD or EMD. The idea behind MMD is to calculate the average distance between the point clouds in the reference set and their closest neighbors in the generated set. However, MMD is not sensitive to low quality points clouds in $S_g$, as they are most likely not matched to any shapes in $S_r$. Therefore, it is also not a reliable metric to measure overall generation quality, and it also does not quantify diversity or mode coverage.

- **1-nearest neighbor accuracy (1-NNA)**: To overcome the drawbacks of COV and MMD, Yang et al. [31] proposed to use 1-NNA as a metric to evaluate point cloud generative models:

$$\text{1-NNA}(S_g, S_r) = \frac{\sum_{X \in S_g} \mathbb{I}[N_X \in S_g] + \sum_{Y \in S_r} \mathbb{I}[N_Y \in S_r]}{|S_g| + |S_r|}, \tag{24}$$

where $\mathbb{I}[\cdot]$ is the indicator function and $N_X$ is the nearest neighbor of $X$ in the set $S_r \cup S_g - \{X\}$ (i.e., the union of the sets $S_r$ and $S_g$, but without the particular point cloud $X$). Hence, 1-NNA represents the leave-one-out accuracy of the 1-NN classifier defined in Eq. (24). More specifically, this 1-NN classifier classifies each sample as belonging to either $S_r$ or $S_g$ based on its nearest neighbor sample within $N_X$ (nearest neighbors can again be computed based on either CD or EMD). If the generated $S_g$ matches the reference $S_r$ well, this classification accuracy will be close to 50%. Hence, this accuracy can be used as a metric to quantify point cloud generation performance. Importantly, 1-NNA directly quantifies distribution similarity between $S_r$ and $S_g$ and measures both quality and diversity.

Following Yang et al. [31], we can conclude that COV and MMD are potentially unreliable metrics to quantify point cloud generation performance and 1-NNA seems like a more suitable evaluation metric. Also the more recent and very relevant PVD [46] follows this and uses 1-NNA as its primary evaluation metric. Note that also Jensen-Shannon Divergence (JSD) is sometimes used to quantify point cloud generation performance. However, it measures only the "average shape" similarity by marginalizing over all point clouds from the generated and reference set, respectively. This makes it an almost meaningless metric to quantify individual shape quality (see discussion in Yang et al. [31]).

In conclusion, we are following Yang et al. [31] and Zhou et al. [46] and use 1-NNA as our primary evaluation metric to quantify point cloud generation performance and we evaluate it generally both using CD and EMD distances, according to the following standard definitions:

$$\text{CD}(X, Y) = \sum_{x \in X} \min_{y \in Y} ||x - y||_2^2 + \sum_{y \in Y} \min_{x \in X} ||x - y||_2^2, \tag{25}$$

$$\text{EMD}(X, Y) = \min_{\gamma: X \to Y} \sum_{x \in X} ||x - \gamma(x)||_2, \tag{26}$$

where $\gamma$ is a bijection between point clouds $X$ and $Y$ with the same number of points. We use released codes to compute CD[3] and EMD[4].

Since COV and MMD are still widely used in the literature, though, we are also reporting COV and MMD for all our models in App. F, even though they may be unreliable as metrics for generation

---

[3]https://github.com/ThibaultGROUEIX/ChamferDistancePytorch (MIT License)
[4]https://github.com/daerduoCarey/PyTorchEMD

quality. Note that for the more meaningful 1-NNA metric, LION generally outperforms all baselines in all experiments.

For fair comparisons and to quantify LION's performance in isolation without SAP-based mesh reconstruction, all metrics are computed directly on LION's generated point clouds, not meshed outputs. However, we also do calculate generation performance after the SAP-based mesh reconstruction in a separate ablation study (see App. F.1.4). In those cases, we sample points from the SAP-generated surface to create the point clouds for evaluation metric calculation. Similarly, when calculating metrics for the IM-GAN [7] baseline we sample points from the implicitly defined surfaces generated by IM-GAN. Analogously, for the GCA [43] baseline we sample points from the generated voxels' surfaces.

### E.3 Details for Unconditional Generation

We list the hyperparameters used for training the unconditional generative LION models in Tab. 10. The hyperparameters are the same for both the single class model and many-class model. Notice that we do not perform any hyperparameter tuning on the many-class model, i.e., it is likely that the many-class LION can be further improved with some tuning of the hyperparameters.

| Keys | Values |
|---|---|
| **VAE Training** | |
| Learning rate | $1 \times 10^{-3}$ |
| Optimizer | Adam |
| Beta_1 of Adam | 0.9 |
| Beta_2 of Adam | 0.99 |
| Batch size | 128 |
| KL weights ($\lambda_{\mathbf{z}}, \lambda_{\mathbf{h}}$) start | $1 \times 10^{-7}$ |
| KL weights ($\lambda_{\mathbf{z}}, \lambda_{\mathbf{h}}$) end | 0.5 |
| KL anneal schedule | Linear |
| # Epochs | 8,000 |
| Shape latent dimension | 128 |
| Latent points dimension | 3 + 1 |
| Skip weight | 0.01 |
| Variance offset | 6.0 |
| **Latent DDM Training** | |
| Learning rate | $2 \times 10^{-4}$ |
| Optimizer | Adam |
| Optimizer weight decay | $3 \times 10^{-4}$ |
| Beta_1 of Adam | 0.9 |
| Beta_2 of Adam | 0.99 |
| Batch size | 160 |
| # Epochs | 24,000 |
| # Warm up epochs | 20 |
| EMA decay | 0.9999 |
| Weight of regularizer (SN and GN) | $1 \times 10^{-2}$ |
| **Diffusion Process Parameters** | |
| $\beta_0$ | $1 \times 10^{-4}$ |
| $\beta_T$ | 0.02 |
| $\beta_t$ schedule | Linear |
| # Time steps $T$ | 1000 |
| **SAP Training** | |
| Batch size | 32 |
| Optimizer | Adam |
| Learning rate | $1 \times 10^{-4}$ |
| Standard deviation of the noise | 0.005 |

Table 10: LION's Training and Model Hyperparameters.

When tuning the model for unconditional generation, we found that the dropout probability and the hidden dimension for the shape latent DDM prior have the largest impact on the model performance. The other hyperparameters, such as the size of the encoders and decoder, matter less.

### E.4 Details for Voxel-guided Synthesis

**Setup.** We use a voxel size of 0.6 for both training and testing. During training, the training data (after normalization) are first voxelized, and the six faces of all voxels are collected. The faces that are shared by two or more voxels are discarded. To create point clouds from the voxels, we sample the voxels' faces and then randomly sample points within the faces. In our experiments, 2,048 points are sampled from the voxel surfaces for each shape. We randomly sample a similar number of points at each face.

**Encoder Fine-Tuning.** For encoder fine-tuning, we initialize the model weights from the LION model trained on the same categories with clean data. Both the shape latent encoder and the latent points encoder are fine-tuned on the voxel inputs, while the decoder and the latent DDMs are frozen. We set the maximum training epochs as 10,000 and perform early-stopping when the reconstruction loss on the validation set reaches a minimum value. In our experiments, training is usually stopped early after around 500 epochs. For example, our model on airplane, chair, and car category are stopped at 129, 470, and 189 epochs, respectively. All other hyperparameters are the same as for the unconditional generation experiments. The training objective can be found in Eq. (13) and Eq. (15).

In Fig. 12 and Fig. 13, we report the reconstruction of input points and IOU of the voxels on the test set. We also evaluate the output shape quality by having the models encode and decode the whole training set, and compute the sample quality metrics with respect to the reference set.

Note that we also tried fine-tuning the encoder of the DPM baseline [47]; however, the results did not substantially change. Hence, we kept using standard DPM models.

**Multimodal Generation.** When performing multimodal generation for the voxel-guided synthesis experiments, we encode the voxel inputs into the shape latent $\mathbf{z}_0$ and the latent points $\mathbf{h}_0$, and run the forward diffusion process for a few steps to obtain their diffused versions. The diffused shape latent ($\mathbf{z}_\tau$) is then denoised by the shape latent DDM. The diffused latent points $\mathbf{h}_\tau$ are denoised by the latent points DDM, conditioned on the shape latent generated by the shape latent DDM (also see App. C.1). Thee number of *diffuse-denoise* steps can be found in Figs. 11, 12, and 13.

### E.5 Details for Denoising Experiments

**Setup.** We perturb the input data using different types of noise and show how well different methods denoise the inputs. The experimental setting for each noise type is listed below:

- *Normal Noise*: for each coordinate of a point, we first sample the standard deviation value of the noise uniformly from 0 to 0.25; then, we perturb the point with the noise, sampled from a normal distribution with zero mean and the sampled standard deviation value.

- *Uniform Noise*: for each coordinate of a point, we add noise sampled from the uniform distribution $U(0, 0.25)$.

- *Outlier Noise*: for a shape consisting of $N$ points, we replace 50% of its points with points drawn uniformly from the 3D bounding box of the original shape. The remaining 50% of the points are kept at their original location.

Similar to the encoder fine-tuning for voxel-guided synthesis (App. E.4), when fine-tuning LION's encoder networks for the different denoising experiments, we freeze the latent DDMs and the decoder and only update the weights of the shape latent encoder and the latent points encoder. The maximum number of epochs is set to 4,000 and the training process is stopped early based on the reconstruction loss on the validation set. The other hyperparameters are the same as for the unconditional generation experiments. To get different generations from the same noisy inputs, we again diffuse and denoise in the latent space. The operations are the same as for the multimodal generation during voxel-guided synthesis (App. E.4).

### E.6 Details for Fine-tuning SAP on LION

**Training the Original SAP.** We first train the SAP model on the clean data with normal noise injected, following the practice in SAP [68]. We set the standard deviation of the noise to 0.005.

**Data Preparation.** The training data for SAP fine-tuning is obtained by having LION encode the whole training set, diffuse and denoise in the latent space for some steps, and then decode the point cloud using the decoder. We ablate the number of steps for the diffuse-denoise process in App. F.1.4. In our experiments, we randomly sample the number of steps from $\{20, 30, 35, 40, 50\}$. The number of points used in this preparation process is 3,000, since the SAP model takes 3,000 points as input (since LION is constructed only from PointNet-based and convolutional networks, it can be run with any number of points). To prevent SAP from overfitting to the sampled points, we generate 4 different samples for each shape, with the same number of diffuse-denoise steps. During fine-tuning, SAP randomly draws one sample as input.

**Fine-Tuning.** When fine-tuning SAP, we use the same learning rate, batch size, and other hyperparameters as during training of the original SAP model, except that we change the input and reduce the maximum number of epochs to 1,000.

### E.7 Training Times

For single-class LION models, the total training time is $\approx 550$ GPU hours ($\approx 110$ GPU hours for training the backbone VAE; $\approx 440$ GPU hours for training the two latent diffusion models). Sampling time analyses can be found in App. F.9.

### E.8 Used Codebases

Here, we list all external codebases and datasets we use in our project.

To compare to baselines, we use the following codes:

- r-GAN, l-GAN [2]: `https://github.com/optas/latent_3d_points` (MIT License)
- PointFlow [31]: `https://github.com/stevenygd/PointFlow` (MIT License)
- SoftFlow [32]: `https://github.com/ANLGBOY/SoftFlow`
- Set-VAE [29]: `https://github.com/jw9730/setvae` (MIT License)
- DPF-NET [33]: `https://github.com/Regenerator/dpf-nets`
- DPM [47]: `https://github.com/luost26/diffusion-point-cloud` (MIT License)
- PVD [46]: `https://github.com/alexzhou907/PVD` (MIT License)
- ShapeGF [45]: `https://github.com/RuojinCai/ShapeGF` (MIT License)
- SP-GAN [19]: `https://github.com/liruihui/sp-gan` (MIT License)
- PDGN [52]: `https://github.com/fpthink/PDGN` (MIT License)
- IM-GAN [7]: `https://github.com/czq142857/implicit-decoder` (MIT license) and `https://github.com/czq142857/IM-NET-pytorch` (MIT license)
- GCA [43]: `https://github.com/96lives/gca` (MIT license)

We use further codebases in other places:

- We use the MitSuba renderer for visualizations [125]: `https://github.com/mitsuba-renderer/mitsuba2` (License: `https://github.com/mitsuba-renderer/mitsuba2/blob/master/LICENSE`), and the code to generate the scene discription files for MitSuba [31]: `https://github.com/zekunhao1995/PointFlowRenderer`.
- We rely on SAP [68] for mesh generation with the code at `https://github.com/autonomousvision/shape_as_points` (MIT License).
- For calculating the evaluation metrics, we use the implementation for CD at `https://github.com/ThibaultGROUEIX/ChamferDistancePytorch` (MIT License) and for EMD at `https://github.com/daerduoCarey/PyTorchEMD`.

- We use Text2Mesh [49] for per-sample text-driven texture synthesis: `https://github.com/threedle/text2mesh` (MIT License)

We also rely on the following datasets:

- ShapeNet [104]. Its terms of use can be found at `https://shapenet.org/terms`.
- The Cars dataset [126] from `http://ai.stanford.edu/~jkrause/cars/car_dataset.html` with ImageNet License: `https://image-net.org/download.php`.
- The TurboSquid data repository, `https://www.turbosquid.com`. We obtained a custom license from TurboSquid to use this data.
- Redwood 3DScan Dataset [127]: `https://github.com/isl-org/redwood-3dscan` (Public Domain)
- Pix3D [128]: `https://github.com/xingyuansun/pix3d`. (Creative Commons Attribution 4.0 International License).

## E.9 Computational Resources

The total amount of compute used in this research project is roughly 340,000 GPU hours. We used an in-house GPU cluster of V100 NVIDIA GPUs.

# F Additional Experimental Results

Overview:

- In App. F.1.1, we present an **ablation study** on LION's hierarchical architecture.
- In App. F.1.2, we present an **ablation study** on the point cloud processing backbone neural network architecture.
- In App. F.1.3, we present an **ablation study** on the extra dimensions of the latent points.
- In App. F.1.4, we show an **ablation study** on the number of diffuse-denoise steps used during SAP fune-tuning.
- In App. F.2, we provide additional experimental results on **single-class unconditional generation**. We show MMD and COV metrics, and also incorporate additional baselines in the extended tables. Furthermore, in App. F.2.1 we visualize additional samples from the LION models.
- In App. F.3, we provide additional experimental results for the **13-class unconditional generation** LION model. In App. F.3.1 we show more samples from our many-class LION model. Additionally, in App. F.3.2 we analyze LION's shape latent space via a two-dimensional t-SNE projection [139].
- In App. F.4, we provide additional experimental results for the **55-class unconditional generation** LION model.
- In App. F.5, we provide additional experimental results for the LION models trained on ShapeNet's **Mug and Bottle classes**.
- In App. F.6, we provide additional experimental results for the LION model trained on 3D **animal shapes**.
- In App. F.7, we provide additional results on **voxel-guided synthesis and denoising** for the chair and car categories.
- In App. F.8, we quantify LION's **autoencoding** performance and compare to various baselines, which we all outperform.
- In App. F.9, we provide additional results on significantly **accelerated DDIM**-based synthesis in LION [106].
- In App. F.10, we use **Text2Mesh** [49] to generate textures based on text prompts for synthesized LION samples.

- In App. F.11, we condition LION on CLIP embeddings of the shapes' rendered images, following CLIP-Forge [34]. This allows us to perform **text-driven 3D shape generation** and **single view 3D reconstruction**.
- In App. F.12, we demonstrate more **shape interpolations** using the three single-class and also the 13-class LION models and we also show shape interpolations of the relevant PVD [46] and DPM [47] baselines.

## F.1 Ablation Studies

### F.1.1 Ablation Study on LION's Hierarchical Architecture

We perform an ablation experiment with the car category over the different components of LION's architecture. We consider three settings:

- LION model without shape latents. But it still has latent points and a corresponding latent points DDM prior.
- LION model without latent points. But it still has the shape latents and a corresponding shape latent DDM.
- LION model without any latent variables at all, *i.e.*, a DDM operates on the point clouds directly (this is somewhat similar to PVD [46]).

When simply dropping the different architecture components, the model "loses" parameters. Hence, a decrease in performance could also simply be due to the smaller model rather than an inferior architecture. Therefore, we also increase the model sizes in the above ablation study (by scaling up the channel dimensions of all networks), such that all models have approximately the same number of parameters as our main LION model that has all components. The results on the car category can be found in Tab. 11. The results show that the full LION setup with both shape latents and latent points performs best on all metrics, sometimes by a large margin. Furthermore, for the models with no or only one type of latent variables, increasing model size does not compensate for the loss of performance due to the different architectures. This ablation study demonstrates the unique advantage of the hierarchical setup with both shape latent variables and latent points, and two latent DDMs. We believe that the different latent variables complement each other—the shape latent variables model overall global shape, while the latent points capture details. This interpretation is supported by the experiments in which we keep the shape latent fixed and only observe small shape variations due to different local point latent configurations (Sec. 5.2 and Fig. 8).

| Shape Latents | Latent Points | Num. Params. | MMD↓ | | COV↑ (%) | | 1-NNA↓ | |
|---|---|---|---|---|---|---|---|---|
| | | | CD | EMD | CD | EMD | CD | EMD |
| ✓ | ✓ | 110 | **0.91** | **0.75** | **50.00** | **56.53** | **53.41** | 51.14 |
| | ✓ | 45 | 1.04 | 0.80 | 47.16 | 52.27 | 56.96 | **50.99** |
| ✓ | | 88 | 1.09 | 0.88 | 38.35 | 35.23 | 75.71 | 76.56 |
| | | 27 | 1.19 | 0.81 | 48.01 | 52.56 | 59.94 | 55.26 |
| | ✓ | 124 | 0.96 | 0.80 | 46.02 | 53.69 | 56.82 | 53.41 |
| ✓ | | 111 | 1.12 | 0.89 | 39.20 | 35.80 | 76.56 | 74.72 |
| | | 110 | 1.12 | 0.82 | 48.86 | 52.84 | 58.66 | 55.40 |

Table 11: Ablation study over LION's hierarchical architecture, on the car category.

### F.1.2 Ablation Study on the Backbone Point Cloud Processing Network Architecture

We ablate different point cloud processing neural network architectures used for implementing LION's encoder, decoder and the latent points prior. Results are shown in Tab. 12 and Tab. 13, using the LION model on the car category as in the other ablation studies. We choose three different popular backbones used in the point cloud processing literature: Point-Voxel CNN (PVCNN) [80], Dynamic Graph CNN (DGCNN) [140] and PointTransformer [141]. For the ablation on the encoder and decoder backbones, we train LION's VAE model (without prior) with different backbones, and compare the reconstruction performance for different backbones. We select the PVCNN as it provides

| Method | CD↓ | EMD↓ |
|---|---|---|
| PVCNN | **0.006** | **0.009** |
| DGCNN (knn=20) | 0.068 | 0.231 |
| DGCNN (knn=10) | 0.081 | 0.331 |
| PointTransformer | 0.015 | 0.029 |

Table 12: Ablation on the backbone of the encoder and decoder of LION's VAE. The point cloud auto-encoding performance on the car category is reported. Both CD and EMD reconstruction values are multiplied with $1 \times 10^{-2}$. The same KL weights are applied in all experiments. For DGCNN, we try different numbers of top-k nearest neighbors (knn).

the strongest performance (Tab. 12). For the ablation on the prior backbone, we first train the VAE model with the PVCNN architecture, as in all of our main experiments, and then train the prior with different backbones and compare the generation performance. Again, PVCNN performs best as network to implement the latent points diffusion model (Tab. 13). In conclusion, these experiments support choosing PVCNN as our point cloud neural network backbone architecture for implementing LION.

Note that all ablations were run with similar hyperparameters and the neural networks were generally set up in such a way that different architectures consumed the same GPU memory.

| Backbone | MMD↓ | | COV↑ (%) | | 1-NNA↓ | |
|---|---|---|---|---|---|---|
| | CD | EMD | CD | EMD | CD | EMD |
| PVCNN | **0.91** | **0.75** | **50.00** | **56.53** | **53.41** | **51.14** |
| DGCNN (knn=20) | 1.05 | 0.80 | 41.19 | 52.27 | 66.48 | 55.82 |
| DGCNN (knn=10) | 1.02 | 0.80 | 42.33 | 50.57 | 66.34 | 53.41 |
| PointTransformer | 3.67 | 3.02 | 10.51 | 10.51 | 99.72 | 99.86 |

Table 13: Ablation on the backbone of the latent points prior, on the car category. For DGCNN, we try different numbers of top-k nearest neighbors (knn).

### F.1.3 Ablation Study on Extra Dimensions for Latent Points

Next, we ablate the extra dimension $D_{\mathbf{h}}$ for the latent points in Tab. 14, again using LION models on the car category. We see that $D_{\mathbf{h}} = 1$ provides the overall best performance. With a relatively large number of extra dimensions, it is observed that the 1-NNA scores are getting worse in general. We use $D_{\mathbf{h}} = 1$ for all other experiments.

| Extra Latent Dim | MMD↓ | | COV↑ (%) | | 1-NNA↓ | |
|---|---|---|---|---|---|---|
| | CD | EMD | CD | EMD | CD | EMD |
| 0 | **0.91** | **0.75** | 46.59 | 52.56 | 56.11 | 52.41 |
| 1 | **0.91** | **0.75** | 50.00 | **56.53** | **53.41** | **51.14** |
| 2 | 0.92 | 0.76 | 49.72 | 53.12 | 57.67 | 57.67 |
| 5 | 0.92 | 0.76 | **52.27** | **55.97** | 59.94 | 52.41 |

Table 14: Ablation on the number of extra dimensions $D_{\mathbf{h}}$ of the latent points, on the car category.

### F.1.4 Ablation Study on SAP Fine-Tuning

After applying SAP to extract meshes from the generated point clouds, it is possible to again sample points from the meshed surface and evaluate the points' quality with the generation metrics that we used for unconditional generation. We call this process *resampling*.

See Tab. 15 for an ablation over the results of *resampling* from SAP with or without fine-tuning. It also contains the ablation over different numbers of diffuse-denoise steps used to generate the training data for the SAP fine-tuning. Without fine-tuning, the reconstructed mesh has slightly lower quality according to 1-NNA, presumably since the noise within the generated points is different from the

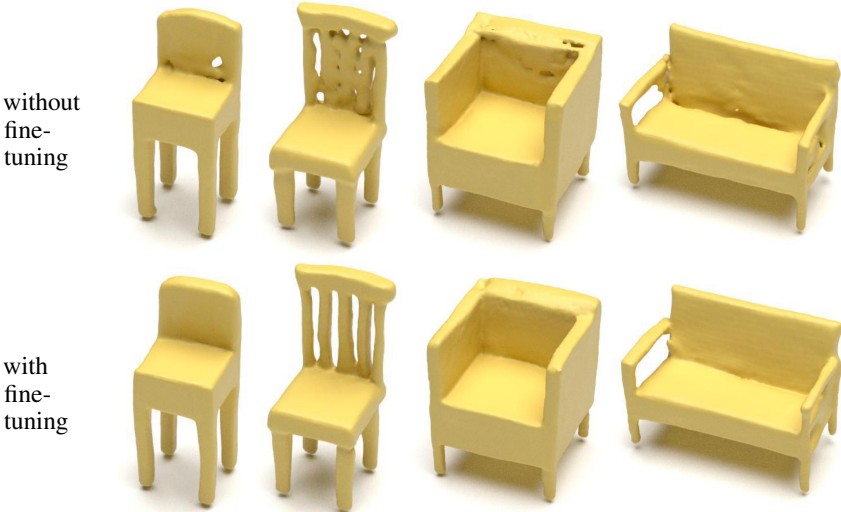

without
fine-
tuning

with
fine-
tuning

Figure 20: Mesh reconstruction results from SAP with the same point cloud inputs. *Top:* without fine-tuning. *Bottom:* with fine-tuning on LION. We see that fine-tuning significantly improves the meshes.

noise which the SAP model is trained on. For the "mixed" number of steps entry in the table, SAP randomly chooses one number of diffuse-denoise steps from the above five values at each iteration when producing the training shapes. This setting tends to give an overall good sample quality in terms of the 1-NNA evaluation metrics. We use this setting in all experiments.

To visually demonstrate the improvement of SAP's mesh reconstruction performance with and without fine-tuning, we show the reconstructed meshes before and after finetuning in Fig. 20. The original SAP is trained with clean point clouds augmented with small Gaussian noise. As a result, SAP can handle small scale Gaussian noise in the point clouds. However, it is less robust to the generated points where the noise is different from the Gaussian noise which SAP is trained with. With our proposed fine-tuning, SAP produces smoother surfaces and becomes more robust to the noise distribution in the point clouds generated by LION.

| resampled from | Num of Step | MMD↓ | | COV↑ (%) | | 1-NNA↓ (%) | |
| | | CD | EMD | CD | EMD | CD | EMD |
|---|---|---|---|---|---|---|---|
| (no resampling, original LION) | - | 2.826 | 1.720 | 47.42 | **50.52** | 60.34 | **57.31** |
| SAP | - | **2.919** | 1.752 | **48.74** | 47.71 | 60.64 | 59.90 |
| SAP fine-tuned | 20 | 2.926 | **1.740** | 48.45 | 48.89 | 59.45 | 58.71 |
| SAP fine-tuned | 30 | 2.931 | 1.754 | 48.15 | 47.42 | 59.45 | 60.04 |
| SAP fine-tuned | 35 | 2.930 | 1.743 | 48.15 | 47.71 | 59.45 | 59.90 |
| SAP fine-tuned | 40 | 2.930 | 1.754 | 48.15 | 47.71 | 60.56 | 59.75 |
| SAP fine-tuned | 50 | 2.939 | 1.746 | 48.15 | 48.45 | 59.38 | 58.64 |
| SAP fine-tuned | mixed | 2.932 | 1.742 | 47.56 | 48.30 | **58.71** | 59.45 |

Table 15: Ablation over number of steps used to generate the training data for SAP, on the chair class. The model is trained on the ShapeNet-vol dataset. 3,000 points are sampled from LION to generate meshes with SAP. However, during evaluation after *resampling*, 2,048 points are used as in all other experiments where we calculate quantitative performance metrics.

## F.2 Single-Class Unconditional Generation

For our three single-class LION models, we show the full evaluation metrics for different dataset splits, and different data normalizations, in Tab. 16, Tab. 17 and Tab. 18. Under all settings and datasets, LION achieves state-of-the-art performance on the 1-NNA metrics, and is competitive on

| Category | Model | MMD↓ | | COV↑ (%) | | 1-NNA↓ (%) | |
|---|---|---|---|---|---|---|---|
| | | CD | EMD | CD | EMD | CD | EMD |
| Airplane | train set | 0.218 | 0.373 | 46.91 | 52.10 | 64.44 | 64.07 |
| | r-GAN [2] | 0.447 | 2.309 | 30.12 | 14.32 | 98.40 | 96.79 |
| | l-GAN (CD) [2] | 0.340 | 0.583 | 38.52 | 21.23 | 87.30 | 93.95 |
| | l-GAN (EMD) [2] | 0.397 | 0.417 | 38.27 | 38.52 | 89.49 | 76.91 |
| | PointFlow [31] | 0.224 | 0.390 | 47.90 | 46.41 | 75.68 | 70.74 |
| | SoftFlow [32] | 0.231 | 0.375 | 46.91 | 47.90 | 76.05 | 65.80 |
| | SetVAE [29] | **0.200** | **0.367** | 43.70 | 48.40 | 76.54 | 67.65 |
| | DPF-Net [33] | 0.264 | 0.409 | 46.17 | 48.89 | 75.18 | 65.55 |
| | DPM [47] | 0.213 | 0.572 | **48.64** | 33.83 | 76.42 | 86.91 |
| | PVD [46] | 0.224 | 0.370 | 48.88 | **52.09** | 73.82 | 64.81 |
| | LION (ours) | 0.219 | 0.372 | 47.16 | 49.63 | **67.41** | **61.23** |
| Chair | train set | 2.618 | 1.555 | 53.02 | 51.21 | 51.28 | 54.76 |
| | r-GAN [2] | 5.151 | 8.312 | 24.27 | 15.13 | 83.69 | 99.70 |
| | l-GAN (CD) [2] | 2.589 | 2.007 | 41.99 | 29.31 | 68.58 | 83.84 |
| | l-GAN (EMD) [2] | 2.811 | 1.619 | 38.07 | 44.86 | 71.90 | 64.65 |
| | PointFlow [31] | **2.409** | 1.595 | 42.90 | 50.00 | 62.84 | 60.57 |
| | SoftFlow [32] | 2.528 | 1.682 | 41.39 | 47.43 | 59.21 | 60.05 |
| | SetVAE [29] | 2.545 | 1.585 | 46.83 | 44.26 | 58.84 | 60.57 |
| | DPF-Net [33] | 2.536 | 1.632 | 44.71 | 48.79 | 62.00 | 58.53 |
| | DPM [47] | 2.399 | 2.066 | 44.86 | 35.50 | 60.05 | 74.77 |
| | PVD [46] | 2.622 | 1.556 | **49.84** | 50.60 | 56.26 | 53.32 |
| | LION (ours) | 2.640 | **1.550** | 48.94 | **52.11** | 53.70 | 52.34 |
| Car | train set | 0.938 | 0.791 | 50.85 | 55.68 | 51.70 | 50.00 |
| | r-GAN [2] | 1.446 | 2.133 | 19.03 | 6.539 | 94.46 | 99.01 |
| | l-GAN (CD) [2] | 1.532 | 1.226 | 38.92 | 23.58 | 66.49 | 88.78 |
| | l-GAN (EMD) [2] | 1.408 | 0.899 | 37.78 | 45.17 | 71.16 | 66.19 |
| | PointFlow [31] | **0.901** | 0.807 | 46.88 | 50.00 | 58.10 | 56.25 |
| | SoftFlow [32] | 1.187 | 0.859 | 42.90 | 44.60 | 64.77 | 60.09 |
| | SetVAE [29] | 0.882 | 0.733 | 49.15 | 46.59 | 59.94 | 59.94 |
| | DPF-Net [33] | 1.129 | 0.853 | 45.74 | 49.43 | 62.35 | 54.48 |
| | DPM [47] | 0.902 | 1.140 | 44.03 | 34.94 | 68.89 | 79.97 |
| | PVD [46] | 1.077 | 0.794 | 41.19 | 50.56 | 54.55 | 53.83 |
| | LION (ours) | 0.913 | **0.752** | **50.00** | **56.53** | **53.41** | **51.14** |

Table 16: Generation performance metrics on Airplane, Chair, Car. MMD-CD is multiplied with $1 \times 10^3$, MMD-EMD is multiplied with $1 \times 10^2$.

the MMD and COV metrics, which, however, can be unreliable with respect to quality (see discussion in App. E.2).

### F.2.1 More Visualizations of the Generated Shapes

More visualizations of the generated shapes from the LION models trained on airplane, chair and car classes can be found in Fig. 32, Fig. 33 and Fig. 34. LION is generating high quality samples with high diversity. We visualize both point clouds and meshes generated with the SAP that is fine-tuned on the VAE-encoded training set.

### F.3 Unconditional Generation of 13 ShapeNet Classes

See Tab. 19 for the evaluation metrics of the sample quality of LION and other baselines, trained on the 13-class dataset. To evaluate the models, we sub-sample 1,000 shapes from the reference set and sample 1,000 shapes from the models. We can see that LION is better than all baselines under this challenging setting. The results are also consistent with our observations on the single-class models. For baseline comparisons, we picked PVD [46] and DPM [47], because they are also DDM-based

| Category | Model | MMD↓ | | COV↑ (%) | | 1-NNA↓ (%) | |
|---|---|---|---|---|---|---|---|
| | | CD | EMD | CD | EMD | CD | EMD |
| Airplane | train set | 0.2953 | 0.5408 | 43.70 | 45.93 | 66.42 | 65.06 |
| | TreeGAN [6] | 0.5581 | 1.4602 | 31.85 | 17.78 | 97.53 | 99.88 |
| | ShapeGF [45] | **0.3130** | **0.6365** | **45.19** | 40.25 | 81.23 | 80.86 |
| | SP-GAN [19] | 0.4035 | 0.7658 | 26.42 | 24.44 | 94.69 | 93.95 |
| | PDGN [52] | 0.4087 | 0.7011 | 38.77 | 36.54 | 94.94 | 91.73 |
| | GCA [43] | 0.3586 | 0.7651 | 38.02 | 36.3 | 88.15 | 85.93 |
| | LION (ours) | 0.3564 | 0.5935 | 42.96 | **47.90** | **76.30** | **67.04** |
| Chair | train set | 3.8440 | 2.3209 | 49.55 | 53.63 | 53.17 | 52.19 |
| | TreeGAN [6] | 4.8409 | 3.5047 | 39.88 | 26.59 | 88.37 | 96.37 |
| | ShapeGF [45] | **3.7243** | 2.3944 | **48.34** | 44.26 | 58.01 | 61.25 |
| | SP-GAN [19] | 4.2084 | 2.6202 | 40.03 | 32.93 | 72.58 | 83.69 |
| | PDGN [52] | 4.2242 | 2.5766 | 43.20 | 36.71 | 71.83 | 79.00 |
| | GCA [43] | 4.4035 | 2.582 | 45.92 | 47.89 | 64.27 | 64.50 |
| | LION (ours) | 3.8458 | **2.3086** | 46.37 | **50.15** | **56.50** | **53.85** |
| Car | train set | 1.0400 | 0.8111 | 50.28 | 53.69 | 55.11 | 49.86 |
| | TreeGAN [6] | 1.1418 | 1.0632 | 40.06 | 31.53 | 89.77 | 94.89 |
| | ShapeGF [45] | **1.0200** | 0.8239 | **44.03** | 47.16 | 61.79 | 57.24 |
| | SP-GAN [19] | 1.1676 | 1.0211 | 34.94 | 31.82 | 87.36 | 85.94 |
| | PDGN [52] | 1.1837 | 1.0626 | 31.25 | 25.00 | 89.35 | 87.22 |
| | GCA [43] | 1.0744 | 0.8666 | 42.05 | 48.58 | 70.45 | 64.20 |
| | LION (ours) | 1.0635 | **0.8075** | 42.90 | **50.85** | 59.52 | **49.29** |

Table 17: Generation performance metrics on Airplane, Chair, Car. Trained on ShapeNet dataset from PointFlow. Both the training and testing data are normalized individually into range [-1, 1].

and most relevant. We also picked TreeGAN [6], as it is also trained on diverse data in their original paper, and DPF-Net [33], as it represents a modern competitive flow-based method that we could train relatively quickly. We did not run all other baselines that we ran for the single-class models due to limited compute resources.

### F.3.1 More Visualizations of the Generated Shapes

See Fig. 35 for more visualizations of the generated shapes from LION trained on the 13-class data. We visualize both point clouds and meshes generated with the SAP that is fine-tuned on the VAE-encoded training set. LION is again able to generate diverse and high quality shapes even when training in the challenging 13-class setting. We also show in Fig. 36 additional samples from the 13-class LION model with fixed shape latent variables, where only the latent points are sampled, similar to the experiments in Sec. 5.2 and Fig. 8. We again see that the shape latent variables seem to capture overall shape, while the latent points are responsible for generating different details.

### F.3.2 Shape Latent Space Visualization

We project the shape latent variables learned by LION's 13-classes VAE into the 2D plane and create a t-SNE [139] plot in Fig. 37. It can be seen that many categories are separated, such as the rifle, car, watercraft, airplane, telephone, lamp, and display classes. The other categories that are hard to distinguish such as bench and table are mixing a bit, which is also reasonable. This indicates that LION's shape latent is learning to represent the category information, presumably capturing overall shape information, as also supported by our experiments in Sec. 5.2 and Fig. 8. Potentially, this also means that the representation learnt by the shape latents could be leveraged for downstream tasks, such as shape classification, similar to Luo and Hu [47].

| Category | Model | MMD↓ | | COV↑ (%) | | 1-NNA↓ (%) | |
|---|---|---|---|---|---|---|---|
| | | CD | EMD | CD | EMD | CD | EMD |
| | train set | 0.8462 | 0.6968 | 52.48 | 53.47 | 50.87 | 50.99 |
| Airplane | IM-GAN [7] | 0.9047 | 0.8205 | 45.54 | 40.10 | 79.70 | 77.85 |
| | DPM [47] | 0.9191 | 1.3975 | 38.86 | 12.13 | 83.04 | 96.04 |
| | PVD [46] | 0.8513 | 0.7198 | 47.52 | 51.73 | 66.46 | 56.06 |
| | LION *(ours)* | **0.8317** | **0.6964** | **53.47** | **53.96** | **53.47** | **53.84** |
| | train set | 2.8793 | 1.6867 | 55.10 | 56.13 | 49.11 | 48.97 |
| Chair | IM-GAN [7] | 2.8935 | 1.7320 | **50.96** | 50.81 | 57.09 | 58.20 |
| | DPM [47] | **2.5337** | 1.9746 | 47.42 | 35.01 | 61.96 | 74.96 |
| | PVD [46] | 2.9024 | 1.7144 | 46.23 | 50.22 | 61.89 | 57.90 |
| | LION *(ours)* | 2.8561 | **1.6898** | 49.78 | **54.51** | **52.07** | **48.67** |
| | train set | 0.8643 | 0.6116 | 52.47 | 52.34 | 49.40 | 51.34 |
| Car | IM-GAN [7] | 1.0843 | 0.7829 | 21.23 | 27.77 | 88.92 | 84.58 |
| | DPM [47] | 0.8880 | 0.8633 | 33.38 | 22.43 | 77.30 | 87.12 |
| | PVD [46] | 0.9041 | 0.6140 | 46.33 | 51.00 | 64.49 | 55.74 |
| | LION *(ours)* | **0.8687** | **0.6092** | **48.20** | **52.60** | 54.81 | **50.53** |

Table 18: Generation performance metrics on Airplane, Chair, Car; trained on ShapeNet-vol dataset version, the same data used by SAP.

| Model | MMD↓ | | COV↑ (%) | | 1-NNA↓ (%) | |
|---|---|---|---|---|---|---|
| | CD | EMD | CD | EMD | CD | EMD |
| train set | 2.4375 | 1.4327 | 52.10 | 54.90 | 50.70 | 50.55 |
| TreeGAN [6] | 5.4971 | 3.4242 | 32.70 | 25.50 | 96.80 | 96.60 |
| PointFlow [31] | 2.7653 | 2.0064 | 47.70 | 49.20 | 63.25 | 66.05 |
| ShapeGF [45] | 2.8830 | 1.8508 | 52.00 | 50.30 | 55.65 | 59.00 |
| SetVAE [29] | 4.7381 | 2.9924 | 40.40 | 32.50 | 79.25 | 95.25 |
| PDGN [52] | 3.4032 | 2.3335 | 42.70 | 37.30 | 71.05 | 86.00 |
| DPF-Net [33] | 3.1976 | 2.0731 | 48.10 | 52.00 | 67.10 | 64.75 |
| DPM [47] | **2.2471** | 2.0682 | 48.10 | 30.80 | 62.30 | 86.50 |
| PVD [46] | 2.3715 | 1.4650 | 48.90 | 53.10 | 58.65 | 57.85 |
| LION *(ours)* | 2.4572 | **1.4472** | 54.30 | 54.40 | 51.85 | **48.95** |

Table 19: Generation performance metrics on 13 classes using our many-class LION model. Trained on ShapeNet-vol dataset.

### F.4 Unconditional Generation of all 55 ShapeNet Classes

We train a LION model *jointly without any class conditioning* on all 55[5] different categories from ShapeNet. The total number of training data is 35,708. Training a single model without conditioning over such a large number of categories is challenging, as the data distribution is highly complex and multimodal. Note that we did on purpose not use class-conditioning to explore LION's scalability to such complex and multimodal datasets. Furthermore, the number of training samples across different categories is imbalanced in this setting: 15 categories have less than 100 training samples and 5 categories have more than 2,000 training samples. We adopt the same model hyperparameters as for the single class LION models here without any tuning.

---

[5] the 55 classes are *airplane, bag, basket, bathtub, bed, bench, birdhouse, bookshelf, bottle, bowl, bus, cabinet, camera, can, cap, car, cellphone, chair, clock, dishwasher, earphone, faucet, file, guitar, helmet, jar, keyboard, knife, lamp, laptop, mailbox, microphone, microwave, monitor, motorcycle, mug, piano, pillow, pistol, pot, printer, remote control, rifle, rocket, skateboard, sofa, speaker, stove, table, telephone, tin can, tower, train, vessel, washer*

We show LION's generated samples in Fig. 21: LION synthesizes high-quality and diverse shapes. It can even generate samples from the *cap* class, which contributes with only 39 training samples, indicating that LION has an excellent mode coverage that even includes the very rare classes. Note that we did not train an SAP model on the 55 classes data. Hence, we only show the generated point clouds in Fig. 21.

This experiment is run primarily as a qualitative scalability test of LION and due to limited compute resources, we did not train baselines here. Furthermore, to the best of our knowledge no previous 3D shape generative models have demonstrated satisfactory generation performance for such diverse and multimodal 3D data without relying on conditioning information. That said, to make sure future works can compare to LION, we report the generation performance over 1,000 samples in Tab. 20. We would like to emphasize that hyperparameter tuning and using larger LION models with more parameters will almost certainly significantly improve the results even further. We simply used the single-class training settings out of the box.

|  | MMD↓ | | COV↑ (%) | | 1-NNA↓ (%) | |
| --- | --- | --- | --- | --- | --- | --- |
| Model | CD | EMD | CD | EMD | CD | EMD |
| LION *(ours)* | 3.4336 | 2.0953 | 48.00 | 52.20 | 58.25 | 57.75 |

Table 20: Generation performance metrics for our LION model that was trained jointly on all 55 ShapeNet classes without class-conditioning.

### F.5 Unconditional Generation of ShapeNet's Mug and Bottle Classes

Next, we explore whether LION can also be trained successfully on very small datasets. To this end, we train LION on the Mug and Bottle classes in ShapeNet. The number of training samples is 149 and 340, respectively, which is much smaller than the common classes like chair, car and airplane. All the hyperparameters are the same as for the models trained on single classes. We show generated shapes in Fig. 22 and Fig. 23 (to extract meshes from the generated point clouds, for convenience we are using the SAP model that was trained for the 13-class LION experiment). We find that LION is also able to generate correct mugs and bottles in this very small training data set situation. We report the performance of the generated samples in Tab.21, such that future work can compare to LION on this task.

| Data | Model | MMD↓ | | COV↑ (%) | | 1-NNA↓ (%) | |
| --- | --- | --- | --- | --- | --- | --- | --- |
|  |  | CD | EMD | CD | EMD | CD | EMD |
| Mug | LION *(ours)* | 4.2163 | 2.8983 | 31.82 | 50.00 | 70.45 | 59.09 |
| Bottle | LION *(ours)* | 2.1941 | 1.4313 | 39.53 | 55.81 | 61.63 | 50.00 |

Table 21: Generation performance metrics for LION trained on ShapeNet's Mug and Bottle classes. Trained on ShapeNet dataset from PointFlow; both the training and testing data are normalized individually into range $[-1, 1]$.

### F.6 Unconditional Generation of Animal Shapes

Furthermore, we also train LION on 553 animal assets from the TurboSquid data repository.[6]. The animal data includes shapes of cats, bears, goats, etc. All the hyperparameters are again the same as for the models trained on single classes. See Fig. 24 for visualizations of the generated shapes from LION trained on the animal data. We visualize both point clouds and meshes. For simplicity, the meshes are generated again with the SAP model that was trained on the ShapeNet 13-classes data. LION is again able to generate diverse and high quality shapes even when training in the challenging low-data setting.

---

[6] https://www.turbosquid.com We obtained a custom license from TurboSquid to use this data.

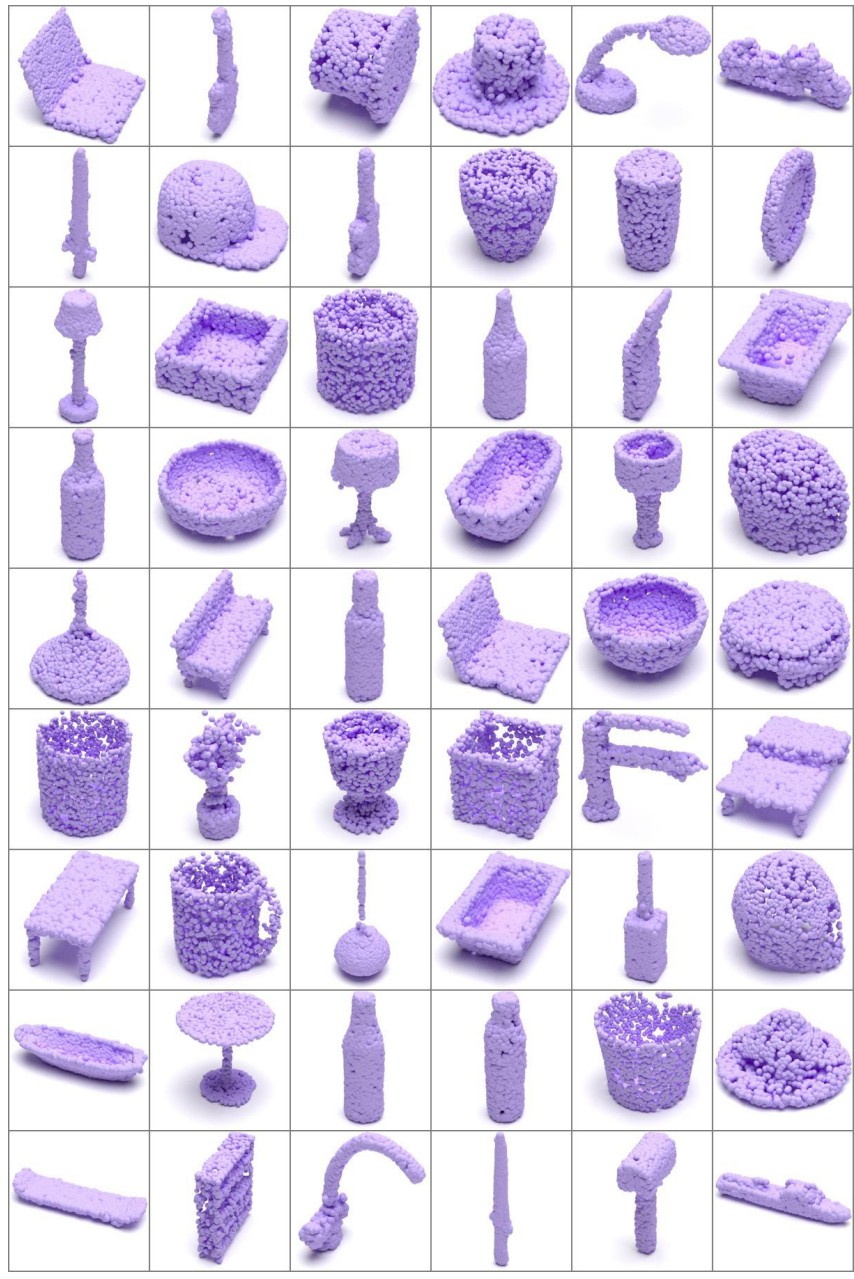

Figure 21: Generated shapes from our LION model that was trained jointly on all 55 ShapeNet classes without class-conditioning.

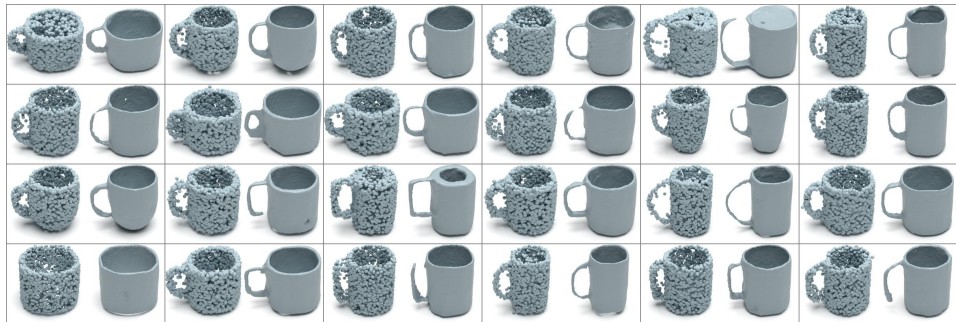

Figure 22: Generated shapes from the LION model trained on ShapeNet's Mug category.

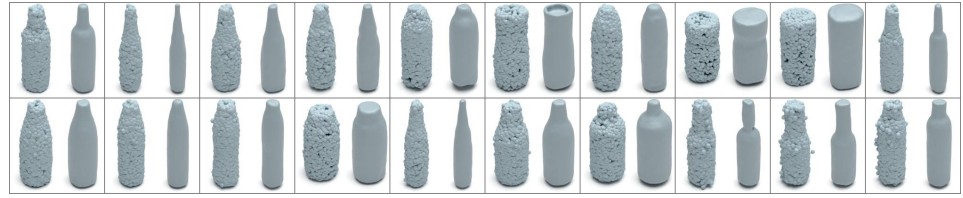

Figure 23: Generated shapes from the LION model trained on ShapeNet's Bottle category.

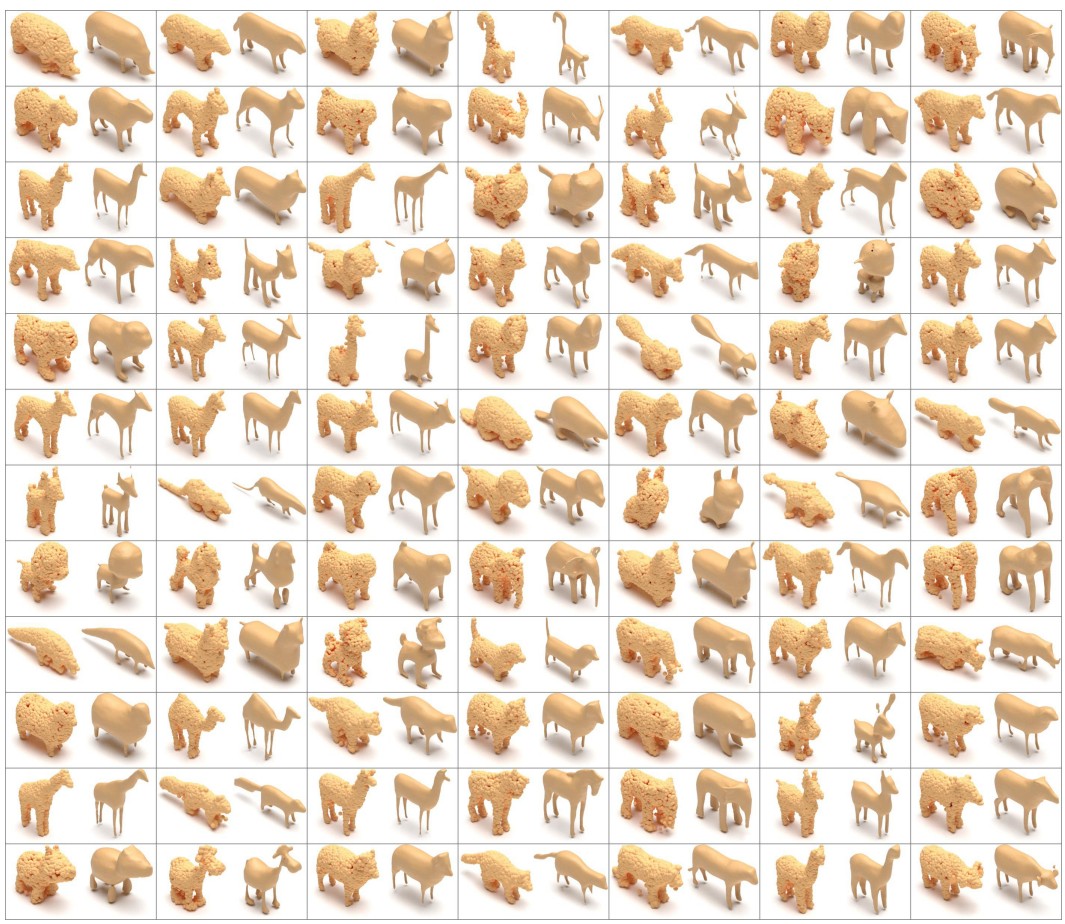

Figure 24: Generated shapes from the animal class LION model.

### F.7 Voxel-guided Synthesis and Denoising

We additionally add the results for voxel-guided synthesis and denoising experiments on the chair and car categories. In Fig. 25 and Fig. 27, we show the reconstruction metrics for different types of input: voxelized input, input with outlier noise, input with uniform noise, and input with normal noise. LION outperforms the other two baselines (PVD and DPM), especially for the voxelized input and the input with outliers, similar to the results presented in the main paper on the airplane class (Sec. 5.4). In Fig. 26 and Fig. 28, we show the output quality metrics and the voxel IOU for voxel-guided synthesis on chair, and car category, respectively. LION achieves high output quality while obeying the voxel input constraint well.

**More on Multimodal Visualization.** In Fig. 30, we show visualizations of multimodal voxel-guided synthesis on different classes. As discussed, we generate various plausible shapes using different numbers of diffuse-denoise steps. We show two different plausible shapes (with the corresponding latent points, and reconstructed meshes) given the same input at each row under different settings. LION is able to capture the structure indicated by the voxel grids: the shapes obey the voxel grid constraints. For example, the tail of the airplane, the legs of the chair, and the back part of the car are consistent with the input. Meanwhile, LION generates diverse and reasonable details in the output shapes.

See Fig. 29 for denoising experiments, with comparisons to other baselines. In Fig. 31, we also show the visualizations for different classes. LION handles different types of input noises and generates reasonable and diverse details given the same input. See the car examples in the first column for the normal noise, uniform noise and the outlier noise.

Notice that we applied the SAP model here only for visualizing the meshed output shapes. The SAP model is not fine-tuned on voxel or noisy input data. This is potentially one reason why some reconstructed meshes do not have high quality.

### F.7.1 LION vs. Deep Matching Tetrahedra on Voxel-guided Synthesis

We additionally compare LION's performance on voxel-guided synthesis to Deep Marching Tetrahedra [41] (DMTet) on the airplane category (see Tab. 22). We train and evaluate DMTet with the same data as was used in our voxel-guided shape synthesis experiments (see Sec. 5.4 and Apps. C.2 and E.4). To compute the evaluation metrics on the DMTet output, we randomly sample points on DMTet's output meshes. LION achieves reconstruction results of similar or slighty better quality than DMTet. However, note that DMTet was specifically designed for such reconstruction tasks and is not a general generative model that could synthesize novel shapes from scratch without any guidance

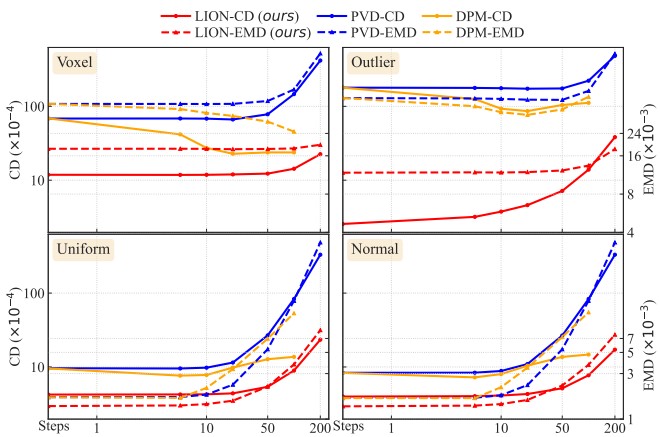

Figure 25: Reconstruction metrics with respect to clean inputs for *chair* category (lower is better) when guiding synthesis with voxelized or noisy inputs (using uniform, outlier, and normal noise, see App. F.7). *x*-axes denote number of *diffuse-denoise* steps.

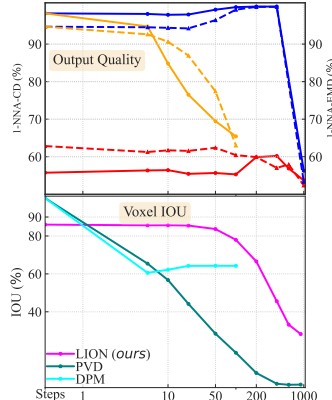

Figure 26: Voxel-guided generation for *chair* category. Quality metrics for output points (lower is better) and voxel IOU with respect to input (higher is better). *x*-axes denote *diffuse-denoise* steps.

| Method | CD↓ | EMD↓ | IOU↑ |
|--------|-----|------|------|
| DMTet [41] | 0.028 | 0.794 | 0.8148 |
| LION *(ours)* | 0.027 | 0.725 | 0.8177 |

Table 22: We compare the performance of LION to DMTet on the voxel-guided synthesis task for the airplane category. When calculating the reconstruction metrics, we do not perform any *diffuse-denoise*.

| Dataset | Metric | PointFlow [31] | ShapeGF [45] | DPM [47] | LION *(ours)* |
|---------|--------|----------------|--------------|----------|---------------|
| Airplane | CD | 0.012 | 0.010 | 0.035 | **0.003** |
|          | EMD | 0.511 | 0.426 | 1.121 | **0.012** |
| Car | CD | 0.065 | 0.053 | 0.083 | **0.006** |
|     | EMD | 1.271 | 0.902 | 1.796 | **0.009** |
| Chair | CD | 0.101 | 0.056 | 0.140 | **0.007** |
|       | EMD | 1.766 | 1.213 | 2.790 | **0.014** |

Table 23: Comparison of point cloud auto-encoding performance. Both CD and EMD reconstruction values are multiplied with $1 \times 10^{-2}$.

signal, unlike LION, which is a highly versatile general 3D generative model. Furthermore, as we demonstrated in the main paper, LION can generate multiple plausible de-voxelized shapes, while DMTet is fully deterministic and can only generate a single reconstruction.

### F.8 Autoencoding

We report the auto-encoding performance of LION and other baselines in Tab. 23 for single-class models. We are calculating the reconstruction performance of LION's VAE component. Additional results for the LION model trained on many classes can be found in Tab. 24. LION achieves much better reconstruction performance compared to all other baselines. The hierarchical latent space is expressive enough for the model to perform high quality reconstruction. At the same time, as we have shown above, LION also achieves state-of-the-art generation quality. Moreover, as shown in App. F.3.2, its shape latent space is also still semantically meaningful.

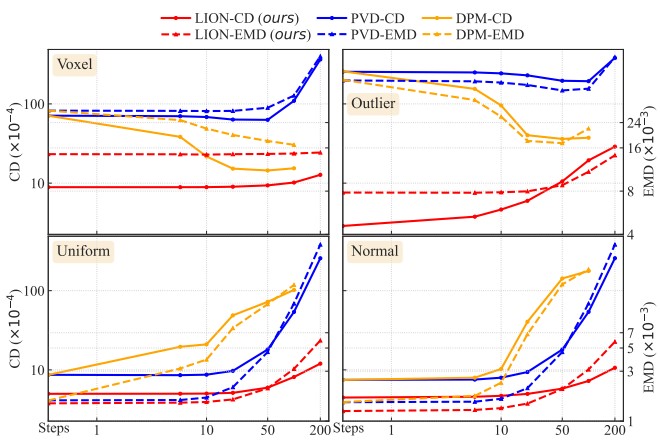

Figure 27: Reconstruction metrics with respect to clean inputs for *car* category (lower is better) when guiding synthesis with voxelized or noisy inputs (using uniform, outlier, and normal noise, see App. F.7). *x*-axes denote number of *diffuse-denoise* steps.

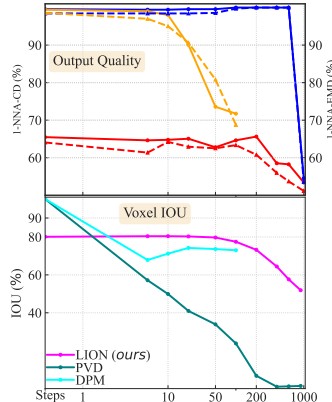

Figure 28: Voxel-guided generation for *car* category. Quality metrics for output points (lower is better) and voxel IOU with respect to input (higher is better). *x*-axes denote *diffuse-denoise* steps.

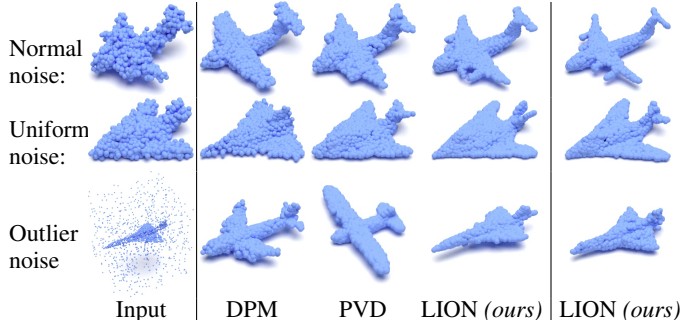

| | Input | DPM | PVD | LION *(ours)* | LION *(ours)* |
| --- | --- | --- | --- | --- | --- |
| Normal noise: | | | | | |
| Uniform noise: | | | | | |
| Outlier noise | | | | | |

Figure 29: Results on denoising experiments. For our LION model, we show two different possible output variations. The right hand side results for our LION model are created by injecting diversity with 200 (first row) or 100 (second and third rows) *diffuse-denoise* steps in latent space.

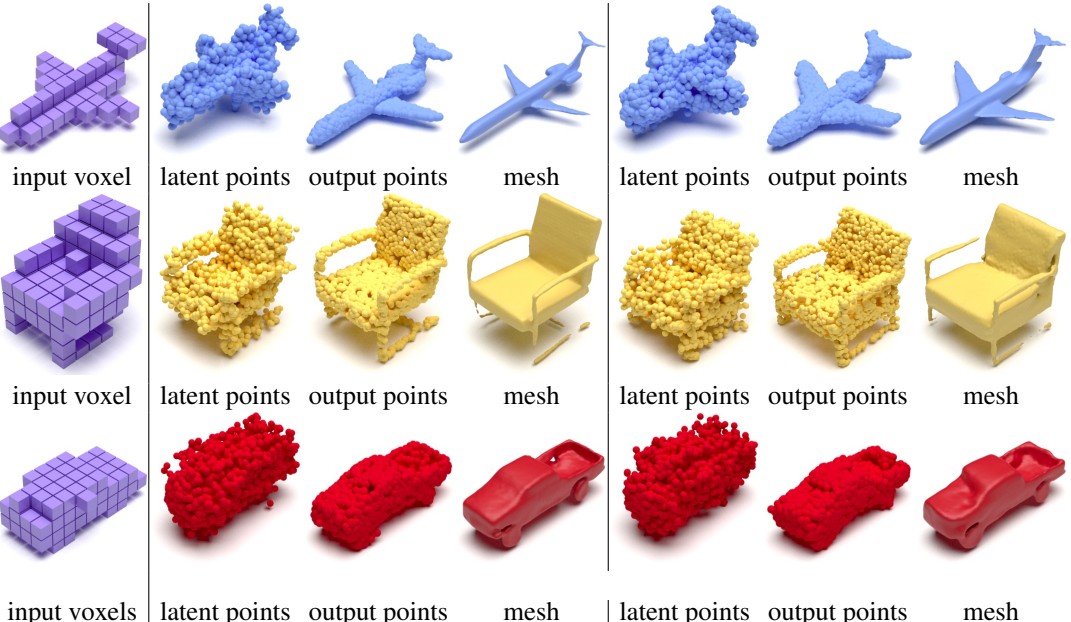

| input voxel | latent points | output points | mesh | latent points | output points | mesh |
| --- | --- | --- | --- | --- | --- | --- |
| input voxel | latent points | output points | mesh | latent points | output points | mesh |
| input voxels | latent points | output points | mesh | latent points | output points | mesh |

Figure 30: Voxel-guided synthesis experiments, on different categories. We run *diffuse-denoise* in latent space to generate diverse plausible clean shapes (*first row, left plane*: 250 diffuse-denoise steps; *first row, right plane*: 200 steps; *second row, left chair*: 200 steps; *second row, right chair*: 200 steps; *third row, left car*: 0 steps; *third row, right car*: 0 steps). Note that even when running no diffuse-denoise, we still obtain slightly different outputs due to different approximate posterior samples from the encoder network.

## F.9 Synthesis Time and DDIM Sampling

Our main results in the paper are all generated using standard 1,000-step DDPM-based ancestral sampling (see Sec. 2). Generating a point cloud sample (with 2,048 points) from LION takes $\approx 27.12$ seconds, where $\approx 4.04$ seconds are used in the shape latent diffusion model and $\approx 23.05$ seconds

| Method | CD | EMD |
| --- | --- | --- |
| DPM [47] | 1.477 | 5.722 |
| LION *(ours)* | **0.004** | **0.009** |

Table 24: Comparison of point cloud auto-encoding performance, for models trained on the many-class dataset. Both CD and EMD reconstruction values are multiplied with $1 \times 10^{-2}$.

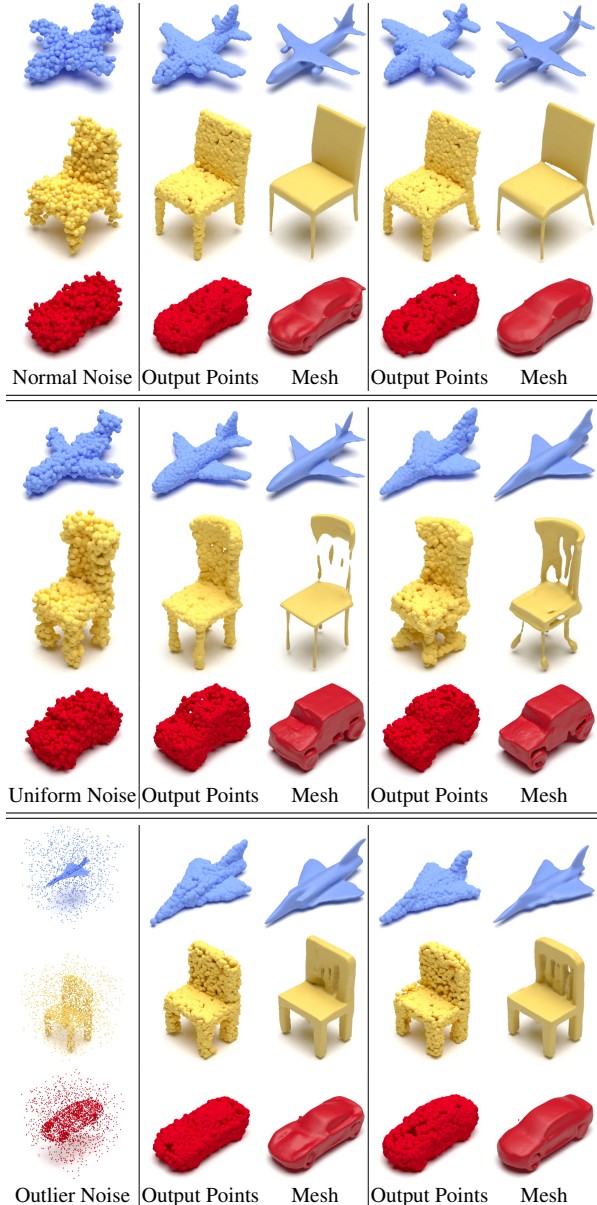

Figure 31: Denoising experiments, on different categories. We run *diffuse-denoise* in latent space with varying numbers of steps to generate diverse plausible clean shapes.

in the latent points diffusion model. Optionally running SAP for mesh reconstruction requires an additional $\approx 2.57$ seconds.

A simple and popular way to accelerate sampling in diffusion models is based on the *Denoising Diffusion Implicit Models*-framework (DDIM). We show DDIM-sampled shapes in Fig. 38 and the generation performance in Tab. 25. For all DDIM sampling, we use $\eta = 0.5$ as stochasticity hyperparameter and the quadratic time schedule as also proposed in DDIM [106]. We also tried deterministic DDIM sampling, but it performed worse (for 50-step sampling). We find that we can produce good-looking shapes in under one second with only 25 synthesis steps. Performance significantly degrades when using $\leq 10$ steps.

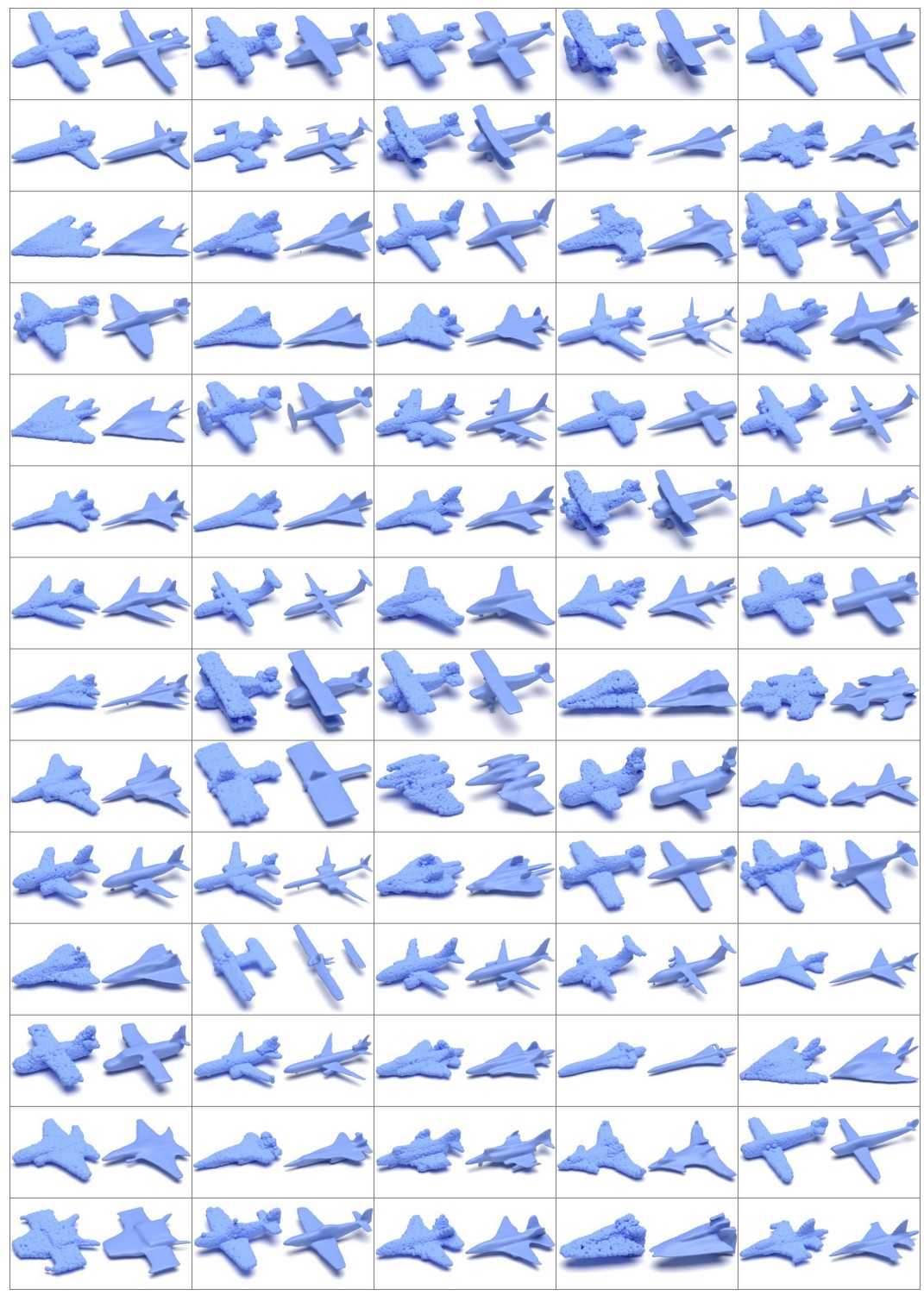

Figure 32: Generated shapes from the airplane class LION model.

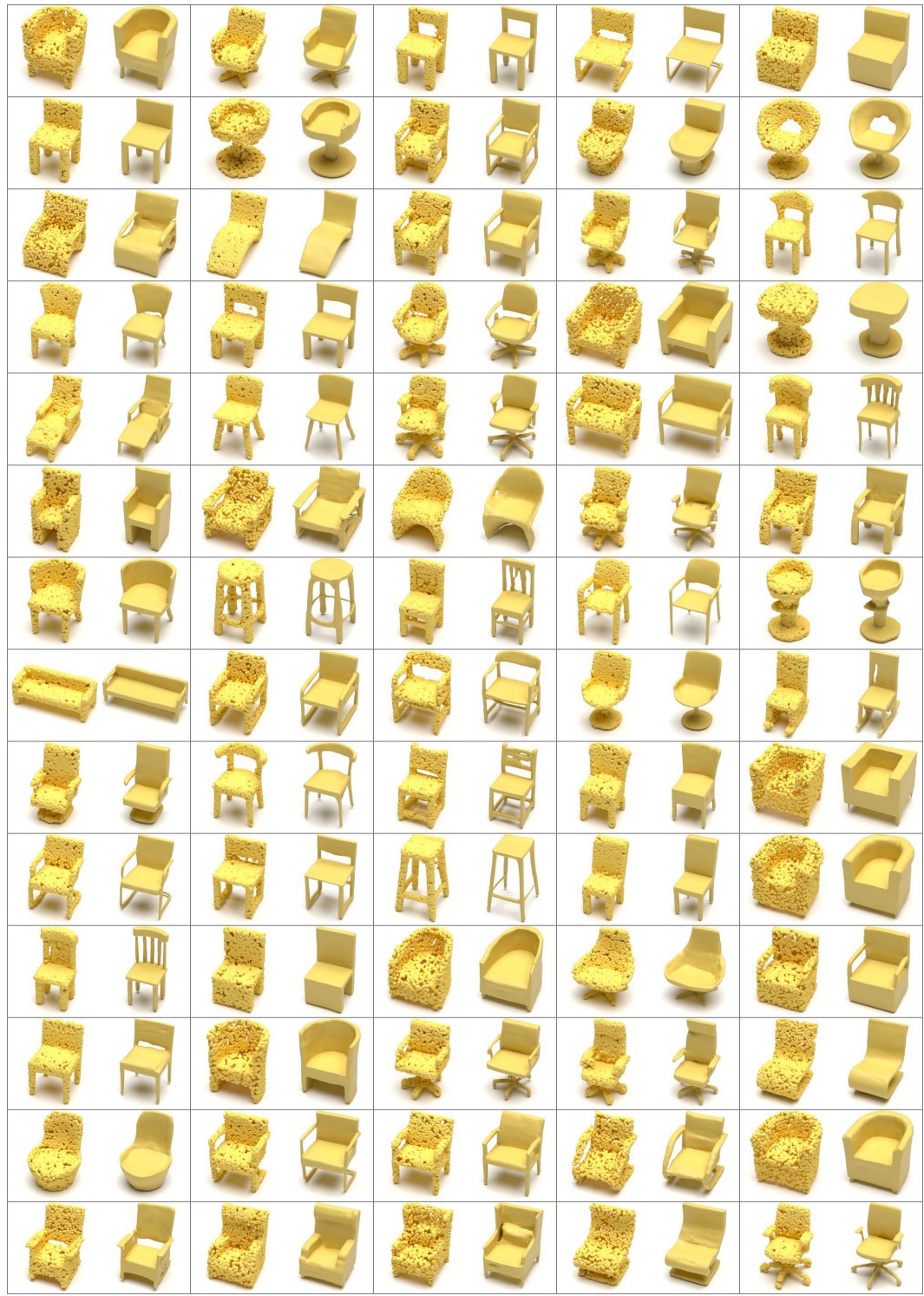

Figure 33: Generated shapes from the chair class LION model.

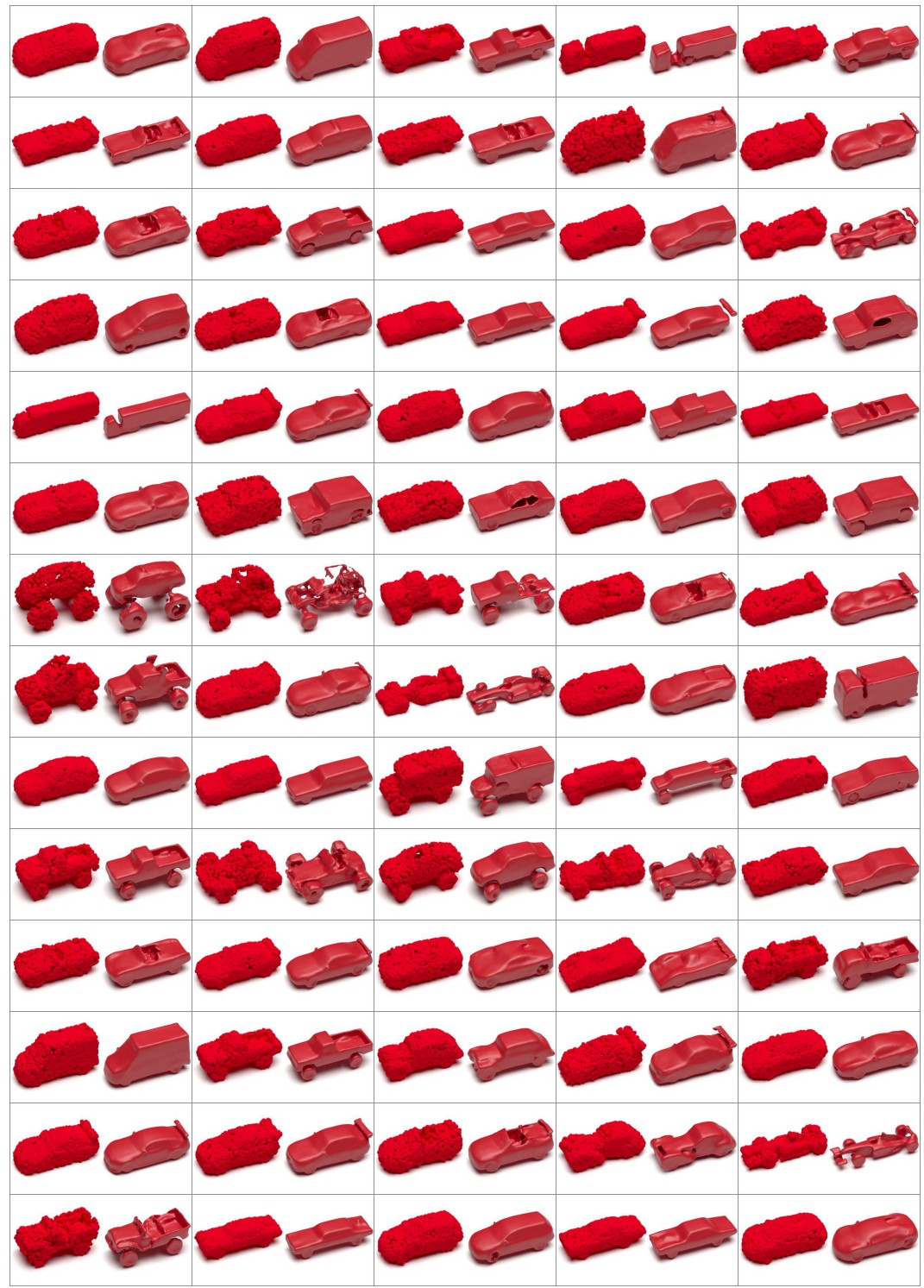

Figure 34: Generated shapes from the car class LION model.

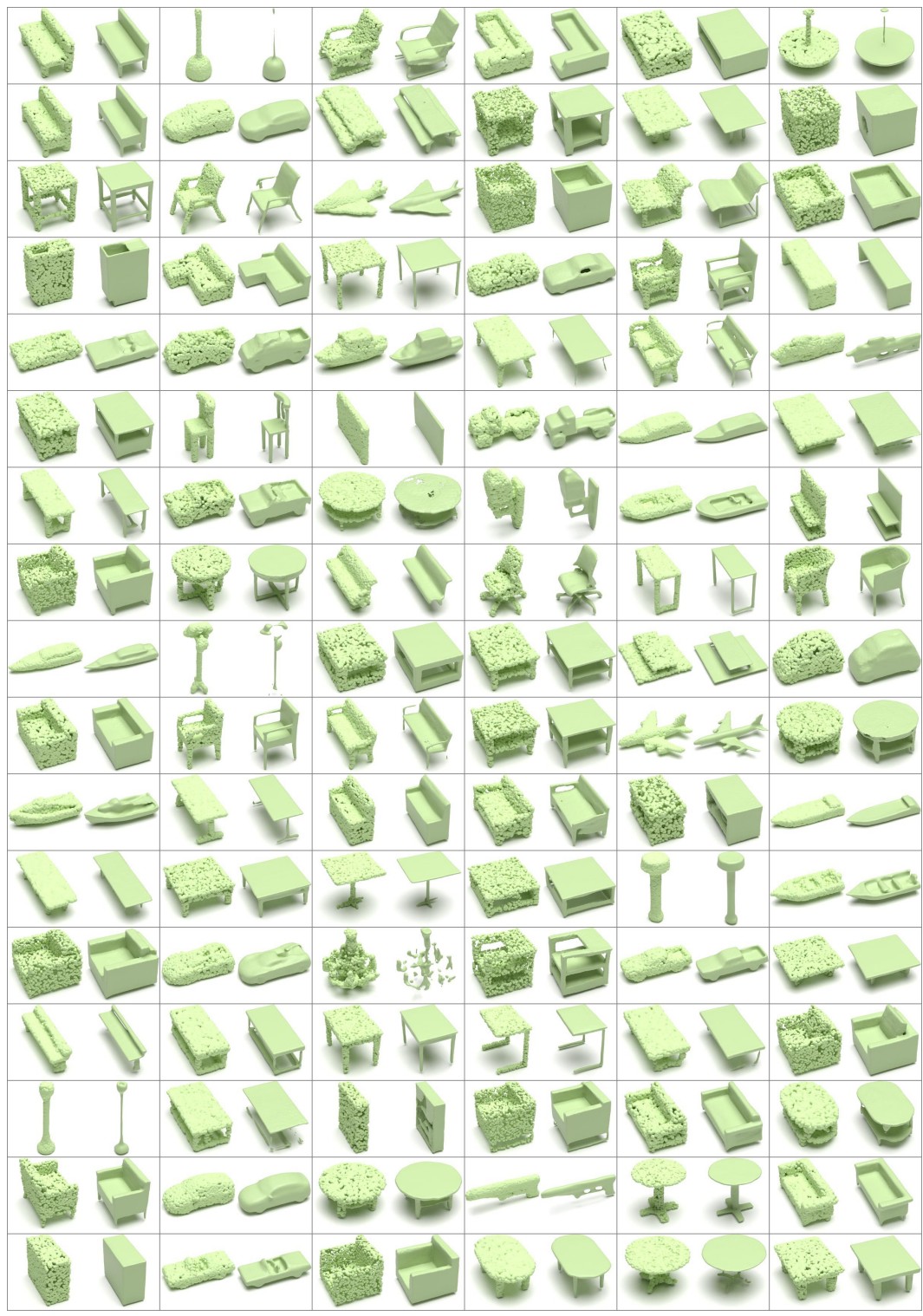

Figure 35: Generated shapes from the many-class LION model.

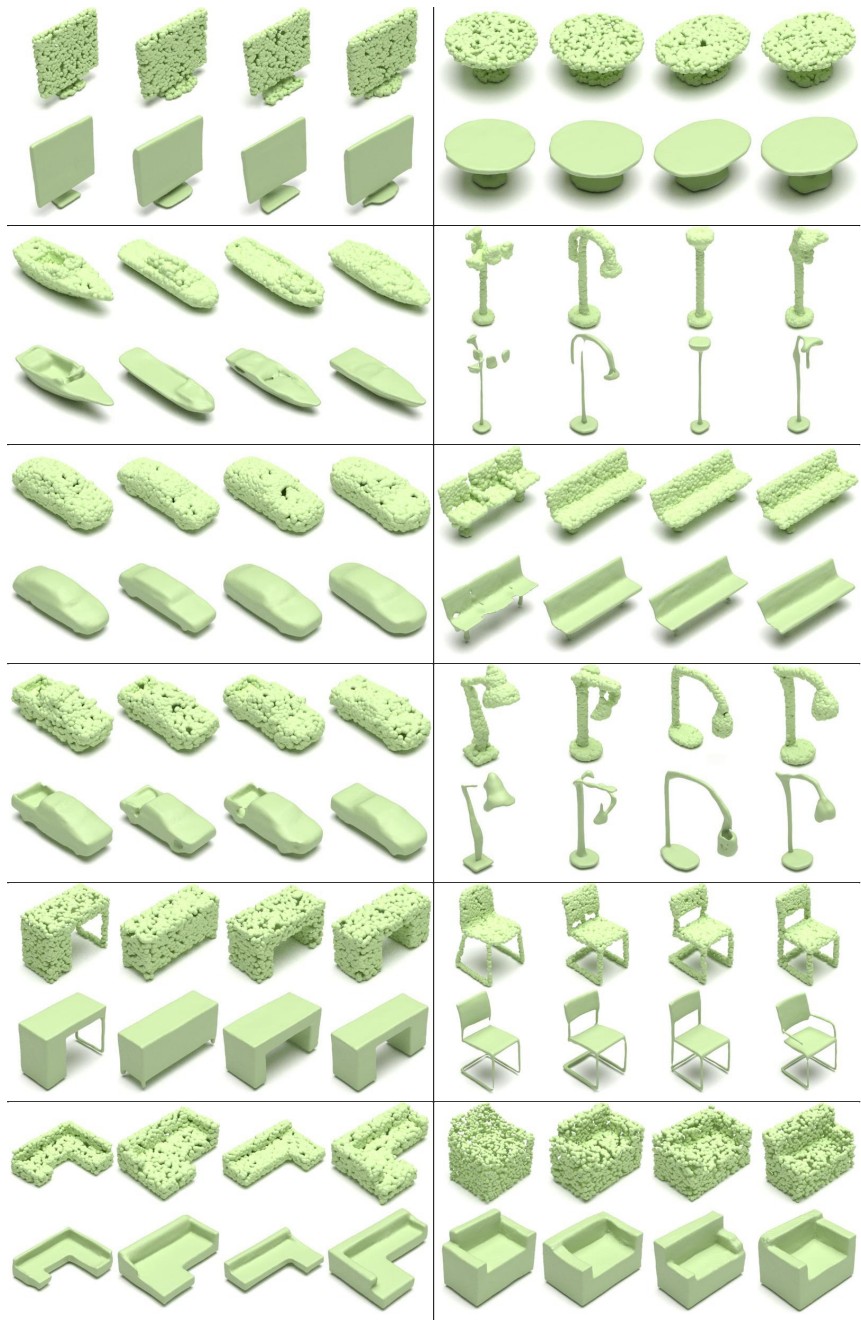

Figure 36: For each group of shapes, we freeze the shape latent variables, and sample different latent points to generate the other shapes.

## F.10    Per-sample Text-driven Texture Synthesis

To demonstrate the value to artists of being able to synthesize meshes and not just point clouds, we consider a downstream application: We apply Text2Mesh[7] [49] on some generated meshes from LION to additionally synthesize textures in a text-driven manner, leveraging CLIP [105]. Optionally, Text2Mesh can also locally refine the mesh and displace vertices for enhanced visual effects. See Fig. 39 for results where we show different objects made of *snow* and *potato chips*, respectively. In Fig. 40, we apply different text prompts on the same generated airplane. We show more diverse

---
[7] https://github.com/threedle/text2mesh

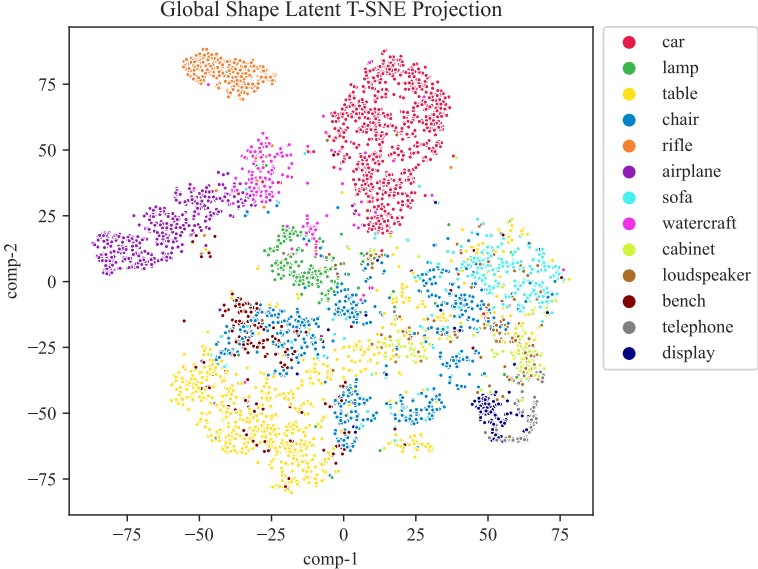

Figure 37: Shape latent t-SNE plot for the many-class LION model.

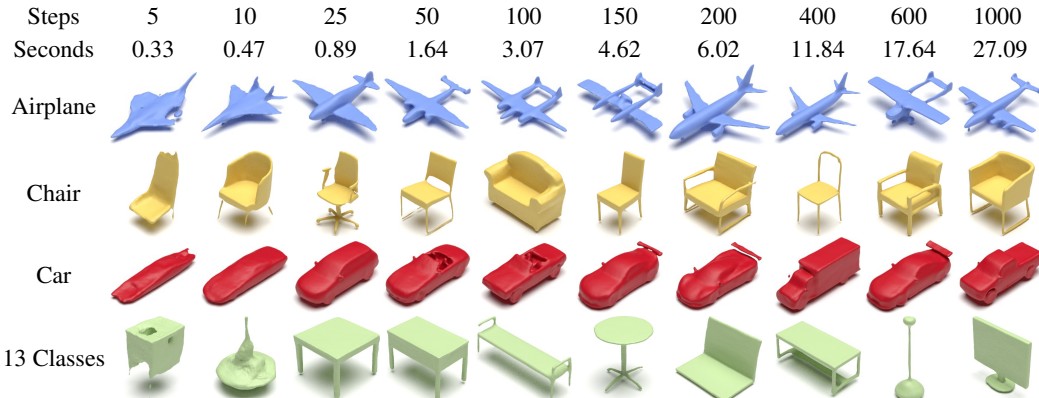

| Steps | 5 | 10 | 25 | 50 | 100 | 150 | 200 | 400 | 600 | 1000 |
|---|---|---|---|---|---|---|---|---|---|---|
| Seconds | 0.33 | 0.47 | 0.89 | 1.64 | 3.07 | 4.62 | 6.02 | 11.84 | 17.64 | 27.09 |

Figure 38: DDIM samples from LION trained on different data. The top two rows show the number of steps and the wall-clock time required for drawing one sample. In general, DDIM sampling with 5 steps fails to generate reasonable shapes and DDIM sampling with more than 10 steps can generate high-quality shapes. With DDIM sampling, we can reduce the time to generate an object from 27.09 seconds (1,000 steps) to less than 1 second (25 steps). The sampling time is computed by calling the prior model and the decoder of LION with batch size as 1, number of points as 2,048.

results on other categories in Fig. 41. Note that this is only possible because of our SAP-based mesh reconstruction.

## F.11    Single View Reconstruction and Text-driven Shape Synthesis

Although our main goal in this work was to develop a strong generative model of 3D shapes, here we qualitatively show how to extend LION to also allow for single view reconstruction (SVR) from RGB data. We render 2D images from the 3D ShapeNet shapes, extracted the images' CLIP [105] image embeddings, and trained LION's latent diffusion models while conditioning on the shapes' CLIP image embeddings. At test time, we then take a single view 2D image, extract the CLIP image embedding, and generate corresponding 3D shapes, thereby effectively performing SVR. We show SVR results from real RGB data in Fig. 42, Fig. 45 and Fig. 46. The RGB images of the chairs are

| DDIM-Steps | time(sec) | MMD↓ | | COV↑ (%) | | 1-NNA↓ (%) | |
|---|---|---|---|---|---|---|---|
| | | CD | EMD | CD | EMD | CD | EMD |
| 5 | 0.33 | 10.117 | 4.6869 | 12.1 | 5.0 | 99.00 | 99.35 |
| 10 | 0.47 | 5.0285 | 2.7719 | 23.6 | 13.1 | 93.25 | 96.50 |
| 25 | 0.89 | 2.9425 | 1.6577 | 43.4 | 39.0 | 70.35 | 73.55 |
| 50 | 1.64 | 2.5965 | 1.5051 | 48.7 | 48.3 | 58.20 | 59.30 |
| 100 | 3.07 | 2.5929 | 1.4853 | 50.7 | 51.8 | 53.85 | 52.45 |
| 150 | 4.62 | 2.4724 | 1.4570 | 49.5 | 52.6 | 53.40 | 51.15 |
| 200 | 6.02 | **2.4321** | 1.4683 | 52.3 | 52.1 | 52.00 | 51.20 |
| 400 | 11.84 | 2.5625 | 1.4956 | 50.6 | **54.4** | 53.95 | 52.85 |
| 600 | 17.64 | 2.4353 | **1.4471** | 52.4 | 51.8 | 52.65 | 50.95 |
| 1,000 | 27.09 | 2.4724 | 1.4841 | 51.2 | 52.1 | 52.05 | 53.20 |
| DDPM-1,000 | 27.09 | 2.4572 | 1.4472 | **54.3** | **54.4** | **51.85** | **48.95** |

Table 25: Generation performance metrics for our LION model that was trained jointly on all 13 ShapeNet classes without class-conditioning. The performance with different numbers of steps used in the DDIM sampler is reported, as well as the 1,000-step DDPM sampling that was used in the main experiments. We also report the required wall-clock time for the sampling when synthesizing one shape. The results are consistent with our visualization (Fig. 38): The DDIM sampler with 5 steps fails, and with more than 10 steps, it starts to generate good shapes.

*a __ made of snow:*

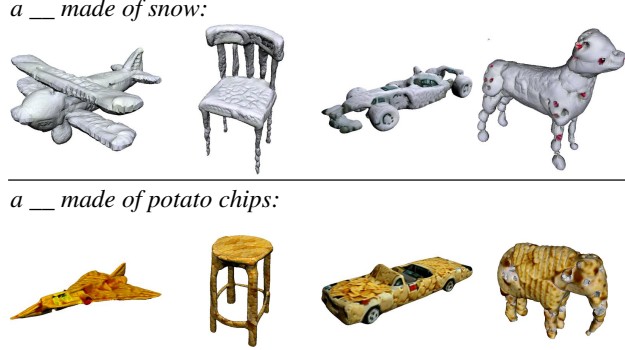

*a __ made of potato chips:*

Figure 39: Text2Mesh results of *airplane, chair, car, animal*. The original mesh is generated by LION.

from Pix3D[8] [128] and Redwood 3DScan dataset[9] [127], respectively. The RGB images of the cars are from the Cars dataset[10] [126]. For each input image, LION is able to generate different feasible

[8]We downloaded the data from https://github.com/xingyuansun/pix3d. The Pix3D dataset is licensed under a Creative Commons Attribution 4.0 International License.

[9]We downloaded the Redwood 3DScan dataset (public domain) from https://github.com/isl-org/redwood-3dscan.

[10]We downloaded the Cars dataset from http://ai.stanford.edu/~jkrause/cars/car_dataset.html. The Cars dataset is licensed under the ImageNet License: https://image-net.org/download.php

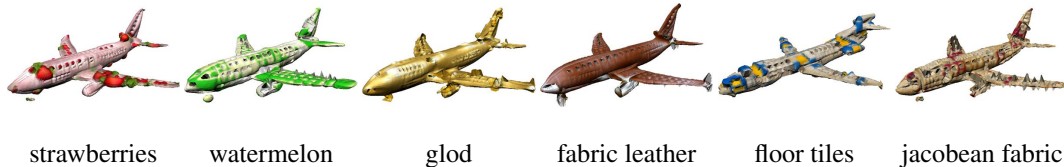

| strawberries | watermelon | glod | fabric leather | floor tiles | jacobean fabric |

Figure 40: Text2Mesh results with text prompt *an airplane made of __*. All prompts are applied on the same generated shapes. The original mesh is generated by LION.

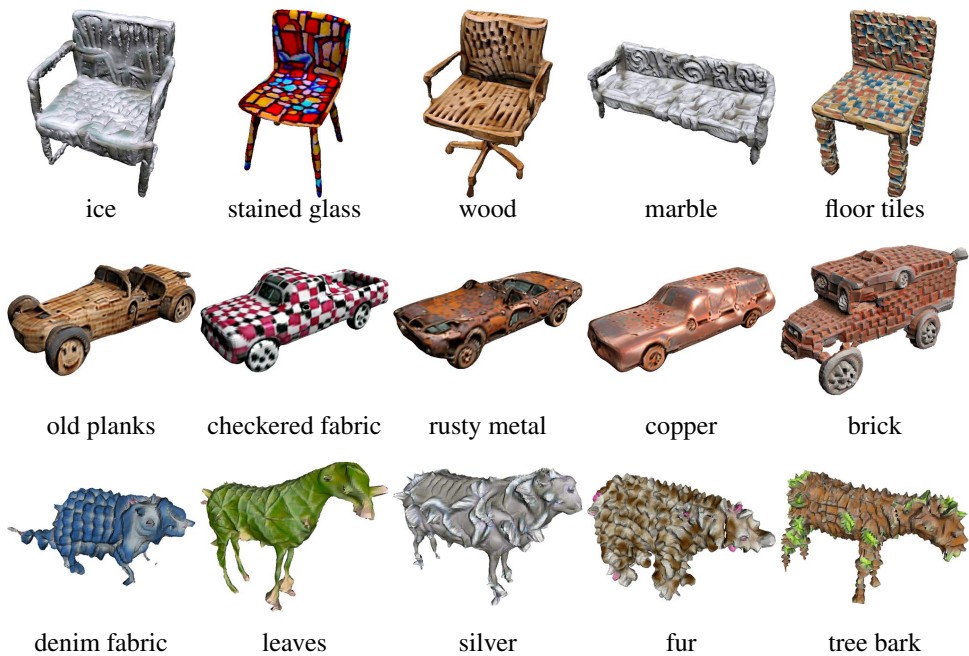

Figure 41: Text2Mesh results with text prompt: *a chair made of __* (top row), *a car made of __* (middle row) and *an __ animal* (bottom row). Each prompt is applied on different generated shapes. The original mesh is generated by LION.

shapes, showing LION's ability to perform multi-modal generation. Qualitatively, our results appear to be of similar quality as the results of PVD [46] for that task, and at least as good or better than the results of AutoSDF [30]. Note that this approach only requires RGB images. In contrast, PVD requires RGB-D images, including depth. Hence, our approach can be considered more flexible. Using CLIP's text encoder, our method additionally allows for text-guided generation as demonstrated in Fig. 43 and Fig. 44. Overall, this approach is inspired by CLIP-Forge [34]. Note that this is a simple qualitative demonstration of LION's extendibility. We did not perform any hyperparameter tuning here and believe that these results could be improved with more careful tuning and training.

## F.12 More Shape Interpolations

We show more shape interpolation results for single-class LION models in Figs. 47, 48, 49, and the many-class LION model in Figs. 50, 51. We can see that LION is able to interpolate two shapes from different classes smoothly. For example, when it tries to interpolate a chair and a table, it starts to make the chair wider and wider, and gradually removes the back of the chair. When it tries to interpolate an airplane and a chair, it starts with making the wings more chair-like, and reduces the size of the rest of the body. The shapes in the middle of the interpolation provide a smooth and reasonable transition.

### F.12.1 Shape Interpolation with PVD and DPM

To be able to better judge the performance of LION's shape interpolation results, we now also show shape interpolations with PVD [46] and DPM [47] in Fig. 52 and Fig. 53, respectively. We apply the *spherical interpolation* (see Sec. C.3) on the noise inputs for both PVD and DPM. DPM leverages a Normalizing Flow, which already offers deterministic generation given the noise inputs of the Flow's normal prior. For PVD, just like for LION, we again use the diffusion model's ODE formulation to obtain deterministic generation paths. In other words, to avoid confusion, in both cases we are interpolating in the normal prior distribution, just like for LION.

Although PVD is also able to interpolate two shapes, the transition from the source shapes to the target shapes appear less smooth than for LION; see, for example, the chair interpolation results of PVD. Furthermore, DPM's generated shape interpolations appear fairly noisy. When interpolating

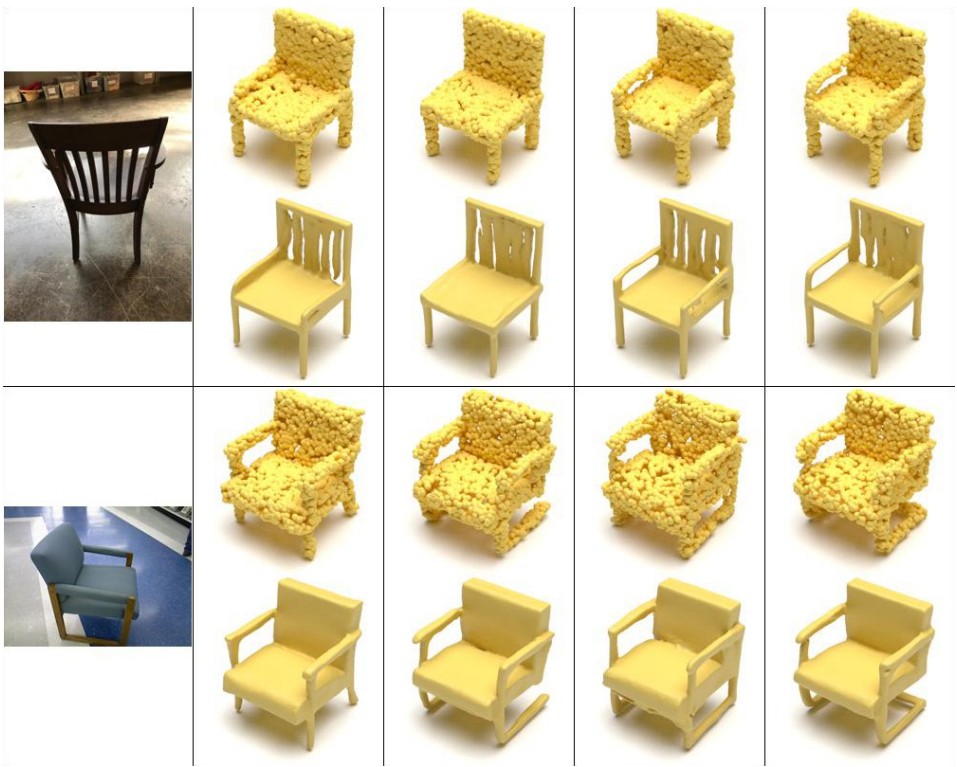

Figure 42: Single view reconstructions of chairs from RGB images. For each input image, LION can generate multiple plausible outputs.

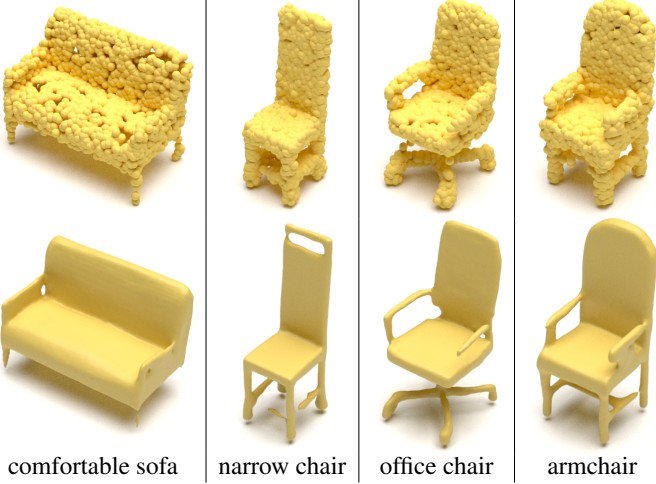

| comfortable sofa | narrow chair | office chair | armchair |

Figure 43: Text-driven shape generation of chairs with LION. Bottom row is the text input.

very different shapes using the 13-classes models, both PVD and DPM essentially break down and do not produce sensible outputs anymore. All shapes along the interpolation paths appear noisy.

In contrast, LION generally produces coherent interpolations, even when using the multimodal model that was trained on 13 ShapeNet classes (see Figs. 47, 48, 49 and 50 for LION interpolations for reference).

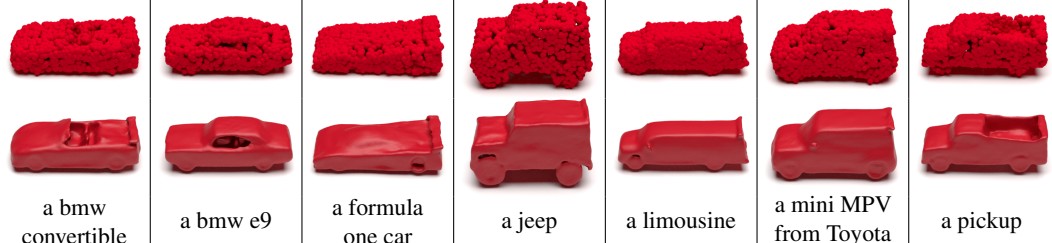

Figure 44: Text-driven shape generation of cars with LION. Bottom row is the text input.

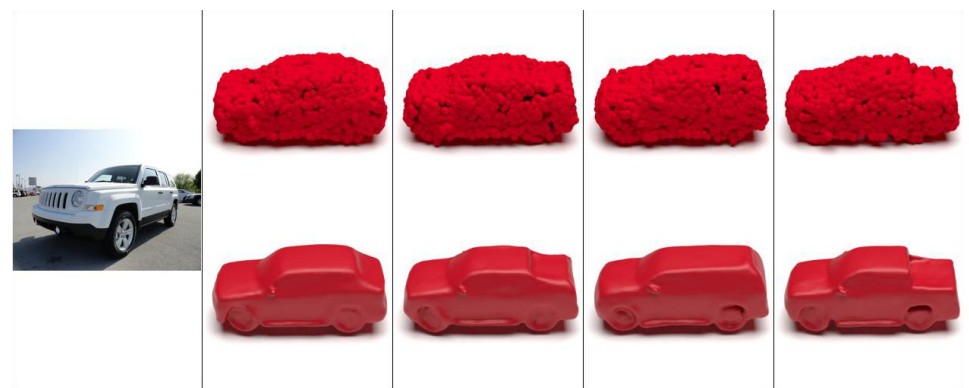

Figure 45: Single view reconstructions of cars from RGB images. For each input image, LION can generate multiple plausible outputs.

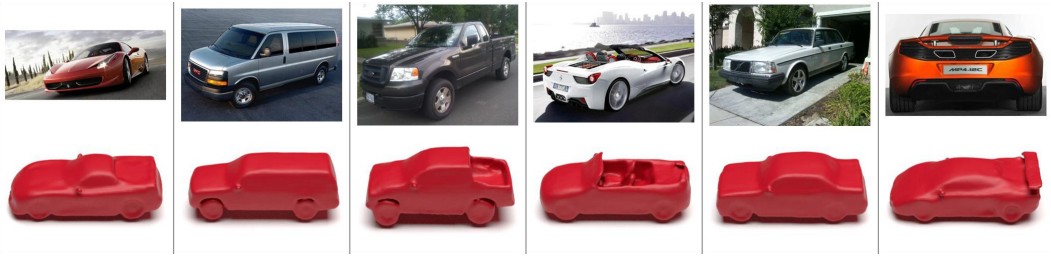

Figure 46: More single view reconstructions of cars from RGB images.

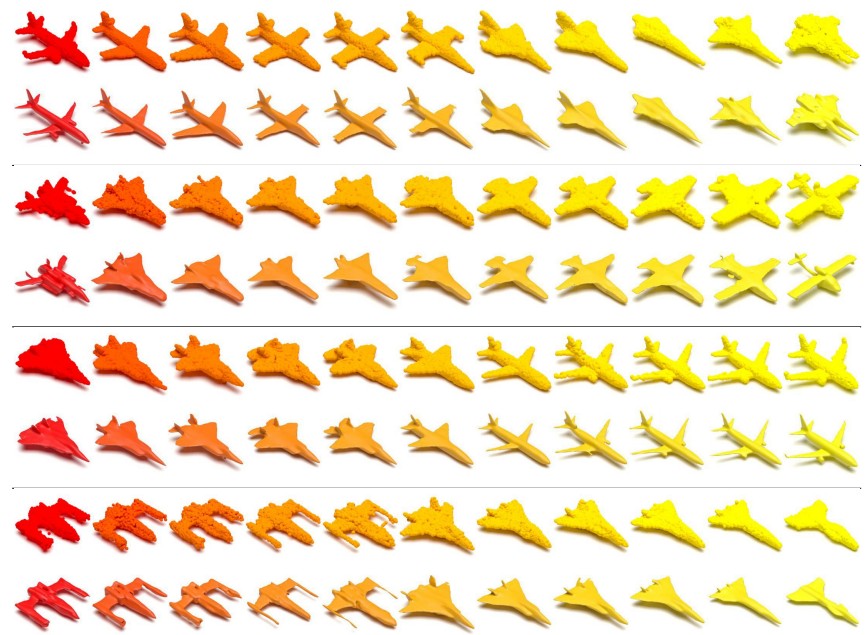

Figure 47: More interpolation results from LION models trained on the airplane class.

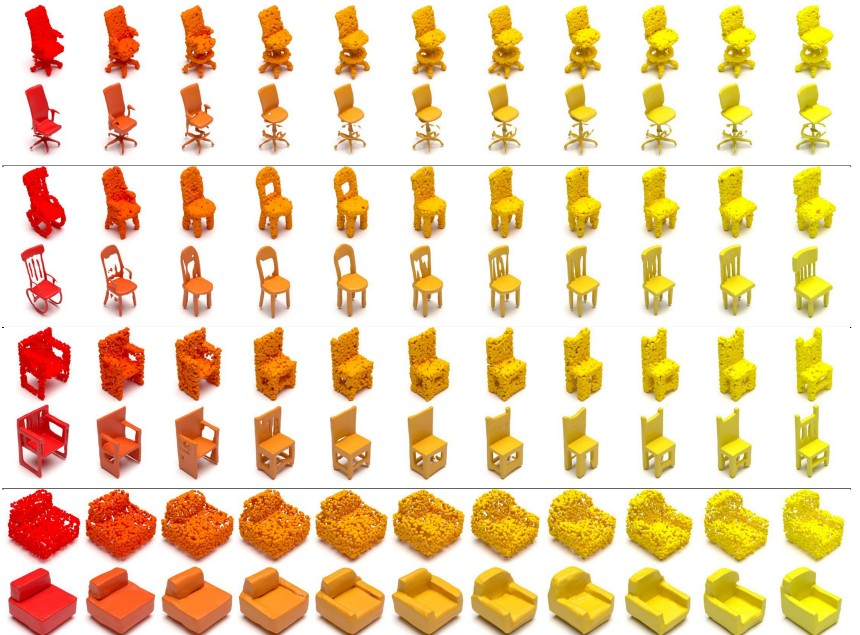

Figure 48: More interpolation results from LION models trained on the chair class.

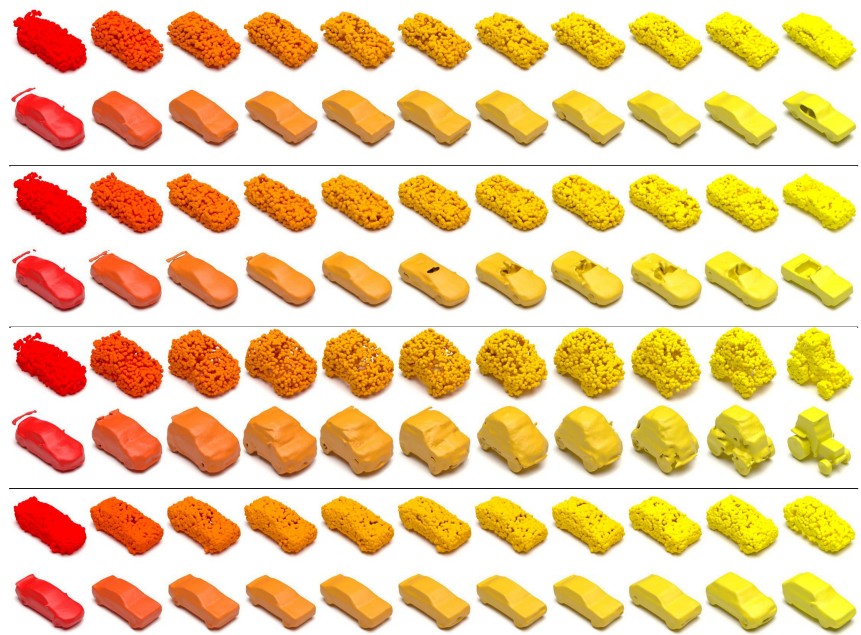

Figure 49: More interpolation results from LION models trained on the car class.

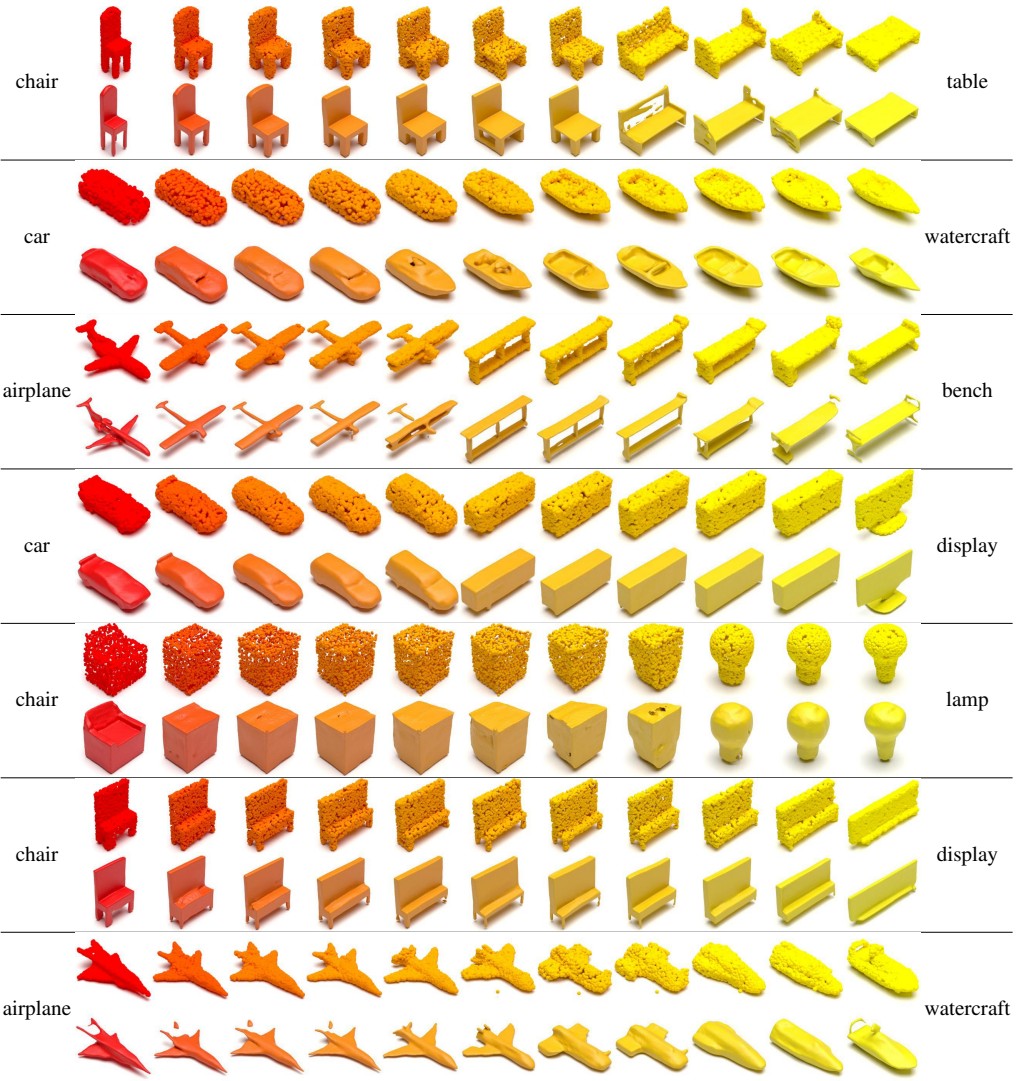

Figure 50: More interpolation results from LION trained on many classes.

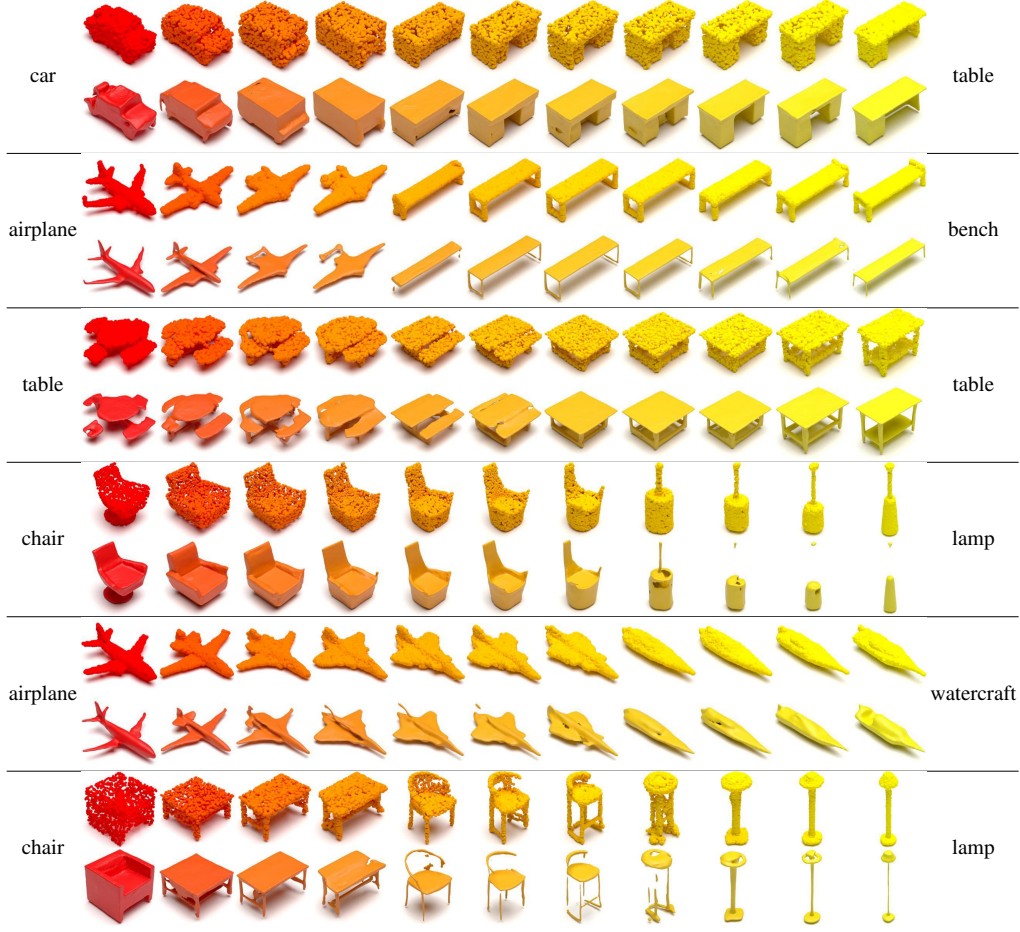

Figure 51: More interpolation results from LION trained on many classes.

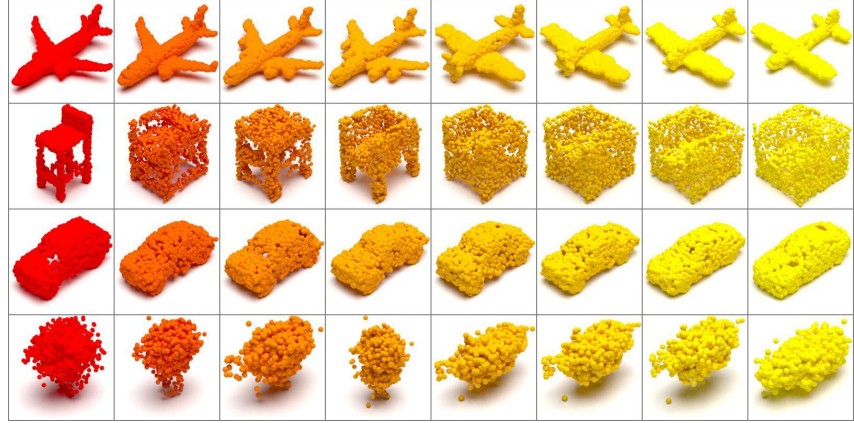

Figure 52: Shape interpolation results of PVD [46]. From top to bottom: PVD trained on airplane, chair, car, 13 classes of ShapeNet.

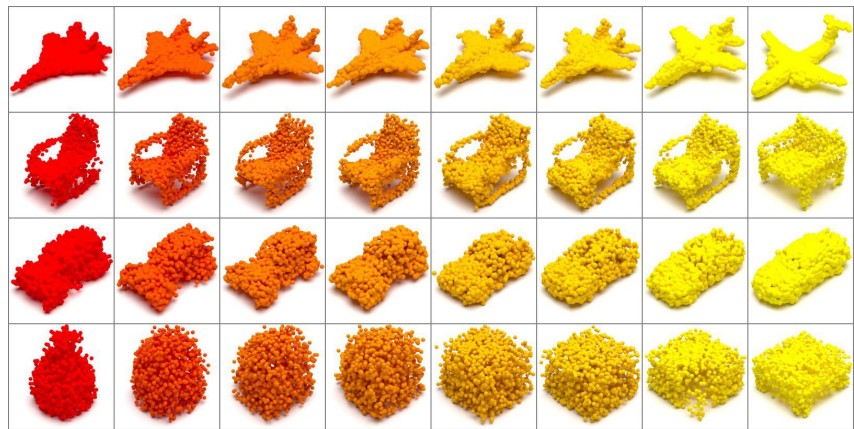

Figure 53: Shape interpolation results of DPM [47]. From top to bottom: DPM trained on airplane, chair, car, 13 classes of ShapeNet.