# OpenReview forum: "LION: Latent Point Diffusion Models for 3D Shape Generation"
_NeurIPS.cc/2022/Conference — NeurIPS 2022 Accept_

### Official Review · Reviewer_FSJk · 2022-07-10

**Rating:** 7
**Confidence:** 4
**Soundness:** 3 good
**Presentation:** 4 excellent
**Contribution:** 3 good

**Summary:**

Authors present a generative model of 3D point clouds. This is a latent-space diffusion model, with variational encoder and decoder mapping between point cloud and latent representation; the latent space itself is split into a 'global' shape embedding, and 'local' embeddings of details, the latter itself structured as a sparser, feature-valued point-cloud. The model is first trained as a 'simple' VAE with gaussian prior/posterior, and then the diffusion model is fitted post-hoc to the aggregate posterior. The model is demonstrated on various tasks using ShapeNet data: unconditional generation; sampling 'variations' on a coarse model; interpolating shapes; and sampling meshes by integration with a differentiable Poisson solver. It achieves qualitative-reasonable results on all tasks, and quantitatively out-performs various point-cloud-based baselines on generation tasks.

**Questions:**

Exactly how computationally expensive is the model, e.g. to generate a single chair or train all parts of the corresponding model? This doesn't seem to be specified anywhere, only the total compute for the entire project.

Discussing only discrete-time diffusion in the paper (sec. 2) in spite of the model using exclusively continuous-time (for which full details are only in the supplementary) seems a little strange. It might be better to pull in a couple of paragraphs from the supplementary into the main text.


**Limitations:**

There is very minimal discussion of limitations; this should be expanded. There is sufficient discussion of broader impact (which is negligible).

**Strengths And Weaknesses:**

Strengths:

- the model is novel, and the approach sensibly motivated

- quantitative results show significantly better performance than state-of-the-art baselines on unconditional (per-class) generation tasks

- qualitative results are impressive – particularly the displayed meshes

- there are very helpful and detailed descriptions of components (e.g. how interpolation is performed) in the supplementary; this mitigates the fact that the overall model is rather complex

- there is an ablation study in the supplementary that provides some additional motivation for various design decisions

- the writing is clear and easy-to-read throughout; overall I found the paper enjoyable and informative to read

Weaknesses:

- the technical contribution of the model itself is fairly small – DDMs on latent space have become fairly common, and even DDMs on point-clouds is not a new idea (though authors show that the proposed method outperforms other such approaches)

- most experiments use only three shapenet classes; it would be nice to see more diversity, or discussion of why this is not possible (e.g. if there are too few shapes in some classes to train per-class models successfully -- this should be stated as a limitation of the method if so)

- the mapping from point-clouds to meshes is responsible for the best-looking results in the paper (the meshes are significantly cleaner than the point-clouds); however this is apparently trained as a post-hoc stage rather than jointly with the generative model. It would be more elegant to train this end-to-end (the same could also be said for training the diffusion jointly with the encoder/decoder, however I appreciate that this is challenging)

---

> ### Author Response · Authors · 2022-08-02
> **Thank You for the Feedback (1)**
>
> We would like to thank the reviewer for their positive feedback and for appreciating our novel and sensibly-motivated method as well as both our qualitative and quantitative results. We are also glad that our writing came across as clear and that our detailed appendix is appreciated. We are happy the reviewer enjoyed reading our manuscript.
>
> Below, we reply to the individual points and questions raised by the reviewer (citations correspond to paper bibliography):
> - **Technical Contributions:** It is correct that models with DDMs in latent space exist in the image-modeling literature, and that DDMs have generally been used for point cloud generation before. However, models with an architecture like ours with two complementary DDMs that both operate in latent space have not been used before, neither for images nor 3D shapes. Furthermore, the idea of latent DDMs has never been used in 3D generation at all, to the best of our knowledge, and it is generally not obvious how to extend it to 3D in the point cloud case and how to structure the latent space. To this end, we introduce the concept of latent points and combine it with a complementary vector-valued latent variable. Hence, we believe our architecture is novel and the experimental results clearly support that our design choices are crucial: We outperform all baselines, achieve state-of-the-art results, and demonstrate that with this architecture we can scale to extremely diverse shape datasets, like modeling 13 or even 55 (see below) ShapeNet categories jointly without conditioning. These cases represent highly complex and multimodal distributions that need to be learnt, thereby stress-testing LION’s scalability. In particular on these diverse tasks we outperform previous DDM-based models for point clouds [46,47] by huge margins (Table 16). This implies the importance and clear superiority of our novel design, rendering it a relevant technical contribution.
>
>     Furthermore, even though we did not train Shape As Points (SAP) end-to-end with LION, we are still fine-tuning SAP on LION. In fact, another technical contribution of our work is to demonstrate how to best combine point cloud generative models with a mesh reconstruction method like Shape As Points (SAP). In particular, to be more robust against the specific kind of noise that is present during synthesis from LION we show how to fine-tune SAP on point clouds augmented with our proposed *diffuse-denoise* technique (this fine-tuning makes a significant difference; see newly added Fig. 14). Overall, we think that demonstrating how to combine SAP with LION is in fact a novel and relevant contribution, and this has never been done before for any point cloud generation method, to the best of our knowledge. Due to the simplicity and relevance (digital artists use meshes in practice, after all), we believe that this idea will be adopted by future works.
>
> - **Diversity in Experiments:** Our main experiments are run on three ShapeNet classes, simply because the majority of previous work focuses on these three individual classes and we wanted to compare to a broad set of previous methods. However, we also run experiments where we jointly model 13 ShapeNet classes without conditioning, which represents a challenging and diverse distribution (see Sec. 5.2). We did this to demonstrate LION’s scalability to much harder and more diverse shape datasets. Hence, we did on purpose not use class conditioning, to make the task difficult. We now added several more experiments: (i) One experiment where we even jointly model all 55 ShapeNet classes, again without class-conditioning (see Appendix F.2, samples in Fig. 33). We see that LION can generate good output samples even when trained on such complex and diverse data. To the best of our knowledge, no previous 3D generative model has demonstrated such scalability. Importantly, the model also still generates data from very rare categories, like the cap class, which contributed with only 39 training samples. This validates that LION does not drop modes and faithfully learns the data distribution (ii) Next, we train on 2 individual ShapeNet classes that contain only very few samples as training data. We train on the mug (149 train samples) and bottle (340 train samples) classes, and can still generate good outputs (see Appendix F.3). (iii) We also used 400 animal assets from the TurboSquid (https://www.turbosquid.com/) dataset and trained a model on those. We again can generate satisfactory outputs (Appendix F.4). These additional experiments validate that LION can be trained both on highly complex diverse data distributions and also on small-scale data.

---

> > ### Author Response · Authors · 2022-08-02
> > **Thank You for the Feedback (2)**
> >
> > - **No End-to-End Training:** We agree with the reviewer that it would be very interesting to explore end-to-end training of the entire model, potentially including both the two latent space diffusion models, and also the SAP shape reconstruction method. As previous work (LSGM) [58] has noted, however, such end-to-end training comes with additional challenges. In LSGM, one of the main reasons for their end-to-end training is improved performance. However, we already obtain state-of-the-art performance even with separate training stages. Hence, we do not consider this aspect of our model a limitation or weakness. Training the different components separately is simpler from the user perspective, and is less memory- and compute-expensive, which will hopefully encourage adoption of our method. That said, we agree with the reviewer. It would be interesting to explore this in future work and potentially improve performance even further. Similarly, it would also be interesting to explore training end-to-end with SAP. Note, though, that we are already fine-tuning SAP on LION, thereby achieving a tight coupling between the two models, even though they are trained one after another. We will add a brief discussion on this in the final version of the paper.
> > - **Question about how computationally expensive the model is:** Generating a point cloud sample (with 2048 points) from LION takes 27.12 sec., where 4.04 sec. are used in the shape latent diffusion model and 23.05 sec. in the latent points diffusion model prior. Optionally running SAP for mesh reconstruction requires an additional 2.57 sec. For single-class LION models, the total training time is 550 GPU hours (110 GPU hours for training the backbone VAE; 440 GPU hours for training the two latent diffusion models). We added the numbers in Appendix F.10.
> > - **Question about discussing discrete-time diffusion formalism:** We believe that there is a misunderstanding: In fact, in all our experiments, we use the discrete-time formalism both during training and sampling (for instance, in our training objectives, we sample from discrete uniform distributions U{0,T} over the steps, as indicated). There is only a single qualitative experiment, where we convert the model into a continuous time model, this is, when we perform shape interpolation, where we require deterministic generation paths based on the probability flow ODE. Other than that our paper entirely relies on the discrete-time formalism, which is why we focused on that in our background section. If the reviewer would be able to point out what specifically led to this misunderstanding, we will be happy to improve our presentation accordingly.
> > - **Limitations:** We are happy to more extensively discuss our method’s limitations. For now, we added a discussion in Appendix F.11. In the final paper, when there is an additional page available for the main text, we may bring this into the main paper. In summary, additional limitations of LION beyond slow sampling are that it cannot directly generate textured shapes out of the box. Furthermore, it relies purely on geometry-based training and currently cannot profit from image-based training with differentiable rendering. Also, LION currently focuses on single object generation only. It would be interesting to extend it to full 3D scene generation (LION has these limitations in common with all the relevant baselines).

---

> > > ### Author Response · Authors · 2022-08-02
> > > **Thank You for the Feedback (3)**
> > >
> > > Finally, we would like to emphasize again that we added several additional results to the paper to make it overall stronger. We would like to point the reviewer to the additional message/comment that we sent to all reviewers for a more detailed overview over these experiments. Here, we summarize the most interesting additional experiments:
> > > - **Appendix F.2:** We are now running LION also on all 55 ShapeNet classes jointly without any conditioning. The qualitative results demonstrate that LION can even be trained on such highly diverse and multimodal data and still generate high-quality outputs with excellent mode coverage. To the best of our knowledge, there is no previous 3D generative model that successfully trains on such diverse 3D data and generates reasonable outputs.
> > > - **Appendices F.3 and F.4:** We also trained LION on the mug and bottle ShapeNet classes (149 and 340 training shapes, respectively) as well as 400 animal assets from the Turbosquid (https://www.turbosquid.com/) dataset. In all cases, LION can reliably generate coherent shapes. The experiment demonstrates that LION can also be trained on very small datasets.
> > > - **Appendix F.5:** To demonstrate the value to artists of being able to synthesize meshes and not just point clouds, we consider a downstream application: We apply Text2Mesh [49] on generated meshes from LION to additionally synthesize textures in a text-driven pers-sample manner, leveraging CLIP. This is only possible because of our SAP-based mesh reconstruction.
> > > - **Appendix F.6:** To demonstrate LION’s extendibility to even more tasks, we also trained a LION model where its latent diffusion models are conditioned on CLIP image embeddings, inspired by CLIP-Forge [34]. This allows LION to generate shapes based on text prompts, leveraging CLIP’s text encoder. Using CLIP’s image encoder, this additionally allows LION to infer and reconstruct 3D shapes from images.
> > >
> > > *If our reply is satisfactory and the additional results are appealing, we would like to kindly ask the reviewer to consider raising their score accordingly. Otherwise, we will be happy to further discuss. Thank you!*

---

> > > > ### Comment · Reviewer_FSJk · 2022-08-07
> > > > **Thanks for updates & clarifications**
> > > >
> > > > Thanks authors for the updates (particularly the experimental results on more classes), and the clarifications. I remain in favor of acceptance, and have upgraded my rating accordingly.

---

### Official Review · Reviewer_DZ2h · 2022-07-11

**Rating:** 5
**Confidence:** 3
**Soundness:** 2 fair
**Presentation:** 2 fair
**Contribution:** 2 fair

**Summary:**

This paper targets 3D shape generation and manipulation. Similar to previous works PVD and DPM [46,47] it uses a denoising diffusion model on point clouds. My understanding is that the novelty compared to these works is:
N1: using a global shape and a point feature (a.k.a. hierarchical latent space)
N2: combining the network with a shape as point [67] network to obtain a surface from the output point cloud.
The other claimed contribution are essentially results, that go beyond [46,47] in terms of performances (clearly better numbers on ShapeNet dataset) and applications (shape interpolation, guided generation)

**Questions:**

see weaknesses (clearer ablation compared with SoA)

**Limitations:**

yes

**Strengths And Weaknesses:**

This paper seems to clearly improve state of the art shape generation (note this is not my area of expertise, so I have to trust the authors the metrics and datasets are meaningful) the evaluation seems exhaustive and the exposition of the method is clear.

What is less clear to me is what the novelty actually is compared to [46,47] and what explains the performance gap. To me, the ablation tables 10 and 11 from the supplementary material would be the most interesting to understand this, but are still unclear:
- I think they should be moved to the main paper
- I couldn't match the results from these two tables to any in the main paper: to which should they be compared? is there a change in the method? or is this the effect of the randomness of training? (if so, some variance really seems necessary)
- having additional dimensions for the latent points seem to provide only a very small boost, I wonder if this isn't just noise that leads to this selection. Moreover, the fact that the selected value is D_h=1 (and using D_h=0 performs almost as well) is important for understanding why the method work and should be emphasized in the main paper
- from table 10, using shape latents also provides only a small boost, this together with the previous point really makes me wonder what part of the performance improvement is really due to the proposed hierarchical architecture v.s. other changes
- putting the two previous points together, I would really want to see numbers comparable to [46,47] in the ablation, to better understand the origin of the performance improvement, which is more unclear to me the more I look at the results.

To conclude, I am unsure about this paper, that falls slightly out of my expertise (not in diffusion models or generation). At first look it seems to really boost state of the art, but looking at the results and paper in more details, I couldn't figure out any clear, strong and well justified novelty (using shape as point is different but very low novelty, the hierarchical architecture seems different if not very original but in the end I am unsure which part of the improvement is really due to it) and I do not really understand where the gain in performance comes from (except the very large 300k hours of V100)

---

> ### Author Response · Authors · 2022-08-02
> **Thank You for the Feedback (1)**
>
> We would like to thank the reviewer for their positive feedback and for appreciating our state-of-the-art experimental results, our exhaustive evaluation, and our exposition.
>
> Below, we reply to the individual points raised by the reviewer (citations correspond to paper bibliography):
> - **Novelty:** First, we would like to point out the methodological novelty compared to PVD [46] and DPM [47]. PVD [46] trains a single diffusion model directly in the point cloud space. In contrast, we are training two diffusion models in a latent space in a hierarchical fashion. This has crucial advantages: As was shown in LSGM [58], it is easier to train diffusion models in smoothly regularized latent spaces together with additional encoder and decoder networks and this can translate to improved expressivity. But most importantly, our hierarchy is crucial: The vector-valued global shape latent essentially allows LION to switch between different modes of the data distribution, whose details are then modeled by the other point cloud latents. In that context, the architecture of how this is implemented is important, this is, the hierarchical conditioning via adaptive group normalization.
>
>     Furthermore, [47] also employs a latent variable, but trains (i) a Normalizing Flow in this latent space instead of a diffusion model and then (ii) uses a very weak diffusion model-based decoder for the point cloud output. This weak diffusion model operates on a per-point basis, thereby behaving entirely differently compared to the diffusion models in LION and PVD [46]. It is ultimately our novel architecture that results in our state-of-the-art results (also see discussion below).
> - **Ablation Experiments:** We will be happy to move Tables 10 and 11 into the main paper in the final version, when we have an additional page available. We agree that these are important experiments. We would also like to point out that we accidentally copied incorrect results into Tables 10 and 11 for our full model, thereby contributing to the confusion. We have now updated these tables and the numbers for the full model are now the same ones as for our LION in the main results Table 13.
> - **Performance Boost by Shape Latents:** The ablation experiment in Table 10 was designed to unambiguously show the effects of the different model components, like the shape latents. Removing the shape latents (line 2), performance drops, even when increasing the number of overall network parameters to compensate and make the comparison fair (line 5). However, as pointed out above, the shape latents’ main job is to switch between modes in the data when training on highly diverse and multimodal shape data, like in our experiment where we jointly train on 13 different ShapeNet classes without conditioning. In that case, it becomes evident that the shape latents encode crucial global shape information (see Figs. 7 and 25, where we keep the shape latent constant). However, to save computational resources the ablation experiment of Table 10 was run with a single-class LION model where this mode-switching effect of the shape latents is not necessary and the shape latents’ effect is hence smaller. Nevertheless, a non-negligible performance boost is even obtained in that case. We believe that the results of this small-scale ablation together with the experiment on diverse data with 13 different shape categories clearly demonstrate that the shape latents encode relevant information and are therefore an important component of the model. Generally, note that LION has been designed in particular with scalability to modeling complex and diverse data in mind and that’s where the shape latents become particularly important.
> - **Latent Point Features D_h:** We believe that D_h is indeed less important and the results in Table 11 validate that, showing only a slightly advantageous performance for D_h=1. Adding these additional features to the latent points is flexibility the model has. However, we chose the value D_h=1 early on in our experiments and then kept using it. It is possible that tuning D_h might further improve our large-scale models that were trained on 13 and 55 (see below) ShapeNet classes, respectively, and that the model could benefit from the additional expressivity for these more challenging setups. We leave this for future research and did not explore this to save computational resources.

---

> > ### Author Response · Authors · 2022-08-02
> > **Thank You for the Feedback (2)**
> >
> > - **Numbers in ablation experiments not comparable to [46,47]:** We believe that our ablations are run in a fair manner. In our ablation experiments in Table 10 and 11, we always used the same hyperparameters, compute budgets, etc., to be able to clearly filter out the effect of only removing individual model components. The 4th and 7th lines in Table 10 actually approximately correspond to PVD-like models [46] in terms of the overall architecture (i.e. no more latents). Our full model’s improved performance compared to these lines can, therefore, only be due to the additional model components and architecture innovations, i.e. the shape latents and the point latents with their corresponding diffusion models. Furthermore, it is expected that these lines do not exactly correspond to performance values reported in [46,47], because the works may have been trained with differently tuned hyperparameters, different numbers of network parameters, for a longer time, etc. However, in these ablations we wanted to run fair internal comparisons for our model. Finally, comparing the performance of LION to [46,47], we would like to point to the results of the 13-class experiment (Table 16). We are outperforming both baselines by a huge margin here, and the main difference compared to PVD [46] in these experiments is essentially our hierarchical VAE structure, as discussed. Consequently, we conclude that it is this architectural novelty that is responsible for the much stronger state-of-the-art performance.
> > - **Novelty regarding Shape-As-Points:** It is true that using Shape As Points (SAP) on LION’s output can be considered a relatively simple trick to extract meshes from the generated point cloud. However, we believe that this is significant and impactful, precisely because it is simple, but also highly relevant for practitioners (digital artists use meshes in practice, after all). Moreover, we are in fact not naively applying SAP on the generated point clouds, but we propose to fine-tune SAP on the LION generations, which makes a significant difference (see newly added Fig. 14). In particular, to be more robust against the specific kind of noise that is present during synthesis from LION we show how to fine-tune SAP on point clouds augmented  with our proposed *diffuse-denoise* process. Overall, we think that demonstrating this capability is in fact a novel and relevant contribution, and this has never been done before for any point cloud generation method, to the best of our knowledge. Due to the simplicity and relevance, we believe that this idea will be adopted by future works.
> >
> > We will try to further clarify these points in the final version of the paper, when we have an additional page available for the main text.
> >
> > Finally, we would like to mention that we added several additional results to the paper to make it overall stronger. We would like to point the reviewer to the additional message/comment that we sent to all reviewers for a more detailed overview over these experiments. Here, we summarize the most interesting additional experiments:
> > - **Appendix F.2:** We are now running LION also on all 55 ShapeNet classes jointly without any conditioning. The qualitative results demonstrate that LION can even be trained on such highly diverse and multimodal data and still generate high-quality outputs with excellent mode coverage. To the best of our knowledge, there is no previous 3D generative model that successfully trains on such diverse 3D data and generates reasonable outputs.
> > - **Appendices F.3 and F.4:** We also trained LION on the mug and bottle ShapeNet classes (149 and 340 training shapes, respectively) as well as 400 animal assets from the Turbosquid (https://www.turbosquid.com/) dataset. In all cases, LION can reliably generate coherent shapes. The experiment demonstrates that LION can also be trained on very small datasets.
> > - **Appendix F.5:** To demonstrate the value to artists of being able to synthesize meshes and not just point clouds, we consider a downstream application: We apply Text2Mesh [49] on generated meshes from LION to additionally synthesize textures in a text-driven pers-sample manner, leveraging CLIP. This is only possible because of our SAP-based mesh reconstruction.
> > - **Appendix F.6:** To demonstrate LION’s extendibility to even more tasks, we also trained a LION model where its latent diffusion models are conditioned on CLIP image embeddings, inspired by CLIP-Forge [34]. This allows LION to generate shapes based on text prompts, leveraging CLIP’s text encoder. Using CLIP’s image encoder, this additionally allows LION to infer and reconstruct 3D shapes from images.
> >
> > *If our reply is satisfactory and the additional results are appealing, we would like to kindly ask the reviewer to consider raising their score accordingly. Otherwise, we will be happy to further discuss. Thank you!*

---

### Official Review · Reviewer_Rrem · 2022-07-11

**Rating:** 7
**Confidence:** 3
**Soundness:** 3 good
**Presentation:** 4 excellent
**Contribution:** 3 good

**Summary:**

This paper proposes the hierarchical Latent Point Diffusion Model (LION) for 3D shape generation. Different from previous works where denoising diffusion models (DDMs) operate on the point clouds directly, LION is based on a VAE structure with a hierarchical latent space that includes a global shape latent space and a local point-structured latent space. Meanwhile, two hierarchical DDMs operate on such two latent spaces instead. Exhaustive experiments demonstrate LION outperforms several state-of-the-art models on multiple ShapeNet benchmarks. Besides, owing to the adopted VAE structure, it is convenient to transfer the LION model to other relevant tasks, which only requires fine-tuning the encoder. Combining with a surface reconstruction model like SAP enables the generation of smooth 3D meshes.

**Questions:**

1. For the shape interpolation, do you interpolate on both latent spaces, or just one?

**Limitations:**

1. A necessary discussion with some closely relevant papers is missing.
2. The explanation of the modest performance on MMD and COV metrics should be provided.

**Strengths And Weaknesses:**

Pros:
1. This paper proposes to employ denoising diffusion models (DDMs) on a hierarchical latent space, instead of raw point clouds. It's a pretty good idea and the effectiveness is supported by the impressive generation results and ablation study contained in the supp. material.
2. The adopted VAE structure makes the LION model flexible to transfer to different relevant tasks without re-training the computationally intensive DDMs. Besides, the experiments on shape denoising and voxel-guided synthesis also demonstrate the power of the hierarchical latent space.
3. LION model achieves state-of-the-art results on both single-class 3D generation and many-class 3D generation tasks on widely used ShapeNet benchmarks.
4. Combining with a surface reconstruction model like SAP enables the generation of smooth 3D meshes.

Cons:
1. The idea of employing DDMs on latent space and incorporating VAE structure is not new. Actually, several relevant papers [1,2] are adopting similar ideas in other domain. However, these papers are not cited and discussed in this manuscript.
2. The authors just select the most compelling results (on purpose) and put them in the main paper. As we all know, there are a lot of metrics for 3D shape generation evaluation. But the main paper only contains one metric, 1-NNA. Actually, the LION model doesn't always outperform baselines in terms of other metrics like MMD and COV according to Table 13 & 14 results in the supp. material.

[1] DiffuseVAE: Efficient, Controllable and High-Fidelity Generation from Low-Dimensional Latents. Kushagra Pandey, et al.
[2] Diffusion Autoencoders: Toward a Meaningful and Decodable Representation. Konpat Preechakul, et al.

---

> ### Author Response · Authors · 2022-08-02
> **Thank You for the Feedback (1)**
>
> We would like to thank the reviewer for their positive feedback and for appreciating both our methodological ideas as well as the strong experimental results.
>
> Below, we reply to the individual points raised by the reviewer (citations correspond to paper bibliography):
> - **Discussion of the papers DiffuseVAE [140] and Diffusion Autoencoders [60]:** We added a detailed discussion on both papers in Appendix F.9. Note that we put this into the Appendix for now due to the main text page limit. If our work is accepted for publication and an additional page for the main text is available, we will incorporate this discussion directly into the related work section in the main paper. We would like to mention that these works operate purely on images, in contrast to our work, which models 3D shapes and also follows an overall different architecture with both diffusion models in latent space. In fact, extending the latent diffusion model concept to 3D synthesis is not trivial. To this end, we proposed the concept of latent points, which is nicely complementary to the vector-valued shape latent. Note that Diffusion Autoencoders was cited already (citation [60] in the paper); however, we agree that a more detailed discussion is appropriate and thank the reviewer for pointing that out.
> - **Evaluation Metrics:** We agree that there are many metrics for quantifying 3D point cloud generation performance. For point cloud generative models, the most popular metrics are arguably 1-NNA, MMD and COV (see explanations in Appendix D.2). However, we did not choose 1-NNA simply because it provided the best results, but it is the only metric among those three metrics that reliably captures both shape diversity and also generation quality. As was pointed out by previous work (PointFlow) [31], MMD and COV can measure diversity and detect mode collapse, but fail to reliably quantify quality (see discussion in section 6.1 in [31]). This is also supported considering that simply with CD-based vs. EMD-based evaluations, different methods rank differently for COV and MMD. Furthermore, the highly relevant work PVD [46] also follows this approach and primarily relies on 1-NNA. Consequently, we also choose 1-NNA as our primary evaluation metric, where we consistently outperform all baselines for all experiments and achieve state-of-the-art results. This is explained in Sec. 5.1 and Appendix D.2. Nevertheless, since the broader 3D generation community also often uses MMD and COV, we also reported those metrics in the Appendix. However, these results are not as easy to interpret, considering the nature of the problematic MMD and COV metrics. The fact that for MMD and COV different methods score best in different experiments, with only small gaps, essentially only implies that none of the more competitive methods suffers from significant mode collapse. But these metrics cannot be used to make definite conclusions about detailed shape quality, unlike 1-NNA. That being said, our results for these metrics are usually still on-par with the best baselines, or we still largely outperform the baselines as for the challenging unconditional 13-class generation task (see Table 16).
> - **Interpolation:** Yes, we can confirm that we simultaneously interpolate in both the shape latent and also the point latent space. This is discussed in detail in Appendix B.3.

---

> > ### Author Response · Authors · 2022-08-02
> > **Thank You for the Feedback (2)**
> >
> > Finally, we would like to mention that we added several additional results to the paper to make it overall stronger. We would like to point the reviewer to the additional message/comment that we sent to all reviewers for a more detailed overview over these experiments. Here, we summarize the most interesting additional experiments:
> > - **Appendix F.2:** We are now running LION also on all 55 ShapeNet classes jointly without any conditioning. The qualitative results demonstrate that LION can even be trained on such highly diverse and multimodal data and still generate high-quality outputs with excellent mode coverage. To the best of our knowledge, there is no previous 3D generative model that successfully trains on such diverse 3D data and generates reasonable outputs.
> > - **Appendices F.3 and F.4:** We also trained LION on the mug and bottle ShapeNet classes (149 and 340 training shapes, respectively) as well as 400 animal assets from the Turbosquid (https://www.turbosquid.com/) dataset. In all cases, LION can reliably generate coherent shapes. The experiment demonstrates that LION can also be trained on very small datasets.
> > - **Appendix F.5:** To demonstrate the value to artists of being able to synthesize meshes and not just point clouds, we consider a downstream application: We apply Text2Mesh [49] on generated meshes from LION to additionally synthesize textures in a text-driven pers-sample manner, leveraging CLIP. This is only possible because of our SAP-based mesh reconstruction.
> > - **Appendix F.6:** To demonstrate LION’s extendibility to even more tasks, we also trained a LION model where its latent diffusion models are conditioned on CLIP image embeddings, inspired by CLIP-Forge [34]. This allows LION to generate shapes based on text prompts, leveraging CLIP’s text encoder. Using CLIP’s image encoder, this additionally allows LION to infer and reconstruct 3D shapes from images.
> >
> > *If our reply is satisfactory and the additional results are appealing, we would like to kindly ask the reviewer to consider raising their score accordingly. Otherwise, we will be happy to further discuss. Thank you!*

---

> > ### Comment · Reviewer_Rrem · 2022-08-07
> > **Response to the authors**
> >
> > After reading the authors' clarifications, I think their comments have resolved most of my concerns. I'm inclined to accept this paper and thus keep my original rating.

---

### Official Review · Reviewer_aV2c · 2022-07-12

**Rating:** 5
**Confidence:** 5
**Soundness:** 2 fair
**Presentation:** 3 good
**Contribution:** 3 good

**Summary:**

The paper proposes a hierarchical architecture for point cloud generation, which is composed of two diffusion models for shape latents and latent points.
The major difference between PVD and this paper is the hierarchical architecture and its ability to reconstruct corresponding meshes even though it comes from an off-the-shelf method SAP.
Both qualitative and quantitative results are performed to show the effectiveness.


**Questions:**

- Why choose PVCNN as the featue extration module? Author should include more insights and perform experimental results to show the reason.

- For the shape interpolation application, the author should include results of other baselines.

- Most generation methods can do single-view reconstruction. The paper may miss a comaprison on SVR from rgb or rgbd data like PVD does.

- As the proposed method can recontruct surface, I think it is possible to compare it with SOTA explicit/implicit surface-based methods like AutoSDF and Deep marching tetrahedra.

- In 'Single-Class Unconditional Generation', I can not see a clear improvement in quantitative results. What is the reason?

**Limitations:**

I only saw one 'slow sampling' limitation in the paper. However, this limitation comes from the generative model they use. So I do not think they fully showed their method's limitations. I did not see any negative societal impact.


**Strengths And Weaknesses:**

Strengths:
- Extensive results and ablations are given to show the effecitieness of the proposed method (in both the paper and supp).

- The paper is well written and easy to understand. It adequately discusses related works.

- In terms of generation quality, it truely beats several SOTA methods.

Weaknesses:

- The proposed pipeline is not particularly novel. It is more like a combination of PVCNN and the diffusion model. The mesh reconstruction method is also off-the-shelf. The main contribution might be the hierarchical generation architecture.

- The paper misses insights for choosing specific backbone for feature extraction. There should be a comparison between different backbones.

- SVR should be a common application for generative networks. However, It does not show any single-view recontruction results.

---

> ### Author Response · Authors · 2022-08-02
> **Thank You for the Feedback (1)**
>
> We would like to thank the reviewer for their positive feedback and for appreciating our extensive experiments and strong generation results. We are also happy that our paper came across as well written and easily understandable, and that our detailed related work discussion is welcomed.
>
> Below, we reply to the individual points and questions raised by the reviewer (citations correspond to paper bibliography):
> - **Technical Contributions:** It is correct that a point cloud denoising diffusion model (DDM) based on PVCNN layers exists, Point Voxel Diffusion (PVD) [46]. However, we are the first to explore the training of multiple DDMs in a latent space. Moreover, the idea of latent DDMs has never been used in 3D generation at all, to the best of our knowledge, and it is generally not obvious how to extend it to 3D in the point cloud case and how to structure the latent space. To this end, we introduce the concept of latent points and furthermore combine it with a complementary vector-valued latent variable, leveraging an efficient coupling via adaptive group normalization. Hence, we believe our architecture is novel and the experimental results clearly support that our design choices are crucial: We outperform all baselines and demonstrate that with this architecture we can scale to extremely diverse shape datasets, like modeling 13 or even 55 (see below) ShapeNet categories jointly without conditioning. These cases represent highly complex and multimodal distributions that need to be learnt, thereby stress-testing LION’s scalability. In particular for these challenging tasks we outperform previous DDM-based models for point clouds [46,47] by a huge margin (Table 16). This implies the importance and clear superiority of our novel hierarchical design, rendering it a relevant technical contribution. This is further supported by our ablation experiments. Ultimately, it is this novel technical contribution that leads to LION’s state-of-the-art results.
>
>     Furthermore, it is true that using Shape As Points (SAP) on LION’s output can be considered a relatively simple trick to extract meshes from the generated point cloud. However, we believe that this is significant and impactful, precisely because it is simple, but also highly relevant for practitioners (digital artists use meshes in practice, after all). That said, we are in fact not naively applying SAP on the generated point clouds, but we propose to fine-tune SAP on LION, which makes a significant difference (see newly added Fig. 14). In particular, to be more robust against the specific kind of noise that is present during synthesis from LION we show how to fine-tune SAP on point clouds augmented  with our proposed diffuse-denoise process. Overall, we think that demonstrating how to combine SAP with LION is in fact a novel and relevant contribution, and this has never been done before for any point cloud generation method, to the best of our knowledge. Due to the simplicity and relevance, we believe that this idea will be adopted by future works.
> - **Different Backbone Architectures:** In fact, we did experiment with different architectures at the early stages of the project. We tried not only point cloud processing networks based on Point-Voxel CNN (PVCNN) [79], but also Dynamic Graph CNN (DGCNN) [134] and Point Transformers [135]. However, the latter two led to worse performance. Consequently, we quickly converged to PVCNNs for our neural networks. We now added an ablation study in Appendix F.1 (Tables 19 and 20).
> - **Single View Reconstruction (SVR):** As pointed out in the paper, we designed LION primarily as a tool for digital artists, focusing on generation quality, controllability, and meshed outputs, and not as a geometry reconstruction method from images. However, we appreciate the reviewer’s suggestion to also explore SVR. In fact, we can easily extend LION to also perform SVR. To this end, we rendered 2D images from the 3D ShapeNet shapes, extracted the images’ CLIP [137] image embeddings, and trained LION’s latent diffusion models while conditioning on the shapes’ CLIP image embeddings. Now, at test time, we can take a single view 2D image, extract the CLIP image embedding, and generate corresponding 3D shapes, thereby effectively performing SVR. We added those experiments in Appendix F.6 and we see that we can achieve realistic 3D reconstructions. This approach is inspired by CLIP-Forge [34]. We leave a more quantitative evaluation and more fine-tuning to future research, but we believe that these experiments demonstrate that LION can indeed be easily extended to and used for SVR. Note that using CLIP’s text encoder, our approach additionally allows for text-guided generation as we now also demonstrate in Appendix F.6. PVD [1] cannot do this. Moreover, we only use RGB images for SVR, whereas PVD also requires depth information.

---

> > ### Author Response · Authors · 2022-08-02
> > **Thank You for the Feedback (2)**
> >
> > - **Question 1 - Why choose PVCNN as the feature extraction module?:** See above “Different backbone architectures”.
> > - **Question 2 - Baselines for Shape Interpolations:** As suggested, we added visualizations of shape interpolations for our two main diffusion model-based baselines, PVD [1] and DPM [2], in Appendix F.8. While PVD and DPM can also generate shape interpolations, LION’s interpolations appear less noisy and more coherent along the interpolation path. In particular, PVD and DPM break down when interpolating with their more complex 13-class models. In that case, they tend to generate very noisy samples all along the interpolation path. LION generates high-quality interpolations even in that challenging setting.
> > - **Question 3 - SVR from rgb or rgbd data as in PVD:** See above “Single View Reconstruction (SVR)”. Qualitatively, our results appear to be of similar quality as the results of PVD for that task, and at least as good or better than the results of AutoSDF. However, PVD requires rgb-d images, including depth, while LION can do SVR directly from rgb images without depth. Moreover, we would like the reviewer to keep in mind that LION was originally not designed for that task and that we outperform PVD by large margins on the other different unconditional generation tasks as well as on the controllable generation tasks (for instance, using voxel guidance). Furthermore, PVD does not incorporate mesh synthesis, which is highly useful in practice for artists.
> > - **Question 4 - Surface Reconstruction:** The reviewer proposed to compare LION to AutoSDF [30] and Deep Marching Tetrahedra (DMTet) [41] for surface reconstruction. AutoSDF seems to be designed for tasks like single view reconstruction and has been discussed above already. In contrast, DMTet is rather designed to reconstruct surfaces from voxel inputs or noisy point clouds. Hence, we agree that for our voxel and noise guidance experiments, DMTet can be an additional baseline. In Appendix F.7, we now also quantitatively compare to DMTet and we achieve similar or slightly better reconstruction results. However, DMTet is purely designed for this specific surface reconstruction task and is not a generative model that can generate shapes from scratch, like our LION can do. Unlike LION, DMTet can not do multimodal surface reconstructions and thereby does not capture the ambiguity of the task. In contrast, LION is a full generative model and significantly more versatile. Nevertheless, we perform as good as DMTet on this specific task.
> > - **Question 5 - Single-Class Unconditional Generation Performance:** There are multiple aspects: (i) As was pointed out in previous work (PointFlow) [31] and as we also discussed in Appendix D.2, COV and MMD are not reliable metrics to quantify generation quality. They can detect mode collapse, but beyond that are not fully reliable (this is also supported considering that simply with CD-based vs. EMD-based evaluations, different methods rank differently for COV and MMD). That is why we use 1-NNA as our primary evaluation metric, which reliably captures both quality and diversity, and here we clearly outperform all baselines in all experiments throughout the entire paper. Nevertheless, we also reported the COV and MMD results in the Appendix, as these metrics are still often used in the wider literature. The fact that for MMD and COV different methods score best in different experiments, with only small gaps, essentially only implies that none of the more competitive methods suffers from significant mode collapse. But these metrics cannot be used to make definite conclusions about detailed shape quality, unlike 1-NNA. (ii) Furthermore, LION was designed as a generative model that is also scalable to more challenging and diverse data. In fact, for the experiments where we jointly model 13 classes, we outperform previous methods on 1-NNA by large margins and are even mostly leading on MMD and COV (see Table 16). It is in these challenging multiclass generation tasks, where our novel hierarchical architecture with two diffusion models becomes most important.
> > - **Limitations:** We are happy to more extensively discuss our method’s limitations. For now, we added a discussion in Appendix F.11. In the final paper, when there is an additional page available for the main text, we may bring this into the main paper. In summary, additional limitations of LION beyond slow sampling are that it cannot directly generate textured shapes out of the box. Furthermore, it relies purely on geometry-based training and currently cannot profit from image-based training with differentiable rendering. Also, LION currently focuses on single object generation only. It would be interesting to extend it to full 3D scene generation (LION has these limitations in common with all the relevant baselines).

---

> > > ### Author Response · Authors · 2022-08-02
> > > **Thank You for the Feedback (3)**
> > >
> > > Finally, we would like to mention that we added several additional results to the paper to make it overall stronger. We would like to point the reviewer to the additional message/comment that we sent to all reviewers for a more detailed overview over these experiments. Here, we summarize the most interesting additional experiments:
> > > - **Appendix F.2:** We are now running LION also on all 55 ShapeNet classes jointly without any conditioning. The qualitative results demonstrate that LION can even be trained on such highly diverse and multimodal data and still generate high-quality outputs with excellent mode coverage. To the best of our knowledge, there is no previous 3D generative model that successfully trains on such diverse 3D data and generates reasonable outputs.
> > > - **Appendices F.3 and F.4:** We also trained LION on the mug and bottle ShapeNet classes (149 and 340 training shapes, respectively) as well as 400 animal assets from the Turbosquid (https://www.turbosquid.com/) dataset. In all cases, LION can reliably generate coherent shapes. The experiment demonstrates that LION can also be trained on very small datasets.
> > > - **Appendix F.5:** To demonstrate the value to artists of being able to synthesize meshes and not just point clouds, we consider a downstream application: We apply Text2mesh [49] on generated meshes from LION to additionally synthesize textures in a text-driven pers-sample manner, leveraging CLIP. This is only possible because of our SAP-based mesh reconstruction.
> > > - **Appendix F.6:** To demonstrate LION’s extendibility to even more tasks, we also trained a LION model where its latent diffusion models are conditioned on CLIP image embeddings, inspired by CLIP-Forge [34]. This allows LION to generate shapes based on text prompts, leveraging CLIP’s text encoder. Using CLIP’s image encoder, this additionally allows LION to infer and reconstruct 3D shapes from images.
> > >
> > > *If our reply is satisfactory and the additional results are appealing, we would like to kindly ask the reviewer to consider raising their score accordingly. Otherwise, we will be happy to further discuss. Thank you!*

---

> > > > ### Comment · Reviewer_aV2c · 2022-08-08
> > > > **Response to authors**
> > > >
> > > > The rebuttal alleviates my concerns and also answers other reviewers' questions. So I keep my attitude as borderline accept.

---

### Author Response · Authors · 2022-08-02
**Comment to all Reviewers - Additional Content and New Results**

We thank all reviewers for engaging in the review process. In our individual replies, we attempted to address specific questions and comments as clearly and detailed as possible.

Moreover, we added several additional results to the paper to make it overall stronger. These results have almost all been added into Appendix F. However, we will reorganize the structure for the final camera-ready version, if our paper is accepted, and potentially bring some of these results into the main text.

Here, we briefly summarize these additional experiments (citations correspond to paper bibliography):
- *Appendix F.1*: We added a further ablation experiment to study different point cloud processing neural network architectures, based on which LION is implemented. We find that the PVCNN architecture used in LION works best.
- *Appendix F.2*: We are now running LION also on all 55 ShapeNet classes jointly without any conditioning (we on purpose avoid conditioning to make the task challenging and thereby test LION’s scalability to complex and diverse multimodal datasets). The qualitative results (see Fig. 33) demonstrate that LION can even be trained on such highly diverse and multimodal data and still generate coherent outputs. Moreover, even very rare classes with only 39 training samples are reproduced, thereby validating excellent mode coverage. To the best of our knowledge, there is no previous 3D generative model that successfully trains on such diverse 3D data without any conditioning information and still generates reasonable outputs.
- *Appendix F.3*: To test whether LION can also be trained on very small data sets, we trained on ShapeNet’s bottle and mug classes with 148 and 340 shapes for training, respectively. As we see in Figs. 34 and 35, LION is also able to generate correct mugs and bottles in this very small training data set situation.
- *Appendix F.4*: Similarly, we also trained LION on 400 animal assets from the Turbosquid (https://www.turbosquid.com/) dataset (see Fig. 36). LION can reliably generate coherent animal meshes.
- *Appendix F.5*: To demonstrate the value to artists of being able to synthesize meshes and not just point clouds, we consider a downstream application: We apply Text2Mesh [49] on some generated meshes from LION to additionally synthesize textures in a per-sample text-driven manner, leveraging CLIP (see Figs. 37, 38, and 39). This is only possible because of our SAP-based mesh reconstruction.
- *Appendix F.6*: To demonstrate LION’s extendibility to even more relevant tasks, we also trained a LION model where its latent diffusion models are conditioned on CLIP image embeddings, inspired by CLIP-Forge [34]. This allows LION to (i) generate shapes based on text prompts, leveraging CLIP’s text encoder (Fig. 41). Using CLIP’s image encoder, this additionally allows LION to (ii) infer and reconstruct 3D shapes from images (Fig. 40). Unlike PVD [46], LION can use RGB images directly and does not require depth information. Note that this is a simple qualitative demonstration of LION’s extendibility. We did not perform any hyperparameter tuning here and believe that these results could be further improved with more careful tuning and training.
- *Appendix F.7*: We provide an additional comparison to Deep Marching Tetrahedra (DMTet)-based shape reconstruction for the experiment in which we synthesize shapes with voxel-based guidance. Quantitatively, LION marginally outperforms DMTet (Tab. 21). However, note that DMTet was specifically designed for such reconstruction tasks and is not a general generative model that could synthesize novel shapes from scratch without any guidance signal, unlike LION, which is a highly versatile general 3D generative model. Furthermore, as we demonstrated in the main paper, LION can generate multiple plausible de-voxelized shapes, while DMTet is fully deterministic and can only generate a single reconstruction.
- *Appendix F.8*: We now also provide additional shape interpolation results with the PVD [46] and DPM [47] baselines (Figs. 42 and 43). We find that LION provides the most coherent and clean interpolation results (see Figs. 28, 29, 30, 31 for reference), in particular when we interpolate using the model that was trained over 13 classes jointly, where PVD and DPM break down and generate noisy interpolations.
- For the 13-class ShapeNet experiment, we ran four additional competitive baselines (PointFlow, ShapeGF, SetVAE, PDGN), which we all outperform (see extended Tab. 16).

We hope that these additional results further strengthen LION’s position as state-of-the-art generative model of 3D shapes and demonstrate its flexibility and versatility.

---

### Author Response · Authors · 2022-08-06
**Discussion Period Ending Soon**

Dear Reviewers,

We would like to kindly remind you that the author-reviewer discussion period ends in 3 days. It would be great if you could have a look at our replies and additional experiments and let us know what you think and whether our replies are satisfactory or whether there are any further follow-up questions. We would appreciate any feedback and would be happy to further discuss.

Thank you very much,

The authors of “LION: Latent Point Diffusion Models for 3D Shape Generation”

---

### Meta-Review · Area_Chair_xBE7 · 2022-08-22

**Recommendation:** Accept
**Confidence:** Certain

**Metareview:**

This paper proposes a latent point diffusion model, LION, for 3D shape generation. The model builds two denoising diffusion models in the latent spaces of a variational autoencoder. The latent spaces combine a global shape latent representation with a point-structured latent space.  Comprehensive experiments are conducted to evaluate the performance of the proposed method.  The authors address the major concerns of the reviewers and strengthen the paper by providing additional empirical results. After the rebuttal, all four reviewers reach an agreement on accepting the paper because of the novelty and the state-of-the-art performance. The AC agrees with the reviewers and recommends accepting the paper.

**Award:**

No

---

### Decision · Program_Chairs · 2022-09-14

Accept